# The Weakest Link: A Nodal Tension Model for Local Network Resilience

## Abstract

The resilience of networked systems, defined by their ability to withstand targeted disruptions between a source and a target, is a critical concern in fields from ecology to infrastructure management. While spectral methods offer global insights, characterising the specific vulnerability of targeted pathways requires a more direct approach. In this paper, we frame this problem of local resilience as a continuous $L_1$ Nodal Tension linear program, built upon the classic dual of the maximum $s - t$ flow problem. While operations research establishes that this LP recovers the combinatorial minimum cut, we explicitly leverage its continuous polyhedral structure to transition into differential graph geometry and representation learning. Our framework formalises the structural redundancy gap between $L_1$ cuts and $L_2$ conductance, proves that the continuous bottleneck isolates negative discrete Ricci curvature, and derives a Local Cheeger Inequality to bound Message Passing Neural Network (MPNN) over-squashing. Furthermore, we extract the Clarke subdifferential of the capacity, establishing the Nodal Tension LP as a structurally sparse and Lipschitz-robust differentiable layer. We validate these theoretical properties computationally against standard algorithms. We then apply our model to a real-world conservation problem: assessing the connectivity of a grizzly bear corridor in the Canadian Rocky Mountains. The analysis reveals a structurally counter-intuitive feature of the landscape: the corridor's weakest link is not a remote bottleneck, but the local perimeter of the source protected area itself. By formalising this "null signal" for a classic choke point through our commute time bounds and subgradient analysis, we demonstrate our model's utility in translating LP bounds and graph theory into physical diagnostics. Our work provides a continuous and differentiable characterisation of local network resilience, bridging classical graph cuts with gradient-based representation learning. The source code to reproduce all results in the paper is available at https://anonymous.4open.science/r/tmlr-ldnr.

## 1 Introduction

In an era of unprecedented habitat fragmentation, analysing the connectivity of ecological landscapes has become a central challenge in conservation science (Vorosmarty et al., 2000). Current analytical approaches generally fall into two distinct paradigms. The first, and most common in applied conservation, focuses on path-finding, using methods like least-cost path analysis to identify optimal wildlife corridors between specific locations (Rosenberg et al., 1997; Hilty et al., 2020). The second, rooted in spectral graph theory, assesses the global health of the entire network, using metrics like algebraic connectivity to quantify its overall robustness to disruption (Chung, 1997). While both paradigms provide valuable ecological information, neither directly answers a crucial third question: *what is the resilience of the specific connection between two chosen protected areas, and where is its absolute weakest link?* In this paper, we argue that this question of localised, pairwise resilience represents a distinct and vital mode of network analysis. To this end, we introduce a framework to formalise this concept and demonstrate that focusing on this local resilience can lead to quantitative spatial diagnostics for conservation management.

Current methods for analysing network structure, while effective, often present a mismatch for such targeted questions. Spectral graph theory, for instance, provides global measures of connectivity through the algebraic

connectivity ($\lambda_2$, the Fiedler vector), which is linked to the graph's sparsest cut via Cheeger's inequality (Chung, 1997). While invaluable for assessing overall network robustness, these global measures do not directly identify the specific weakest link between a given source and target. Semidefinite programming (SDP) relaxations offer the tightest known approximations for the general sparsest cut problem (Arora et al., 2009), but their computational expense and conceptual complexity can be prohibitive for environmental practitioners. While discrete combinatorial algorithms efficiently compute the raw capacity of the minimum $s - t$ cut (Ford Jr & Fulkerson, 1956), their outputs are inherently non-differentiable. The critical gap is the need for a continuous formulation that not only identifies the local bottleneck, but natively integrates its structural limits into gradient-based machine learning pipelines.

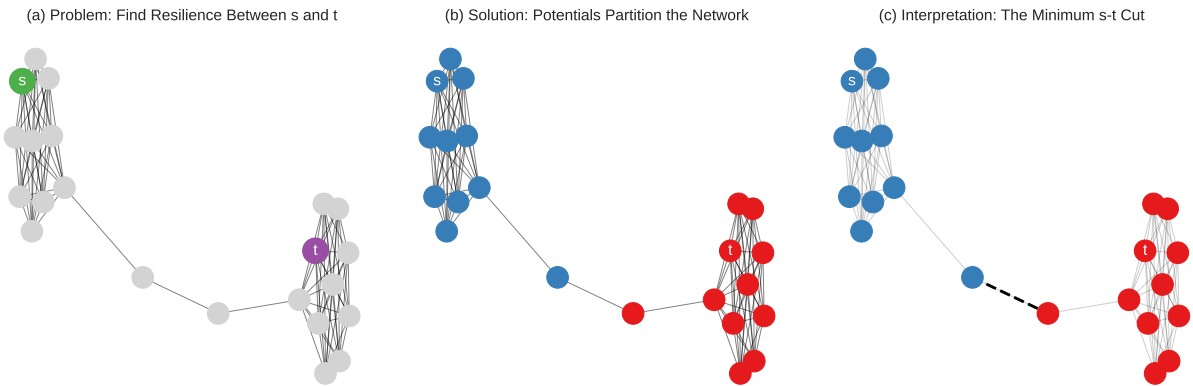

Figure 1: Illustration of our Nodal Tension model for local network resilience. **(a)** The problem is to quantify the resilience of the connection between a source node ($s$) and a target node ($t$) in a network containing a structural bottleneck. **(b)** Our linear program solves for a potential at each node. The optimal potentials, proven to be integer-valued ($\{0, 1\}$), partition the network's nodes into two distinct sets (blue and red). **(c)** The interpretation of this partitioning is the minimum $s - t$ cut. The single edge crossing the partition (dashed line) represents the most critical link, and its capacity, the optimal value $Z^*_{s,t}$, provides a direct, quantitative measure of the pathway's resilience.

We address this gap by formulating the local resilience problem as a continuous $L_1$ Nodal Tension linear program (illustrated in Figure 1). Instead of modelling flows, our primal LP assigns a continuous scalar potential $p_i$ to each node in the network. The objective is to minimise the total tension across all edges—defined as the sum of absolute potential differences—subject to a fixed potential gradient between a designated source $s$ and target $t$. While the duality between max-flow and min-cut is a well-established cornerstone of operations research (Ford Jr & Fulkerson, 1956; Kleinberg & Tardos, 2006), we explicitly utilise this continuous LP formulation because it preserves the polyhedral geometry of the structural bottleneck. Retaining this continuous structure is necessary to transition from raw combinatorial capacity computation to geometric analysis and differentiable programming.

Our framework is established in two phases. First, leveraging standard total unimodularity (Schrijver, 1998), we prove that the continuous optimisation natively recovers the exact combinatorial structure of the minimum cut, yielding binary node potentials. Second, we utilise this continuous formulation to interrogate the geometric and dynamical properties of the resulting local bottlenecks. We define the structural redundancy gap between $L_1$ topological cuts and $L_2$ electrical conductance (Doyle & Snell, 1984), map the continuous tension variables to negative discrete Ricci curvature, and derive a Local Cheeger Inequality to bound random walk commute times and localised over-squashing in Message Passing Neural Networks (MPNNs) (Alon & Yahav, 2021; Topping et al., 2022). Finally, we prove the model's exact polyhedral differentiability via the Clarke subdifferential (Clarke, 1990), establishing the Nodal Tension LP as a structurally sparse and Lipschitz-robust layer for gradient-based graph structure learning.

We conduct a systematic experimental validation to support our theory. We first computationally verify the model's correctness and exact integrality against standard discrete flow algorithms. Moving beyond combi-

natorial properties, we then empirically validate the geometric bounds derived in Section 4, demonstrating the structural redundancy gap, the isolation of negative curvature, and the exact extraction of sparse subgradients across a diverse range of graph topologies. Finally, we showcase the model's practical utility in a real-world scientific application. Using geospatial habitat data, we construct a large-scale connectivity network for grizzly bears in the Northern Rocky Mountains and apply our continuous model to identify the most structurally restrictive choke points in the habitat corridor between the Yellowstone and Selkirk ecosystems, providing actionable differentiable blueprints for conservation planning.

Our primary contributions in this paper are:

- A continuous formulation of local network resilience, the Nodal Tension model, grounded in the linear programming dual of the maximum flow problem.

- A polyhedral characterisation of this model, proving its exact equivalence to the minimum $s-t$ cut and the guaranteed integrality of its potentials.

- A theoretical framework establishing the bottleneck's geometric and dynamical limits, including the structural redundancy gap, isolation of negative discrete curvature, and local bounds on MPNN over-squashing.

- The derivation of the model's polyhedral differentiability and Lipschitz continuity, providing the basis for sparse backpropagation in parameterised graph models.

- A quantitative ecological finding from a real-world case study, demonstrating the absence of a remote interior corridor choke point, revealing that local perimeter severance dictates the functional isolation of the network, and demonstrating how polyhedral subgradients provide a sparse blueprint for spatial restoration.

This paper is structured as follows. We review prior work in Section 2. In Section 3, we define our Nodal Tension LP and outline its fundamental combinatorial properties. We present our geometric, dynamical, and continuous learning theorems in Section 4. The validation experiments and grizzly bear case study are detailed in Section 5. We conclude with a discussion in Section 6. Detailed proofs for Section 3 are provided in Appendix A, and proofs for Section 4 are provided in Appendix B.

## 2    Related Work

Our work is positioned at the intersection of several distinct but related fields. To properly situate our contribution, we first survey the broad, global perspectives on network resilience to establish the dominant paradigms. We then narrow our focus to the specific domain of ecological connectivity, highlighting the practical tools currently in use and their inherent focus on path-finding. Next, we examine the related but distinct field of graph partitioning, clarifying the difference between the unsupervised, global goal of community detection and our supervised, local goal. We then review the foundational principles of Spectral Graph Theory and optimisation-based methods to demonstrate how our work offers a simpler, more direct solution for local resilience. Finally, we contextualise our expansion into Graph Representation Learning, where diagnosing structural bottlenecks is critical for continuous message passing and differentiable programming.

### 2.1    Network Resilience and Robustness

The study of network resilience, broadly concerned with the ability of a system to maintain its function in the face of perturbations, has roots in ecology (Holling et al., 1973) and has become a cornerstone of modern network science (Newman, 2018). Much of the foundational work has focused on understanding the topological underpinnings of robustness. The differing properties of random graphs (Bollobás, 2011) versus scale-free networks (Barabási & Albert, 1999) provided a key insight: network structure dictates resilience. Scale-free networks, characterised by a few highly connected hubs, demonstrate strong tolerance to the random removal of nodes but are exceptionally fragile to targeted attacks on their hubs (Albert et al., 2000).

Resilience is often quantified using global metrics such as the percolation threshold required to fragment the network (Callaway et al., 2000; Cohen et al., 2000), changes in the size of the giant component under node or edge removal (Gao et al., 2016), or the degradation of network efficiency, a measure of communication cost (Latora & Marchiori, 2001). More dynamic approaches move beyond static topology to consider phenomena like cascading failures (Motter & Lai, 2002; Buldyrev et al., 2010) and the principles of network controllability (Liu et al., 2011; Ruths & Ruths, 2014). While this extensive body of work provides a standard framework for assessing the global stability of complex systems (Strogatz, 2001; Boccaletti et al., 2006), our focus is on a different, more localised question: the resilience of specific pathways between two designated points, a problem less commonly addressed by these global metrics.

## 2.2 Ecological Connectivity and Conservation

In conservation science, network analysis is a fundamental tool for modelling habitat connectivity and its impact on the long-term viability of wildlife populations (Urban & Keitt, 2001; Taylor et al., 1993). The concept of wildlife corridors, which serve to mitigate the adverse effects of habitat fragmentation, is a direct application of pathfinding and connectivity analysis in landscape graphs (Cameron et al., 2022; Haddad et al., 2003). In this paradigm, landscapes are often modelled as resistance surfaces, where each pixel is assigned a cost for a given species to traverse it (Spear et al., 2010; Adriaensen et al., 2003). Circuit theory, which draws an analogy between random walks on a graph and electrical circuits, has emerged as a particularly popular method. It can identify multiple important pathways and pinch-points by modelling the flow of "current" between habitat patches (McRae & Beier, 2007; Dickson et al., 2019). The outputs of these models are often validated through landscape genetics, which links the genetic differentiation in real populations to the landscape features that impede or facilitate gene flow (Manel et al., 2003; Storfer et al., 2007).

Our work offers a different perspective. While circuit theory identifies a probabilistic distribution of likely movement paths, it does not solve for the deterministic, worst-case vulnerability of the connection. It highlights the most-used routes, but not necessarily the cheapest-to-sever link. Our LP-based model is designed to answer this specific, adversarial question: what is the absolute weakest link—the minimum cut—that would guarantee a disconnection? This provides a complementary, decisive, and more direct measure of a corridor's resilience to disruption. As we formalise in Section 4.1, bridging these two paradigms by explicitly quantifying the structural redundancy gap between the $L_1$ minimum cut and the $L_2$ circuit conductance provides a strict classification of a bottleneck's physical geometry.

## 2.3 Graph Partitioning and Community Detection

The problem of partitioning a graph into densely connected subgraphs, or communities, is a central theme in network science, with applications ranging from social network analysis to functional genomics (Fortunato, 2010). The seminal algorithm of Girvan and Newman, based on the iterative removal of edges with high betweenness centrality, established a conceptual foundation for divisive clustering (Girvan & Newman, 2002). However, the most widely adopted paradigm is modularity optimisation, which seeks to find a partition that maximises a quality function comparing the density of intra-community edges to what would be expected in a random network (Newman, 2004). The Louvain method is an efficient heuristic for modularity maximisation that is widely used on large networks (Blondel et al., 2008). Other prominent approaches include information-theoretic methods like the map equation, which frames the problem in terms of finding a compressed description of a random walk (Rosvall & Bergstrom, 2008), and principled statistical inference using stochastic block models (Karrer & Newman, 2011). A crucial distinction defines our work: all these methods are global and unsupervised, designed to discover all communities in a network. Our problem is local and supervised: we aim to find the single, optimal partition that separates two pre-specified nodes, which places our work in the classic domain of minimum $s - t$ cuts (Cormen et al., 2022; Karger & Stein, 1996). While algorithms for this problem are foundational (Edmonds & Karp, 1972), our contribution is the primal-dual perspective that provides a direct, potential-based interpretation of this cut, preserving the continuous polyhedral geometry necessary for advanced sensitivity analysis (Section 4.4).

## 2.4 Spectral Graph Theory

Spectral graph theory provides the most established link between a graph's global combinatorial structure and its algebraic properties, primarily through the spectrum of the graph Laplacian matrix (von Luxburg, 2007; Mohar et al., 1991). The second smallest eigenvalue of the Laplacian, known as the algebraic connectivity or Fiedler value $\lambda_2$, is of critical importance as it quantifies how well-connected the graph is as a whole (Fiedler, 1973). The fundamental Cheeger's inequality provides the crucial, albeit often loose, bound connecting this algebraic quantity to a combinatorial one—the sparsest cut, or Cheeger constant, $\phi(G)$ (Cheeger, 2015; Chung, 1997). This connection forms the basis of spectral partitioning algorithms, which use the signs of the components of the Fiedler vector (the eigenvector corresponding to $\lambda_2$) to find approximations of the sparsest cut (Spielman & Teng, 2007; Pothen et al., 1990). This entire framework is inherently global, designed to find the most significant cutting in the entire graph. While these spectral approaches have been applied in ecology to develop global connectivity indices (Pascual-Hortal & Saura, 2006), their focus remains on assessing the overall network structure. Our work is complementary; instead of approximating a global property using spectral methods, we use a combinatorial optimisation approach to find an exact local property—the minimum $s - t$ cut—which provides a more direct answer to targeted conservation questions. Furthermore, as we demonstrate analytically in Section 4.3, global spectral bounds frequently fail to capture the severity of highly localised pathway restrictions, motivating our derivation of a Local Cheeger Inequality to bound $s - t$ commute times.

## 2.5 Optimisation-Based Approaches to Graph Cuts

Finding the exact sparsest cut is an NP-hard problem (Hartmanis, 1982), which has motivated extensive research into approximation algorithms using advanced optimisation. Linear programming (LP) relaxations provide a foundational approach. The seminal work of Leighton and Rao established an $O(\log n)$-approximation for the sparsest cut using an LP based on maximising concurrent multicommodity flows (Leighton & Rao, 1999). This result was later shown to have connections to the theory of metric embeddings into $L_1$ spaces (Linial et al., 1995), and has been the basis for faster algorithms (Leighton et al., 1991). To achieve tighter bounds, researchers have turned to the framework of semidefinite programming (SDP). The foundational work of Goemans and Williamson on the Max-Cut problem demonstrated the utility of SDP for graph optimisation (Goemans & Williamson, 1995). The current state-of-the-art approximation for the sparsest cut, achieving an $O(\sqrt{\log n})$ guarantee, was developed by Arora, Rao, and Vazirani using an SDP relaxation that embeds graph vertices onto a high-dimensional sphere (Arora et al., 2009), a result that has spurred a long line of research (Saranurak & Wang, 2019). Our work intentionally diverges from this line of research. While these advanced methods provide the best-known approximations for the NP-hard global sparsest cut problem, they are computationally expensive for the specific, local problem of s-t resilience, which is solvable in polynomial time. Our contribution is to show that by returning to a classic, simpler formulation—the direct dual of max-flow—we can achieve an exact, integral, and interpretable solution for this local problem. Crucially, by retaining this continuous LP formulation rather than relying on discrete combinatorial solvers, we unlock the ability to extract exact polyhedral subgradients (Section 4.4), directly bridging classical graph cuts with differentiable programming.

## 2.6 Graph Representation Learning and Differentiability

In modern machine learning, the topology of a graph governs the flow of information in Message Passing Neural Networks (MPNNs). It is a well-documented challenge that MPNNs suffer from "over-squashing" when forced to propagate feature vectors through severe structural bottlenecks (Alon & Yahav, 2021; Topping et al., 2022). Current diagnostic methods in Graph Structure Learning frequently rely on discrete differential geometry, such as Forman-Ricci curvature, to identify these constrained edges for targeted rewiring (Arnaiz-Rodríguez et al., 2022). However, discrete curvature is inherently a local metric; it identifies all negatively curved structural bridges but cannot isolate which specific bridge actually constrains the macroscopic connectivity between a given source and target. In Section 4, we resolve this diagnostic ambiguity, proving that the continuous Nodal Tension LP acts as an exact global filter for local discrete curvature (Section 4.2) and bounds MPNN Jacobian sensitivity (Section 4.3). Furthermore, to actively utilise structural bottlenecks in

end-to-end learning architectures, the metric must be differentiable. Because combinatorial algorithms lack continuous gradients, we leverage non-smooth continuous optimisation—specifically Danskin's Theorem and the Clarke subdifferential (Danskin, 1966; Clarke, 1990)—to prove that our $L_1$ formulation natively yields exact, structurally sparse backpropagation updates that are robust to out-of-distribution noise (Section 4.4).

## 3 Preliminaries: The $L_1$ Nodal Tension Foundation

In this section, we define our linear programming model for characterising local network resilience. While the duality between max-flow and min-cut is a foundational concept in operations research (Ford Jr & Fulkerson, 1956; Schrijver, 1998; Kleinberg & Tardos, 2006), we explicitly formulate the Nodal Tension LP here to establish the continuous $L_1$ framework required for our subsequent geometric and learning analysis in Section 4. We begin by describing our network representation and its environmental justification, then present the model, clarifying its scope and underlying assumptions. A summary of key symbols and notations is provided in Table 1.

Table 1: Key symbols and notations used throughout the paper.

| Symbol | Description |
|---|---|
| **Graph Representation** | |
| $G = (V, E, w)$ | A weighted, undirected graph representing the network. |
| $V$ | The set of $n$ nodes (e.g., habitat patches). |
| $E$ | The set of $m$ edges (e.g., movement pathways). |
| $w_{ij}$ | The positive weight (resistance) of the edge between nodes $i$ and $j$. |
| $s, t$ | The designated source and target nodes in $V$. |
| **Nodal Tension LP Model** | |
| $p_i$ | The potential (a continuous scalar variable) assigned to node $i$. |
| $d_{ij}$ | The tension (a continuous scalar variable) on the edge between nodes $i$ and $j$. |
| $Z_{s,t}^*$ | The optimal capacity of the local minimum $s - t$ cut (Nodal Tension objective). |
| **Theoretical Concepts** | |
| $S$ | A subset of nodes defining one side of an $s - t$ cut, where $s \in S$. |
| $E(S, V \setminus S)$ | The set of edges crossing the cut $(S, V \setminus S)$. |
| $\mathcal{C}^*$ | The set of all valid active minimum $s - t$ cuts (minimal separating surfaces). |
| $d^{C^*}$ | The binary indicator tension vector for a specific active cut $C^*$. |
| $\lambda_2$ | The algebraic connectivity (Fiedler value) of the graph. |
| $\phi(G)$ | The Cheeger constant (sparsest global cut) of the graph. |
| $\text{vol}(U)$ | The volume of a set of nodes $U$ (sum of weighted degrees). |
| **Graph Geometry, Dynamics, and Learning** | |
| $C_{s,t}$ | The effective conductance ($L_2$ average-case connectivity) between $s$ and $t$. |
| $R_{s,t}$ | The effective resistance between $s$ and $t$ (inverse of $C_{s,t}$). |
| $\Delta_{s,t}$ | The structural redundancy gap between the $L_1$ minimum cut and $L_2$ conductance. |
| $\mathbf{F}(u, v)$ | The discrete (Forman-Ricci) curvature of the edge $(u, v)$. |
| $\Phi_{s,t}^*$ | The local Nodal Conductance evaluated across the $s - t$ bottleneck. |
| $H(s, t)$ | The expected commute time of a random walk across the bottleneck. |
| $\partial Z_{s,t}^*(W)$ | The exact Clarke subdifferential of the capacity with respect to edge weights. |
| $W_\theta$ | Differentiable continuous edge weights parameterised by a neural network model $\theta$. |

### 3.1 Network Representation

We model an environmental landscape as a weighted, undirected graph $G = (V, E, w)$, where $V$ is a set of $n$ nodes and $E$ is a set of $m$ edges. In the context of ecological connectivity, nodes represent discrete patches of suitable habitat or key landscape locations, while edges represent potential pathways for movement or

dispersal between these nodes. Each edge $(i, j) \in E$ is assigned a positive weight $w_{ij} > 0$, which represents the ecological *resistance* for a species to traverse that pathway (Spear et al., 2010; Adriaensen et al., 2003). A high weight may correspond to a perilous or energy-intensive path (e.g., crossing a road or an area with poor cover), while a low weight signifies a safe and easy connection (e.g., a forested riparian corridor). This resistance-based framework is a standard paradigm in landscape ecology for quantifying connectivity. Our model is designed to operate on this representation to solve for the minimum-resistance cut between a designated source node $s \in V$ and a target node $t \in V$. While ecological resilience is a broad concept that includes recovery and adaptation, our model focuses on a specific, critical component: the structural robustness of the connection, which we define as its vulnerability to complete severance.

The scope of our model is intentionally focused on local, s-t resilience, guided by several key assumptions that align with common environmental challenges.

1. **Static Network:** We assume that the network topology and edge weights are static over the period of analysis. This reflects the model's application as a tool for strategic planning, such as identifying locations for permanent wildlife corridors or prioritising land acquisitions, rather than tracking real-time dynamic movements.

2. **Undirected Edges:** We model connections as bidirectional, which is appropriate for many mobile terrestrial species that can travel back and forth between habitat patches (Urban & Keitt, 2001). We acknowledge that for systems governed by directional processes, such as river networks (hydrological flow) or wind-based seed dispersal, a directed graph formulation would be necessary. Crucially, if the model is extended to a directed graph with asymmetric edge weights to capture anisotropic movement, the constraint matrix preserves the structure of a directed node-edge incidence matrix. Therefore, the total unimodularity (TUM) property and the strict integrality guarantees established in Section 3.3 firmly hold for the directed formulation as well.

3. **Local Resilience:** Our model is designed to answer a local question ("What is the weakest link between A and B?") rather than a global one ("How connected is the entire system?"). This is motivated by the fact that many conservation and management questions are inherently local, focusing on the viability of specific corridors between protected areas or the pathway of a pollutant from a specific source to a sensitive downstream location (Rosenberg et al., 1997; Hilty et al., 2020).

## 3.2 Linear Programming Formulation

We adopt the classic LP dual of the max-flow problem to find the minimum-cost cut between $s$ and $t$. While combinatorial algorithms (e.g., Push-Relabel (Goldberg & Tarjan, 1988; Kleinberg & Tardos, 2006; Cormen et al., 2022)) are vastly faster for computing raw cut capacity, they act as discrete black boxes. We explicitly formulate this as a continuous linear program because its continuous primal-dual structure is the key that unlocks the exact polyhedral differentiability and geometric interpretations derived in Section 4. The objective is to minimise the total weighted tension across all edges, subject to a fixed separation in potential between the source and target.

The Nodal Tension LP is defined as:

$$\text{Minimise} \quad Z = \sum_{(i,j) \in E} w_{ij} d_{ij} \tag{1}$$

subject to the constraints:

$$d_{ij} \geq p_i - p_j \qquad\qquad \forall (i, j) \in E \tag{2}$$
$$d_{ij} \geq p_j - p_i \qquad\qquad \forall (i, j) \in E \tag{3}$$
$$p_s = 0 \tag{4}$$
$$p_t = 1 \tag{5}$$

The variables in this optimisation are the node potentials $\{p_i\}_{i \in V}$ and the edge tensions $\{d_{ij}\}_{(i,j) \in E}$. The objective function equation 1 minimises the sum of weighted tensions across the network. The first two

constraints, equation 2 and equation 3, together enforce that the tension variable $d_{ij}$ must be at least the absolute difference in potential between its incident nodes, i.e., $d_{ij} \geq |p_i - p_j|$. In a minimisation problem, this inequality will be met with equality at the optimal solution. The final two constraints, equation 4 and equation 5, anchor the problem by fixing the potentials of the source and target nodes, creating a "potential gradient" across the network.

The optimal value of this LP, $Z_{s,t}^*$, represents the cost of the cheapest set of edges that must be removed to disconnect all paths from $s$ to $t$. It thus provides a single, quantitative score for the resilience of the corridor, where a higher value indicates a more robust and resilient connection. This value, $Z_{s,t}^*$, can be interpreted as a direct measure of the corridor's capacity to absorb disturbance; it is the minimum cumulative "damage" (in the form of increased resistance) that the landscape must sustain before the connection between source and target is functionally severed, directly operationalising the concepts of ecological resilience (Holling et al., 1973).

### 3.3  Fundamental Properties of the LP

In this subsection, we review the fundamental properties that underpin our Nodal Tension model. By leveraging classic results from operations research and polyhedral combinatorics, we state that the optimal value of our linear program is equal to the capacity of the minimum $s - t$ cut, and we establish the strict integrality of the optimal node potentials. These properties ensure that our continuous LP formulation recovers the combinatorial structure of the local bottleneck, serving as the necessary stepping stone for the geometric and dynamical analysis presented in Section 4.

**Theorem 3.1** (Equivalence to Minimum s-t Cut (Ford Jr & Fulkerson, 1956))**.** *Let $G = (V, E, w)$ be a weighted, undirected graph with edge weights $w_{ij} > 0$ interpreted as capacities. For any two distinct nodes $s, t \in V$, the optimal value, $Z_{s,t}^*$, of the Nodal Tension LP (Equations 1-5) is exactly equal to the capacity of the minimum weighted $s - t$ cut.*

*Proof Sketch.* This result is a direct consequence of standard network flow duality. The proof proceeds in two parts by leveraging the max-flow min-cut theorem and strong LP duality. First, we show $Z_{s,t}^* \leq \text{min-cut}(s,t)$ by construction. Let $(S^*, V \setminus S^*)$ be the minimum $s - t$ cut, and let $C^*$ denote the set of edges crossing this partition. We construct a feasible LP solution by setting $p_i = 0$ for all $i \in S^*$ and $p_i = 1$ for all $i \in V \setminus S^*$. For this solution, the only non-zero tensions are on the edges crossing the cut, where $|p_i - p_j| = 1$. The objective value is $\sum_{(i,j) \in C^*} w_{ij} \cdot 1 = \text{min-cut}(s,t)$. Since the true optimum $Z_{s,t}^*$ must be less than or equal to the value of any feasible solution, $Z_{s,t}^* \leq \text{min-cut}(s,t)$.

Second, we show $Z_{s,t}^* \geq \text{min-cut}(s,t)$. The Nodal Tension LP is the exact linear dual of the maximum $s - t$ flow problem. The max-flow min-cut theorem states that the value of the maximum flow is equal to the capacity of the minimum cut. By the strong duality theorem of linear programming, the optimal value of a primal problem (our LP) is equal to the optimal value of its dual (the max-flow problem). Therefore, $Z_{s,t}^*$ is equal to the max-flow, which is equal to the min-cut. As $Z_{s,t}^*$ is both $\leq$ and $\geq$ the min-cut, it must be equal. (See Appendix A.1 for the detailed proof). $\square$

**Environmental Interpretation:** This theorem provides a concrete meaning for the model's output. By mapping the LP optimum to the exact $L_1$ capacity of the network's weakest link, we establish that the optimal value $Z_{s,t}^*$ is not an abstract index but a direct, quantitative measure of a corridor's structural resilience. For a wildlife corridor, it represents the minimum cumulative resistance that must be overcome—or the minimum ecological damage that must be inflicted—to guarantee the severance of the connection between two critical habitats (Adriaensen et al., 2003). This value operationalises the concept of ecological resilience (Holling et al., 1973) by quantifying the corridor's capacity to absorb landscape change before failing. This single value allows for objective, comparative analyses; for instance, conservation planners can compare the resilience scores ($Z_{s,t}^*$) of several proposed corridor designs to identify the most robust option. Furthermore, it provides a crucial tool for scenario planning. By modifying edge weights to reflect future land-use changes and re-calculating $Z_{s,t}^*$, managers can quantify the projected impact on corridor viability, a central challenge in conservation planning (Haddad et al., 2003). A higher value of $Z_{s,t}^*$ signifies a more resilient connection.

Crucially, quantifying this exact bottleneck capacity forms the basis for evaluating the corridor's geometric fragility and dynamical limits, which we formalise in Section 4.

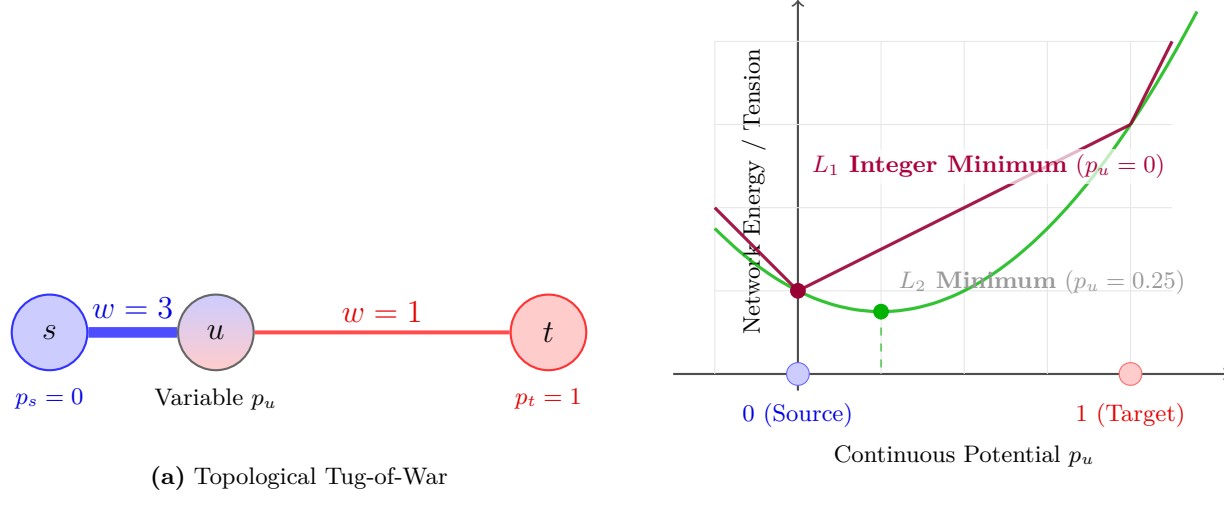

**(a)** Topological Tug-of-War

**(b)** Continuous Energy Landscape

Figure 2: Visualisation of the geometric mechanism driving the integrality of Nodal Tension potentials (Theorem 3.2). **(a)** The topological tug-of-war on a local path. Node $u$ is suspended between the blue source $s$ ($p_s = 0$) via a strong edge ($w = 3$) and the red target $t$ ($p_t = 1$) via a weak edge ($w = 1$). Its continuous potential $p_u$ seeks to minimise tension. **(b)** The continuous optimisation landscapes. Standard $L_2$ Dirichlet energy, explicitly evaluated as $3(p_u - 0)^2 + 1(1 - p_u)^2$ (green curve), produces a smooth, differentiable parabola whose fractional minimum ($p_u = 0.25$) "smears" information along the path. Conversely, the $L_1$ Nodal Tension objective, evaluated as $3|p_u - 0| + 1|1 - p_u|$ (purple piecewise linear shape), creates a "sharp" asymmetric geometry. Because the slopes are linear and mismatched ($3 \neq 1$), the absolute minimum cannot exist in the fractional domain; it slides down the steepest gradient and lands directly into the non-differentiable boundary $p_u = 0$ (the left-side "V" shape). This geometric sharpness is the physical counterpart to the Total Unimodularity proof, forcing continuous variables into discrete binary $\{0, 1\}$ states and uniquely identifying the topological minimal surface.

**Theorem 3.2** (Integrality of Potentials (Schrijver, 1998)). *For a graph with arbitrary positive edge weights $w_{ij}$ and integer separation constraints ($p_s = 0, p_t = 1$), an optimal vertex solution to the Nodal Tension LP exists where every node potential $p_i \in \{0, 1\}$.*

*Proof Sketch.* This property follows directly from the total unimodularity (TUM) of network matrices. A key theorem states that if an LP's constraint matrix $A$ is totally unimodular and the right-hand side vector $b$ is integer, then all of its vertex solutions are integer, regardless of whether the objective coefficients (the edge weights $w_{ij}$) are integers. The constraint matrix of our LP, defined by the relationships between potentials and tensions (e.g., $d_{ij} - p_i + p_j \geq 0$), inherently shares the structure of a directed node-edge incidence matrix (by arbitrarily orienting edges to define potential differences). It is a classic result in polyhedral combinatorics that such matrices are totally unimodular (Schrijver, 1998). Since our right-hand side vector $b$ (composed of 0s and 1s) is integer, any optimal solution found by a vertex-following algorithm like the Simplex method will have integer potentials. Furthermore, to minimise the objective $\sum w_{ij}|p_i - p_j|$ where $w_{ij} > 0$, any integer potential $p_k^*$ must lie within the range of the fixed potentials $[0, 1]$. An integer value outside this range would increase the total tension unnecessarily. Geometrically, the sharp, piecewise linear nature of the $L_1$ objective actively rejects fractional states within this range, sliding down the steepest gradient to land directly into these non-differentiable integer boundaries (as demonstrated in Figure 2). Therefore, the combination of polyhedral integrality and $L_1$ sharpness forces all optimal continuous potentials into the discrete binary set $\{0, 1\}$. (See Appendix A.2 for the detailed proof). □

**Environmental Interpretation:** This theorem is crucial for practical application because it proves that the model's output resolves into a decisive, binary classification of the landscape. The LP automatically partitions all nodes into two distinct sets: those with potential 0 (the source-set) and those with potential 1 (the target-set). This provides an unambiguous, discrete partition in contrast to the probabilistic or gradient-based outputs of methods like circuit theory (McRae & Beier, 2007). This is particularly valuable for defining clear management units or conservation boundaries on the ground (Cameron et al., 2022). For example, the set of all nodes with $p_i = 0$ can be designated as the "source-zone" of the corridor, allowing managers to apply consistent land-use policies across a well-defined geographical area to protect the integrity of the weakest link. However, from a machine learning perspective, while this theorem guarantees that the final vertex solution is a discrete $\{0, 1\}$ partition, formulating it as an LP preserves the continuous geometry of the underlying feasible polytope . This continuous geometric structure is what differentiates our formulation from simple discrete indicator functions or combinatorial black boxes, allowing us to derive exact subgradients and curvature metrics in Section 4.

**Theorem 3.3** (LP-Spectral Bound). *Let $Z_{s,t}^*$ be the optimal value of the Nodal Tension LP for a given source $s$ and target $t$ on a weighted graph. Let $(S, V \setminus S)$ be the corresponding minimum $s - t$ cut, with $vol(S) \leq vol(V)/2$. The algebraic connectivity of the graph, $\lambda_2$, is bounded above by:*

$$\lambda_2 \leq 2\frac{Z_{s,t}^*}{vol(S)}$$

*Proof Sketch.* This proof connects our local $L_1$ result to the global Cheeger inequality for weighted graphs. The weighted Cheeger inequality states that $\lambda_2 \leq 2\phi(G)$, where $\phi(G)$ is the weighted Cheeger constant, defined as the minimum cut sparsity over all possible partitions of the graph. The sparsity of a cut $(U, V \setminus U)$ is the ratio of the total weight of the edges crossing the cut to the volume of the smaller partition, where volume is the sum of weighted degrees of the nodes in the partition. Our LP finds the value of a specific cut, with capacity $Z_{s,t}^* = w(S, V \setminus S)$. The sparsity of this specific cut is $\frac{Z_{s,t}^*}{\text{vol}(S)}$. Since $\phi(G)$ is the minimum sparsity over all possible cuts, the sparsity of our specific cut must be greater than or equal to $\phi(G)$. Thus, $\phi(G) \leq \frac{Z_{s,t}^*}{\text{vol}(S)}$. Substituting this into the Cheeger inequality gives the final result: $\lambda_2 \leq 2\phi(G) \leq 2\frac{Z_{s,t}^*}{\text{vol}(S)}$. (See Appendix A.3 for the detailed proof). $\square$

**Theoretical Limitations and Transition:** While this theorem connects our local min-cut capacity to the global algebraic connectivity $\lambda_2$, this bound is often analytically uninformative in practice. When a local cut is highly unbalanced—meaning the volume of the separated component $\text{vol}(S)$ is extremely small compared to the overall network—the ratio $Z_{s,t}^*/\text{vol}(S)$ becomes exceptionally loose. In our case study, the model found a cut with a very low resilience score ($Z_{s,t}^* = 3.07$) but also a very small partition size $\text{vol}(S)$. This results in a correct, but very loose, upper bound on the global connectivity, rendering it practically ineffective for diagnosing the specific vulnerability of the $s - t$ pathway. This limitation highlights why global spectral gaps ($\lambda_2$) are inadequate for assessing local resilience. To quantify the fragility of such bottlenecks—and to understand their impact on random walks and graph representation learning—we must move beyond loose global bounds. In Section 4, we establish a new framework that directly models the structural redundancy, discrete geometry, and local dynamical limits of the Nodal Tension bottleneck.

## 4 The Geometry and Learning of Local Resilience

Building on the $L_1$ Nodal Tension model defined in Section 3, we analyse the geometric and dynamical properties of the resulting local bottlenecks. As demonstrated by the analytical limitations of the global spectral bound (Theorem 3.3), evaluating the specific vulnerability of an $s-t$ pathway requires moving beyond macroscopic matrix eigenvalues to directly interrogate the structural partition. While discrete algorithms compute cut capacities efficiently, the continuous linear programming formulation allows us to examine the exact polyhedral structure of the optimal solution. In this section, we relate the continuous LP variables to the differential geometry, random walk dynamics, and algorithmic robustness of the graph.

We structure this analysis into four parts. First, we formalise the structural redundancy gap between the $L_1$ topological bottleneck and $L_2$ average-case metrics to physically classify the nature of the vulnerability

(Section 4.1). Second, we show that the optimal tension variables correlate with negative discrete Ricci curvature, establishing a link between the continuous cut formulation and local bottleneck geometry (Section 4.2). Third, to address the limitations of the global spectral bound, we derive a Local Cheeger Inequality that bounds random walk commute times exclusively across the cut, with direct implications for information flow and over-squashing in Message Passing Neural Networks (Section 4.3). Finally, we analyse the model's polyhedral differentiability and its Lipschitz continuity under $L_\infty$ perturbations to edge weights (Section 4.4). This sequence bridges the model's initial environmental scope with a bounded basis for Graph Structure Learning.

## 4.1 The Structural Redundancy Gap ($L_1$ vs. $L_2$)

In both landscape ecology and graph representation learning, network connectivity is conventionally evaluated using either $L_1$-based topological metrics, such as the minimum cut (Ford Jr & Fulkerson, 1956), or $L_2$-based electrical metrics, such as effective resistance and Circuit Theory (Chandra et al., 1989). The optimal value of our Nodal Tension LP, $Z_{s,t}^*$, represents the $L_1$ bottleneck capacity, identifying the worst-case structural failure point of a pathway. Conversely, the effective conductance $C_{s,t}$ (the inverse of effective resistance $R_{s,t}$) measures the average-case flow distribution across all available parallel paths by minimising the $L_2$ Dirichlet energy of the network (Spielman & Srivastava, 2008).

While these two metrics are foundational, the explicit difference between them is rarely operationalised as a diagnostic feature in spatial analysis. The literature frequently treats minimum cuts and electrical networks as divergent or competing methodologies (Adriaensen et al., 2003). However, relying on the $L_1$ capacity $Z_{s,t}^*$ in isolation leaves a diagnostic gap: a cut with a specific low capacity could manifest structurally as a single, isolated critical edge, or it could be composed of numerous parallel, highly constrained edges.

By bounding the algebraic gap between the $L_1$ weakest link and the $L_2$ effective conductance, we can quantify the structural redundancy of the local pathway. Establishing this relationship is necessary because it allows us to classify the physical shape of a bottleneck—differentiating a strict topological bridge from a diffuse region of high resistance—which directly informs both targeted conservation strategies and the behaviour of graph learning algorithms.

To formalise this gap, we first define the standard $L_2$ electrical metric. For a graph $G = (V, E, w)$, let $p \in \mathbb{R}^{|V|}$ denote a vector of continuous node potentials. By Dirichlet's Principle, the effective conductance $C_{s,t}$ between a source $s$ and target $t$ is the minimum $L_2$ Dirichlet energy of the network, subject to unit boundary conditions:

$$C_{s,t} = \min_{p \in \mathbb{R}^{|V|}, p_s=0, p_t=1} \sum_{(i,j) \in E} w_{ij}(p_i - p_j)^2 \qquad (6)$$

The effective resistance is its reciprocal, $R_{s,t} = 1/C_{s,t}$. This formulation can be directly contrasted with our Nodal Tension LP optimal value $Z_{s,t}^*$. As shown in the proof of Theorem 3.1 (Appendix A.1), the LP objective under separation constraints is equivalent to minimising the $L_1$ total variation of the exact same node potentials:

$$Z_{s,t}^* = \min_{p \in \mathbb{R}^{|V|}, p_s=0, p_t=1} \sum_{(i,j) \in E} w_{ij}|p_i - p_j| \qquad (7)$$

The algebraic distinction between the exponent in these two objectives governs their physical and algorithmic behaviour. The quadratic penalty in the $L_2$ formulation (Equation 6) heavily penalises large potential differences across any single edge. To minimise the total energy, the underlying flow is forced to distribute across all available parallel pathways. Therefore, $C_{s,t}$ measures the aggregate average-case connectivity, increasing whenever new redundant routes are added.

Conversely, the linear $L_1$ penalty in Equation 7 imposes no such distribution requirement. It evaluates the absolute minimal capacity required to sever the $s-t$ connection, ignoring the presence of parallel redundancy once the critical bottleneck is found. Understanding this mechanical difference is necessary to prove how the gap between them quantifies structural vulnerability.

**Lemma 4.1** ($L_1$-$L_2$ Capacity Bound). *For any weighted, undirected graph $G = (V, E, w)$ and distinct nodes $s, t \in V$, the $L_2$ effective conductance $C_{s,t}$ is bounded above by the $L_1$ optimal Nodal Tension $Z_{s,t}^*$:*

$$C_{s,t} \leq Z_{s,t}^* \tag{8}$$

*Furthermore, the magnitude of the redundancy gap, $\Delta_{s,t} = Z_{s,t}^* - C_{s,t}$, is non-negative and quantifies the structural redundancy of the pathway: $\Delta_{s,t}$ grows as the capacity of parallel, edge-disjoint routes bypassing the primary bottleneck increases.*

*Proof Sketch.* The upper bound is established directly via the variational principle of Dirichlet energy. By definition (Equation 6), $C_{s,t}$ is the global infimum of the $L_2$ objective over all valid continuous potential vectors $p \in \mathbb{R}^{|V|}$ satisfying $p_s = 0, p_t = 1$.

Let $p^*$ be the optimal vertex solution to the Nodal Tension LP. By Theorem 3.2, we know $p^*$ is binary ($p_i^* \in \{0, 1\}$). Because $p^*$ satisfies the boundary conditions, it is a feasible candidate for the $L_2$ minimisation. Crucially, for any binary variables, the square of their difference is identical to their absolute difference. Since $(p_i^* - p_j^*) \in \{-1, 0, 1\}$, it follows that $(p_i^* - p_j^*)^2 = |p_i^* - p_j^*|$. Therefore, evaluating the $L_2$ Dirichlet objective at the $L_1$ optimal potentials $p^*$ yields the $L_1$ minimum cut capacity:

$$\sum_{(i,j) \in E} w_{ij}(p_i^* - p_j^*)^2 = \sum_{(i,j) \in E} w_{ij}|p_i^* - p_j^*| = Z_{s,t}^* \tag{9}$$

Because $C_{s,t}$ is the absolute minimum over all continuous vectors, it must be less than or equal to the energy of any specific feasible vector, including $p^*$. Thus, $C_{s,t} \leq Z_{s,t}^*$.

To quantify the physical meaning of the gap $\Delta_{s,t} \geq 0$, we invoke Kirchhoff's Current Law and Ohm's Law across the minimum cut $C^*$ (Doyle & Snell, 1984; Bollobás, 2011). Letting $\phi^*$ be the unique continuous $L_2$ minimizer, the total flow must equal $C_{s,t}$. Algebraic substitution allows us to express the gap as $\Delta_{s,t} = \sum_{(u,v) \in C^*} w_{uv}\big(1 - (\phi_u^* - \phi_v^*)\big)$.

Because $\phi^*$ is bounded, this term is non-negative. More importantly, it proves that $\Delta_{s,t} = 0$ if and only if the entire unit potential difference drops directly across the min-cut edges ($\phi_u^* - \phi_v^* = 1$). This represents a strict topological bridge with infinite conductance (zero resistance) everywhere else. Conversely, any positive gap ($\Delta_{s,t} > 0$) isolates and quantifies the structural series resistance—the requisite fractional potential drops—occurring across redundant parallel pathways outside the primary bottleneck. (See Appendix B.1 for the complete algebraic derivation). □

**Corollary 4.2** (Bottleneck Typology). *For a given $L_1$ local bottleneck capacity $Z_{s,t}^*$, the magnitude of the redundancy gap $\Delta_{s,t} = Z_{s,t}^* - C_{s,t}$ uniquely classifies the structural topology of the vulnerability into two functional extremes:*

1. ***Strict Topological Bridge** ($\Delta_{s,t} \to 0$): The effective conductance approaches the minimum cut capacity ($C_{s,t} \approx Z_{s,t}^*$). The network resistance is strictly concentrated at a singular structural interface.*

2. ***Diffuse Bottleneck** ($\Delta_{s,t} \to Z_{s,t}^*$): The effective conductance is fractionally smaller than the minimum cut ($C_{s,t} \ll Z_{s,t}^*$). The network resistance is distributed across a prolonged, fragmented sequence of edges outside the primary cut.*

*Proof Sketch.* This classification follows as a direct algebraic consequence of the gap quantification derived in Lemma 4.1. We established that $\Delta_{s,t} = \sum_{(u,v) \in C^*} w_{uv}\big(1 - (\phi_u^* - \phi_v^*)\big)$, where $\phi^*$ is the $L_2$ optimal potential and $C^*$ is the minimum cut set.

Because the total potential difference is bounded ($\phi_t^* - \phi_s^* = 1$), the gap $\Delta_{s,t}$ approaches 0 if and only if ($\phi_u^* - \phi_v^*) \to 1$ for the edges in $C^*$. This condition requires that the aggregate resistance of all network components outside of $C^*$ approaches zero (infinite conductance). Topologically, this defines a strict bridge: two dense, highly connected subgraphs joined only by the cut edges.

Conversely, $\Delta_{s,t}$ approaches its maximum possible value of $Z_{s,t}^*$ as $C_{s,t} \to 0$. This occurs when the potential difference across the cut edges $(\phi_u^* - \phi_v^*) \to 0$, meaning the unit potential drop is heavily consumed by series resistance in the rest of the network. Topologically, this defines a diffuse bottleneck: a long, poorly connected sequence of edges lacking redundant parallel paths, where the absolute weakest link $(Z_{s,t}^*)$ is only marginally weaker than the surrounding corridor. (See Appendix B.2 for the limit behaviour analysis). $\qquad\square$

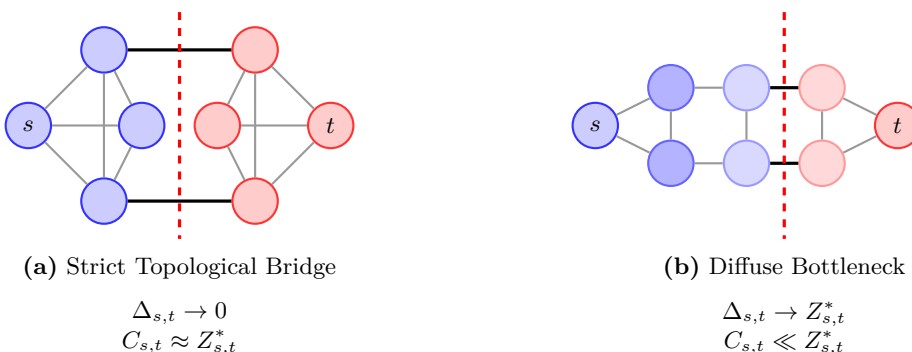

**(a)** Strict Topological Bridge
$$\Delta_{s,t} \to 0$$
$$C_{s,t} \approx Z_{s,t}^*$$

**(b)** Diffuse Bottleneck
$$\Delta_{s,t} \to Z_{s,t}^*$$
$$C_{s,t} \ll Z_{s,t}^*$$

Figure 3: Visualisation of the structural redundancy gap $\Delta_{s,t}$ formalised in Lemma 4.1 and Corollary 4.2. The red dashed line denotes the active $L_1$ separating manifold identifying the Nodal Tension capacity $Z_{s,t}^*$. **(a)** In a strict topological bridge $(\Delta_{s,t} \to 0)$, the network possesses dense redundant parallel paths outside the minimal surface. The $L_2$ effective conductance approaches the $L_1$ capacity $(C_{s,t} \approx Z_{s,t}^*)$, forcing the unit potential difference to drop strictly across the cut edges. **(b)** In a diffuse bottleneck $(\Delta_{s,t} \to Z_{s,t}^*)$, network resistance is distributed across a prolonged, un-bypassed sequence of edges. The continuous $L_2$ electrical potential $(\phi^*)$ drops incrementally across the corridor (indicated by the structural node colour gradient). Here, $C_{s,t} \ll Z_{s,t}^*$, confirming that the global flow is choked by series resistance, leaving the capacity of the $L_1$ weakest link severely under-utilised (Corollary 4.3).

**Environmental Interpretation:** Operationalising this $L_1$-$L_2$ gap provides a diagnostic tool for both landscape ecology and graph representation learning. In ecological network analysis, identifying a strict topological bridge $(\Delta_{s,t} \to 0$, as illustrated in Figure 3a) implies that highly localised interventions—such as constructing a single wildlife crossing over a barrier—will immediately and maximally restore global connectivity, as the surrounding habitat is already dense and robust. However, identifying a diffuse bottleneck $(\Delta_{s,t} \to Z_{s,t}^*$, Figure 3b) proves that localised intervention at the weakest link will fail; augmenting the absolute capacity $Z_{s,t}^*$ alone will not meaningfully increase overall flow $(C_{s,t})$ due to the overriding series resistance of the surrounding degraded corridor, necessitating broad-scale habitat restoration.

**Algorithmic Interpretation for Graph Representation Learning:** Beyond environmental applications, quantifying this structural redundancy gap resolves a critical diagnostic ambiguity in Graph Neural Networks (GNNs). Standard Message Passing Neural Networks (MPNNs) compute node representations by aggregating features across local neighbourhoods, a process restricted by the underlying graph topology. It is a well-documented failure mode that MPNNs suffer from "over-squashing" when forced to propagate information from an exponentially growing receptive field through a narrow structural bottleneck (Alon & Yahav, 2021; Topping et al., 2022).

Previously, the literature has broadly treated all graph bottlenecks as identical pathologies. However, our topological classification isolates the geometric nature of the vulnerability, dictating how an MPNN will fail. A strict topological bridge $(\Delta_{s,t} \to 0)$ forces large information compression across a singular structural interface. This induces acute, highly localised over-squashing in a single message-passing step, as the intermediate node vectors cannot losslessly compress the neighbourhood features. Conversely, a diffuse bottleneck $(\Delta_{s,t} \to Z_{s,t}^*)$ forces messages to propagate along a prolonged, low-redundancy corridor. Traversing this resistance requires deep network architectures with numerous message-passing hops, inevitably triggering vanishing gradients and compound spatial over-smoothing over depth.

In practical terms, identifying whether a bottleneck is strict or diffuse is a fundamental prerequisite for modern graph rewiring techniques (Arnaiz-Rodríguez et al., 2022). By jointly tracking the $L_1$ capacity $Z_{s,t}^*$ and the $L_2$ conductance $C_{s,t}$, algorithmic frameworks can explicitly condition their topological interventions: a strict bridge necessitates surgically adding a single cross-cut edge to bypass the pinch point, whereas a diffuse resistance corridor necessitates introducing multi-hop spatial shortcuts to reduce the effective graph diameter.

**Corollary 4.3** (Flow Saturation and Cut Under-utilisation)**.** *For any $L_1$ minimum $s - t$ cut set $C^*$, the redundancy gap $\Delta_{s,t}$ is equivalent to the aggregate unutilised structural capacity of the cut edges under $L_2$ optimal flow. Consequently, in a diffuse bottleneck ($\Delta_{s,t} \to Z_{s,t}^*$), the absolute weakest link $C^*$ is severely flow-starved; its maximum capacity is unutilised due to the overriding series resistance of the surrounding network.*

*Proof Sketch.* Let $f_{uv}^* = w_{uv}(\phi_u^* - \phi_v^*)$ denote the exact $L_2$ electrical flow across a directed edge $(u,v) \in C^*$ induced by the unit boundary potential, as governed by Ohm's Law. Building directly upon the algebraic formulation from Lemma 4.1, we can factor the gap equation:

$$\Delta_{s,t} = \sum_{(u,v)\in C^*} w_{uv} - \sum_{(u,v)\in C^*} w_{uv}(\phi_u^* - \phi_v^*) = \sum_{(u,v)\in C^*} (w_{uv} - f_{uv}^*) \tag{10}$$

Since $Z_{s,t}^* = \sum_{C^*} w_{uv}$ represents the maximum capacity of the bottleneck, and $C_{s,t} = \sum_{C^*} f_{uv}^*$ represents the actual realised flow crossing it, the residual term $(w_{uv} - f_{uv}^*) \geq 0$ isolates the unused bandwidth of each edge. As $\Delta_{s,t} \to Z_{s,t}^*$, it is guaranteed that $f_{uv}^* \to 0$ for all cut edges. Thus, the global flow is choked by the degraded exterior network before it can even saturate the primary $L_1$ bottleneck. (See Appendix B.3 for capacity bounds). $\square$

This corollary formalises a known limitation in network intervention: augmenting the topological minimum cut does not necessarily increase global flow if the cut is not the primary restrictor. In applied ecology, when a bottleneck is diffuse ($\Delta_{s,t} \to Z_{s,t}^*$), localised capacity addition at $C^*$ yields diminishing returns for overall connectivity, as the effective resistance is dominated by the extended surrounding landscape (McRae & Beier, 2007). In the context of Graph Neural Networks, this under-utilisation metric provides a structural explanation for why local edge addition (graph rewiring) often fails to alleviate over-squashing in highly resistive neighbourhoods. As information propagates through a diffuse bottleneck, representation vectors are attenuated by the extended sequence of low-capacity edges prior to reaching the actual $L_1$ partition (Topping et al., 2022). Consequently, resolving such bottlenecks in Message Passing Neural Networks generally requires multi-hop or global attention mechanisms to bypass the extended resistive corridor entirely, rather than localised topological adjustments (Alon & Yahav, 2021).

## 4.2 Bottleneck Geometry and Discrete Ricci Curvature

While Section 4.1 established how the macroscopic redundancy gap limits global $s - t$ flow, it is equally critical to understand the precise local structural shape of the $L_1$ bottleneck itself. In modern Graph Representation Learning, local structural bottlenecks are conventionally diagnosed using discrete differential geometry, specifically Forman-Ricci curvature (Topping et al., 2022). Forman-Ricci curvature evaluates an edge based on its local neighbourhood dispersion: edges connecting dense, highly triangulated clusters possess positive curvature, whereas isolated topological bridges connecting disjoint neighbourhoods exhibit negative curvature.

However, discrete curvature is inherently a myopic, local metric. A large network may contain hundreds of negatively curved edges acting as local bridges. Geometric metrics alone cannot determine which of these isolated local structures actually constrain the global, mesoscale connectivity between a specific source $s$ and target $t$.

In this subsection, we establish a link proving that the continuous $L_1$ Nodal Tension variables $p^*$ natively interface with this discrete geometry. We demonstrate that the continuous LP localises the unit tension drop exclusively onto the most negatively curved separating manifold. Consequently, Nodal Tension acts

as an exact global filter for local differential geometry, resolving the diagnostic ambiguity of spurious local bottlenecks.

To formalise this geometric filter, we first explicitly define the discrete curvature metric. Following the standard combinatorial formulation for unweighted graphs (Topping et al., 2022), the Forman-Ricci curvature of an edge $e = (u, v)$ is given by:

$$\mathbf{F}(u, v) = 4 - d_u - d_v + 3|\triangle_{uv}| \tag{11}$$

where $d_u$ and $d_v$ are the respective degrees of the incident nodes, and $|\triangle_{uv}|$ denotes the number of shared triangles (common neighbours) that include the edge $(u, v)$.

This algebraic definition governs the topological dispersion of the local neighbourhood. If an edge connects two high-degree hubs but lacks any shared triangles ($|\triangle_{uv}| = 0$), the curvature $\mathbf{F}(u, v)$ becomes heavily negative, identifying the edge as a strict, isolated topological bridge. Conversely, if the edge is embedded within a dense, highly clustered neighbourhood, the proliferation of shared 2-hop parallel paths ($|\triangle_{uv}| \gg 0$) drives the curvature positive.

By evaluating our continuous Nodal Tension LP variables $p^*$ against the explicit structural components of Equation 11, we can prove how the $L_1$ global optimal solution inherently binds to this local differential geometry.

**Lemma 4.4** (Nodal Tension Bounds Local Curvature)**.** *Let $p^*$ be the optimal continuous solution to the Nodal Tension LP, and let $Z^*_{s,t}$ denote its optimal objective value (the $L_1$ bottleneck capacity). For an unweighted graph, if an edge $e = (u, v)$ sustains maximal tension such that $|p^*_u - p^*_v| = 1$, the geometric severance of the partition limits its shared triangles to $|\triangle_{uv}| \leq Z^*_{s,t} - 1$. Consequently, its discrete Forman-Ricci curvature is explicitly bounded from above by the global LP capacity and the local degrees:*

$$\mathbf{F}(u, v) \leq 4 - d_u - d_v + 3(Z^*_{s,t} - 1) \tag{12}$$

*Proof Sketch.* By Theorem 3.2, the condition $|p^*_u - p^*_v| = 1$ guarantees that the edge $(u, v)$ belongs to the strict $L_1$ minimum cut set $C^*$, which partitions the vertex set into a source component $V_s$ and a target component $V_t$. Without loss of generality, assume $u \in V_s$ and $v \in V_t$.

For any shared triangle to exist on $(u, v)$, there must be a third common neighbour $w \in V$ such that the edges $(u, w)$ and $(w, v)$ both exist in the graph. Because $V_s \cup V_t = V$ forms a strict binary partition, the node $w$ must reside in either $V_s$ or $V_t$.

If $w \in V_s$, the edge $(w, v)$ spans from $V_s$ to $V_t$ and must therefore belong to the cut set $C^*$. Conversely, if $w \in V_t$, the edge $(u, w)$ spans the partition and must belong to $C^*$. This geometric constraint dictates that every distinct shared triangle incident to $(u, v)$ injectively maps to at least one additional, unique edge within the global minimum cut (illustrated in Figure 4a).

In an unweighted graph, the total number of edges in the cut is the optimal LP capacity, $|C^*| = Z^*_{s,t}$. Because the edge $(u, v)$ itself accounts for 1 unit of this capacity, the injective mapping bounds the local triangle count $|\triangle_{uv}|$ by the cardinality of the residual cut, establishing $|\triangle_{uv}| \leq Z^*_{s,t} - 1$. Substituting this triangle limit directly into Equation 11 yields the stated curvature bound. In the limit case of a strict topological bridge ($Z^*_{s,t} = 1$), it is algebraically guaranteed that $|\triangle_{uv}| = 0$, driving the differential geometry negative ($\mathbf{F}(u, v) = 4 - d_u - d_v$) in direct proportion to the local neighbourhood dispersion. (See Appendix B.4 for the injective mapping derivation). $\square$

**Lemma 4.5** (Positive Curvature Resists Tension Gradients)**.** *Let $p^*$ be the optimal continuous solution to the Nodal Tension LP, and $Z^*_{s,t}$ the global bottleneck capacity. The optimal tension gradient across any edge $e = (u, v)$ is restricted by its local positive curvature. Specifically, assigning maximal tension across an edge requires simultaneously exhausting the capacity of all its shared triangles, establishing the strict algebraic bound:*

$$|p^*_u - p^*_v| \leq \frac{Z^*_{s,t}}{1 + |\triangle_{uv}|} \tag{13}$$

*Substituting the components of Forman-Ricci curvature (Equation 11), the continuous tension gradient is explicitly inversely bounded by the positive local geometry:*

$$|p_u^* - p_v^*| \leq \frac{3Z_{s,t}^*}{\mathbf{F}(u,v) + d_u + d_v - 1} \tag{14}$$

*Proof Sketch.* By the integrality property of the Nodal Tension LP (Theorem 3.2), the potential difference $|p_u^* - p_v^*|$ is binary, taking the value 1 if $(u,v) \in C^*$ and 0 otherwise.

Assume the LP assigns a maximal tension gradient $|p_u^* - p_v^*| = 1$. As established by the injective mapping in Lemma 4.4, the edge $(u,v)$ spans the $s - t$ partition. Consequently, for every shared triangle formed by a common neighbour $w$, at least one of the adjacent edges $(u,w)$ or $(w,v)$ must also uniquely span the partition.

To maintain a valid topological partition, the minimum cut $C^*$ must therefore contain the primary edge $(u,v)$ plus at least one additional, distinct edge for every shared triangle. This imposes a strict geometric minimum on the global bottleneck capacity required to cut the edge: if $|p_u^* - p_v^*| = 1$, the capacity must satisfy $Z_{s,t}^* \geq 1 + |\triangle_{uv}|$.

Because the optimal tension gradient is binary ($|p_u^* - p_v^*| \in \{0,1\}$), the condition $1 \leq \frac{Z_{s,t}^*}{1+|\triangle_{uv}|}$ (when tension is 1) and the trivial $0 \leq \frac{Z_{s,t}^*}{1+|\triangle_{uv}|}$ (when tension is 0) combine seamlessly into the generalised continuous inequality: $|p_u^* - p_v^*| \leq \frac{Z_{s,t}^*}{1+|\triangle_{uv}|}$.

Algebraically isolating $|\triangle_{uv}|$ from the definition of Forman-Ricci curvature yields $|\triangle_{uv}| = \frac{\mathbf{F}(u,v)-4+d_u+d_v}{3}$. Substituting this into the denominator of our continuous bound yields the final inverse relationship between the optimal tension gradient and the local differential geometry.

Thus, if an edge is embedded in a locally dense, positively curved manifold such that its supporting triangles exceed the available global capacity ($1+|\triangle_{uv}| > Z_{s,t}^*$), the LP is forced to assign it zero tension ($|p_u^* - p_v^*| = 0$). This proves that Nodal Tension inherently routes around locally robust substructures. (See Appendix B.5 for the constraint evaluation). $\qquad\square$

**Theorem 4.6** (Geometric Localisation Theorem). *Let $p^*$ be the optimal continuous solution to the Nodal Tension LP, yielding the minimal $s - t$ separating manifold $C^*$ with capacity $Z_{s,t}^*$. The global minimisation of the $L_1$ objective localises the $s - t$ partition onto the separating manifold exhibiting minimal discrete curvature. Consequently, the aggregate discrete Forman-Ricci curvature of the optimal bottleneck is explicitly bounded from above by the local neighbourhood dispersions and the square of the global capacity:*

$$\sum_{(u,v)\in C^*} \mathbf{F}(u,v) \leq \sum_{(u,v)\in C^*} (4 - d_u - d_v) + 3Z_{s,t}^*(Z_{s,t}^* - 1) \tag{15}$$

*Proof Sketch.* The continuous Nodal Tension LP objective natively minimises the sum of absolute potential differences across the network ($\min \sum |p_u - p_v|$). As proven in Lemma 4.5, the optimal tension gradient is repelled by positively curved local neighbourhoods; assigning a unit potential drop across a highly triangulated edge requires a monotonic increase in the required capacity $Z_{s,t}^*$. Thus, to minimise the global objective, the continuous optimisation inherently routes the $s - t$ partition through the subgraph exhibiting the lowest topological cohesion (minimal shared triangles).

When the optimal manifold $C^*$ is isolated, Lemma 4.4 dictates that the geometric severance of the partition bounds the shared triangles of every individual cut edge $(u,v) \in C^*$ to $|\triangle_{uv}| \leq Z_{s,t}^* - 1$.

To evaluate the differential geometry of the entire bottleneck, we aggregate the discrete Forman-Ricci curvature over the strict $L_1$ partition. By summing the individual edge curvature bounds from Lemma 4.4 over all $|C^*| = Z_{s,t}^*$ edges comprising the minimal manifold, we obtain:

$$\sum_{(u,v)\in C^*} \mathbf{F}(u,v) \leq \sum_{(u,v)\in C^*} \left[ 4 - d_u - d_v + 3(Z_{s,t}^* - 1) \right] \tag{16}$$

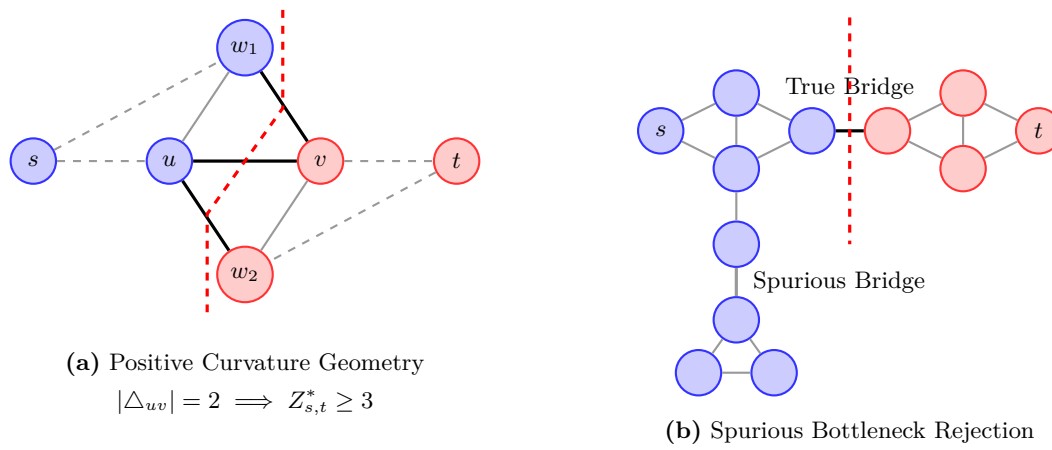

**(a)** Positive Curvature Geometry

$$|\triangle_{uv}| = 2 \implies Z^*_{s,t} \geq 3$$

**(b)** Spurious Bottleneck Rejection

$$\mathbf{F} < 0, \text{ but } |p^*_x - p^*_y| = 0$$

Figure 4: Visualisation of the microscopic geometric filtering performed by the continuous Nodal Tension variables $(p^*)$. **(a)** Illustration of the injective mapping (Lemma 4.4). An edge $(u, v)$ embedded in a dense manifold possesses positive discrete curvature due to its shared triangles $(\triangle_{uv})$. Because the node potentials $(p^* \in \{0, 1\})$ enforce a strict bipartite partition between the macroscopic source $s$ and target $t$, separating $u$ and $v$ geometrically forces the active minimal surface (dashed red line) to uniquely sever at least one additional edge for every shared triangle. This explains why assigning maximal tension across a positively curved structure inherently requires a large global capacity $(Z^*_{s,t})$. **(b)** Illustration of spurious bottleneck rejection (Corollary 4.7). The network contains two identical topological bridges. Both lack shared triangles, yielding negative Forman-Ricci curvature ($\mathbf{F} < 0$). However, the spurious bridge connects only to a local sub-cluster. Because it does not separate the macroscopic $s - t$ pathway, both of its endpoints natively receive the exact same optimal continuous potential ($p^* = 0$, shown in blue). The tension gradient filters out this local geometric fragility, collapsing to 0 and showing that the LP acts as an exact global-to-local filter.

By the linearity of summation, the global capacity term $3(Z^*_{s,t} - 1)$ is algebraically separated. Because this constant is summed $Z^*_{s,t}$ times across the manifold, it evaluates to the quadratic term $3Z^*_{s,t}(Z^*_{s,t} - 1)$.

This explicit upper bound proves that solving the continuous Nodal Tension LP is geometrically equivalent to identifying the discrete minimal surface—the deepest structural valley in the local curvature landscape—that separates $s$ and $t$. (See Appendix B.6 for the manifold aggregation). □

**Corollary 4.7** (Spurious Bottleneck Rejection). *Let $C^*$ be the optimal $L_1$ separating manifold defining the true $s - t$ topological bottleneck identified by Theorem 4.6. For any locally negatively curved edge $e = (x, y) \in E \setminus C^*$ (where $\mathbf{F}(x, y) < 0$) that is topologically disjoint from this global minimal surface, the optimal continuous tension gradient is forced to zero:*

$$|p^*_x - p^*_y| = 0 \tag{17}$$

*Proof Sketch.* By the integrality of the Nodal Tension LP (Theorem 3.2), the optimal continuous potentials partition the vertex set into a source component ($p^* = 0$) and a target component ($p^* = 1$). The optimal separating manifold $C^*$ consists exclusively of the edges that span across these two components. If an arbitrary edge $(x, y)$ does not belong to $C^*$, it does not cross this partition. Consequently, both endpoints $x$ and $y$ must reside within the same vertex component, meaning they share the exact same assigned potential (either $p^*_x = p^*_y = 0$ or $p^*_x = p^*_y = 1$). Therefore, regardless of the edge's local negative curvature $\mathbf{F}(x, y) < 0$, its optimal tension gradient evaluates to zero: $|p^*_x - p^*_y| = 0$. (See Appendix B.7 for the partition evaluation). □

This corollary provides a practical mechanism to filter local geometric metrics. As visually depicted in Figure 4b, a large network often contains many isolated edges exhibiting negative Forman-Ricci curvature.

However, not all of these local bridges structurally constrain the macroscopic flow between a specific source $s$ and target $t$. By solving the continuous Nodal Tension LP, we can identify which of these negatively curved edges actively participate in the global $s - t$ bottleneck. For Graph Neural Networks, this dictates that spatial rewiring algorithms do not need to exhaustively augment every negatively curved structure in the graph to alleviate over-squashing along specific message-passing paths. Instead, topological interventions can be surgically focused only on edges where the continuous tension gradient is positive ($|p_x^* - p_y^*| > 0$). Similarly, in applied spatial ecology, this diagnostic filtering confirms whether a locally degraded habitat actually severs a specific wildlife corridor, preventing the misallocation of restoration resources to spurious topological dead-ends.

### 4.3 Local Cheeger Inequality and Commute Times

In Section 4.2, we established that the continuous Nodal Tension LP localises the $s - t$ bottleneck onto a negatively curved minimal surface. This surface separates two topologically cohesive, positively curved components: the source set $V_s$ and the target set $V_t$. While this defines the static geometry of the minimal cut, we now evaluate its dynamical consequences. Specifically, we analyse how this localised structural restriction governs the expected time required for information, modelled as a random walk, to propagate across the $s - t$ pathway.

Standard spectral graph theory bounds the mixing time of a random walk using the global Cheeger inequality, derived from the graph's algebraic connectivity (the Fiedler value) (Fiedler, 1973; Chung, 1997; Levin & Peres, 2017). However, these spectral metrics are global. They evaluate the conductance of the entire network as a single entity, which frequently fails to capture the localised restriction of a specific $s - t$ corridor. If the broader graph is highly connected, the global Cheeger constant remains large, masking the presence of a severe local bottleneck separating $s$ and $t$.

To address this limitation, we construct a localised Cheeger framework. By leveraging the exact binary subsets $V_s$ and $V_t$ generated by the optimal continuous variables $p^*$, we define a local isoperimetric ratio based on the $L_1$ bottleneck capacity $Z_{s,t}^*$. This allows us to bound the random walk commute time across the isolated $s - t$ cross-section, providing a dynamical measure of local signal decay without requiring a global eigendecomposition.

Before formalising the dynamical properties of the Nodal Tension partition, we establish the standard global bound for network bottlenecks as a baseline.

**Theorem 4.8** (Global Cheeger Inequality (Fiedler, 1973; Chung, 1997))**.** *Let $G = (V, E)$ be a graph with local vertex degrees $d_v$ for $v \in V$. The volume of any subset $S \subset V$ is defined as the sum of its internal degrees, $vol(S) = \sum_{v \in S} d_v$. The global Cheeger constant (or graph conductance), denoted $h_G$, identifies the absolute minimum isoperimetric ratio over all possible non-empty subsets $S \subset V$ satisfying $vol(S) \leq \frac{1}{2} vol(V)$:*

$$h_G = \min_S \frac{|E(S, V \setminus S)|}{vol(S)} \tag{18}$$

*where $E(S, V \setminus S)$ represents the set of edges spanning $S$ and its complement. The global Cheeger constant is bounded by the second smallest eigenvalue of the normalised graph Laplacian, denoted $\lambda_2$ (the algebraic connectivity or Fiedler value):*

$$\frac{\lambda_2}{2} \leq h_G \leq \sqrt{2\lambda_2} \tag{19}$$

**The Limitation for Local Resilience:** The Cheeger constant $h_G$ evaluates the absolute structural minimum of the entire network. Crucially, the optimisation $\min_S$ is agnostic to any specific source $s$ or target $t$. If a graph $G$ contains a highly connected core but sustains a severe structural bottleneck restricted to the $s - t$ pathway, the global eigenvalue $\lambda_2$ may remain large (dominated by the robust core). Conversely, the global minimisation for $h_G$ may isolate a disconnected leaf node that is irrelevant to the flow between $s$ and $t$. Consequently, global spectral bounds cannot provide a tight, guaranteed limit on the commute time or signal decay across the minimal surface $C^*$ identified by the Nodal Tension LP.

**Lemma 4.9** (Local Nodal Conductance)**.** *Let $p^* \in \{0, 1\}^{|V|}$ be the optimal integer solution to the Nodal Tension LP, partitioning the vertex set into a source component $V_s = \{v \in V \mid p_v^* = 0\}$ and a target*

component $V_t = \{v \in V \mid p_v^* = 1\}$. Let the optimal $s - t$ capacity be $Z_{s,t}^* = |C^*|$. We define the local Nodal Conductance, $\Phi_{s,t}^*$, to evaluate the isoperimetric restriction of the specific $s - t$ pathway:

$$\Phi_{s,t}^* = \frac{Z_{s,t}^*}{\min(vol(V_s), vol(V_t))} \tag{20}$$

Because the vertex partitions $V_s$ and $V_t$ constitute a specific, valid subset of the graph where $\min(vol(V_s), vol(V_t)) \leq \frac{1}{2} vol(V)$, the local Nodal Conductance is bounded from below by the absolute global Cheeger constant $h_G$:

$$h_G \leq \Phi_{s,t}^* \tag{21}$$

*Proof Sketch.* By definition, the global Cheeger constant $h_G$ identifies the absolute minimum isoperimetric ratio over all possible non-empty subsets $S \subset V$ satisfying a specific volume constraint (Theorem 4.8). The continuous Nodal Tension LP converges on one distinct integer partition of the graph, defined by the mutually exclusive and collectively exhaustive subsets $V_s$ and $V_t$.

Because $V_s \cup V_t = V$, the sum of their volumes equals the total volume of the graph. Therefore, the subset with the smaller volume, denoted $S^*$, cannot exceed half the total volume: $\mathrm{vol}(S^*) = \min(\mathrm{vol}(V_s), \mathrm{vol}(V_t)) \leq \frac{1}{2}\mathrm{vol}(V)$.

This establishes $S^*$ as a valid candidate subset within the domain of the global conductance minimisation. Evaluating the isoperimetric ratio specifically for $S^*$ yields the local Nodal Conductance, $\Phi_{s,t}^*$, because the edges spanning $S^*$ and its complement constitute the optimal $s - t$ capacity $Z_{s,t}^*$. Because $h_G$ is defined as the absolute minimum over all valid subsets, the ratio evaluated at this specific subset $S^*$ must logically be greater than or equal to the global minimum, establishing the bound $h_G \leq \Phi_{s,t}^*$.

This inequality demonstrates that global spectral bounds serve as a lower limit and do not necessarily capture the precise severity of a localised $s - t$ bottleneck (as demonstrated in Figure 5). When the LP isolates a negatively curved minimal surface separating dense components (Theorem 4.6), the local conductance $\Phi_{s,t}^*$ accurately quantifies the specific geometric restriction governing $s - t$ signal propagation, bypassing the global minimum. (See Appendix B.8 for the subset evaluation). □

**Theorem 4.10** (Local Bound on Commute Times). *Let $G = (V, E)$ be an unweighted graph where a discrete-time random walk $(X_k)_{k \geq 0}$ transitions from node $u$ to an adjacent node $v$ with uniform probability $P_{uv} = \frac{1}{d_u}$. The hitting time $h(s, t)$ is explicitly defined as the expected number of steps required for the walk to first reach the target node $t$ given a starting position at the source node $s$:*

$$h(s,t) = \mathbb{E}\left[\min\{k \geq 0 \mid X_k = t\} \mid X_0 = s\right] \tag{22}$$

*The expected commute time $H(s, t)$ is the strict sum of the directional hitting times, representing the total expected duration of the round trip:*

$$H(s,t) = h(s,t) + h(t,s) \tag{23}$$

*Let $p^*$ be the optimal integer solution to the Nodal Tension LP, yielding the minimal $s - t$ separating surface $C^*$ with capacity $Z_{s,t}^*$. The expected commute time is bounded from below by the total graph volume and the inverse of this optimal capacity:*

$$H(s,t) \geq \frac{vol(V)}{Z_{s,t}^*} \tag{24}$$

*By substituting the local Nodal Conductance $\Phi_{s,t}^*$, the expected commute time is bounded by the local isoperimetric ratio and the volume of the smaller partitioned component:*

$$H(s,t) \geq \frac{vol(V)}{\Phi_{s,t}^* \min(vol(V_s), vol(V_t))} \tag{25}$$

*Proof Sketch.* By the fundamental identity (Chandra et al., 1989), the expected round-trip time $H(s, t)$ of a standard random walk is proportional to the effective electrical resistance $R_{s,t}$ between the two nodes, governed by the equality $H(s,t) = \mathrm{vol}(V)R_{s,t}$.

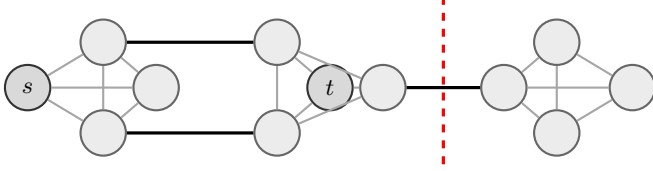

**(a)** Global Cheeger Minimisation ($h_G$)

Absolute Sparsest Cut
(Irrelevant to $s - t$ flow)

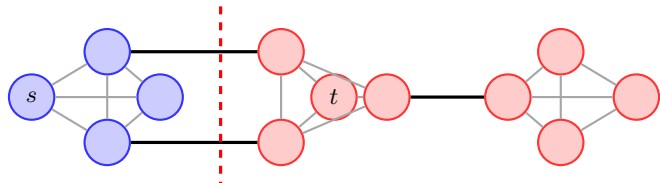

**(b)** Local Nodal Conductance ($\Phi_{s,t}^*$)

Local Minimal Surface
(Governs $(P^K)_{ts}$ Commute Time)

Figure 5: Illustration of the diagnostic limitation of the global Cheeger inequality (Theorem 4.8) and the necessity of local Nodal Conductance (Lemma 4.9). The "tri-barbell" network contains two topological bridges: Bridge A (capacity 2, between $s$ and $t$) and Bridge B (capacity 1, connecting a dense sub-cluster). **(a)** The global Cheeger constant ($h_G$) evaluates the absolute minimum isoperimetric ratio of the entire graph. It isolates Bridge B. However, this global spectral cut leaves $s$ and $t$ within the identical un-partitioned component, showing that global eigenvalues ($\lambda_2$) provide loose, uninformative bounds for specific pathway restrictions. **(b)** The continuous Nodal Tension LP actively filters out the global topological minimum. The optimal integer potentials ($p^* \in \{0, 1\}$, shown in blue and red) uniquely partition the graph at Bridge A, defining the local Nodal Conductance ($\Phi_{s,t}^* \geq h_G$). By isolating this $s - t$ cross-section, the metric establishes the strict geometric threshold that restricts random walk commute times and forces localised Message Passing over-squashing (Corollary 4.11).

To establish a strict lower bound on $R_{s,t}$, we evaluate its exact variational definition. The effective conductance, defined as $1/R_{s,t}$, is equivalent to the minimum Dirichlet energy evaluated over all valid potential functions satisfying the $s - t$ boundary conditions. Because this variational form seeks an absolute minimum, evaluating the Dirichlet energy at any single valid test function establishes a strict upper bound on the effective conductance.

We construct a valid test function directly from the optimal binary variables of the Nodal Tension LP ($p^*$). Because these variables are restricted to $\{0, 1\}$, the squared potential difference across any edge equals its absolute difference. Consequently, the Dirichlet energy evaluated at this specific test function equals the sum of the absolute potential drops, which is the optimal LP capacity $Z_{s,t}^*$.

This exact substitution guarantees the strict inequality $1/R_{s,t} \leq Z_{s,t}^*$, which algebraically inverts to $R_{s,t} \geq 1/Z_{s,t}^*$. Multiplying both sides by the total graph volume produces the first explicit bound: $H(s,t) \geq \frac{\text{vol}(V)}{Z_{s,t}^*}$.

Finally, we algebraically isolate the capacity from the definition of local Nodal Conductance established in Lemma 4.9: $Z_{s,t}^* = \Phi_{s,t}^* \min(\text{vol}(V_s), \text{vol}(V_t))$. Substituting this exact expression into the denominator yields the final inverse relationship between the expected commute time and the local isoperimetric ratio. (See Appendix B.9 for the variational derivation). □

**Corollary 4.11** (Localised Over-Squashing in Message Passing)**.** *Let $G = (V, E)$ be processed by a linear Message Passing Neural Network (MPNN) utilising normalised aggregation $X^{(k)} = D^{-1}AX^{(k-1)}$ for $K$ discrete layers. The Jacobian sensitivity of a target node $t$'s final representation to a source node $s$'s initial*

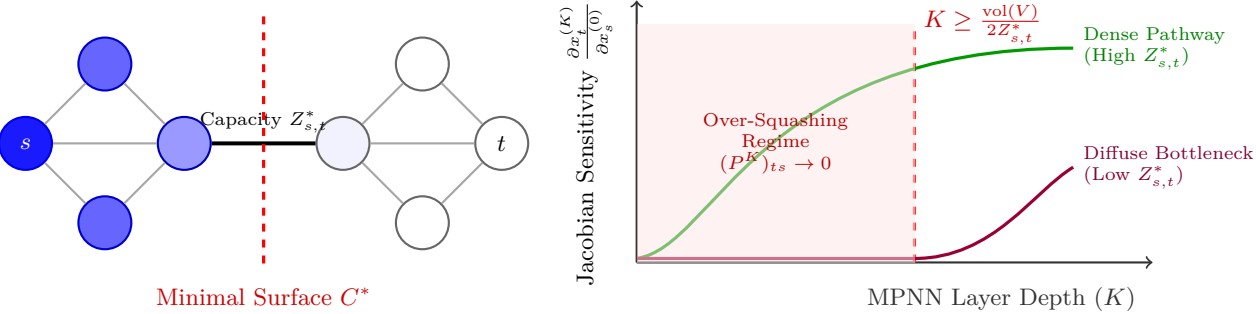

**(a)** Topological Receptive Field Choke
Signal mass $(P^K)_{ts}$ pools at the bottleneck

**(b)** The Jacobian Sensitivity Threshold
Gradients vanish until expected hitting time is met

Figure 6: Visualisation of localised over-squashing in Message Passing Neural Networks driven by local Nodal Conductance (Corollary 4.11). **(a)** The topological mechanism of the receptive field choke. As the MPNN layer depth $K$ increases, the probability mass of the transition matrix $(P^K)$ expands outward from the source $s$. Node fill colours represent relative signal intensity. Because the minimal surface $C^*$ possesses a tiny capacity $Z_{s,t}^*$, the message mass is structurally blocked, pooling on the source side and leaving the target $t$ flow-starved. **(b)** The analytical Jacobian sensitivity threshold. For a dense pathway, sensitivity rises rapidly as $K$ increases (green curve). However, for a diffuse local bottleneck, the expected hitting time imposes a strict lower bound on traversal. If the layer depth $K$ is less than the critical threshold $\frac{\text{vol}(V)}{2Z_{s,t}^*}$ (the red shaded over-squashing regime), the $s-t$ signal is geometrically terminated. The Jacobian sensitivity curve (purple) remains functionally flat at zero, proving that standard MPNNs fail to capture long-range interactions across structurally restricted minimal surfaces regardless of global graph connectivity.

*feature, denoted $\frac{\partial x_t^{(K)}}{\partial x_s^{(0)}}$, is proportional to the $K$-step random walk transition probability $(P^K)_{ts}$. By Theorem 4.10, the expected commute time dictates that the maximum of the directional hitting times is bounded from below by the local bottleneck capacity $Z_{s,t}^*$:*

$$\max(h(s,t), h(t,s)) \geq \frac{vol(V)}{2Z_{s,t}^*} = \frac{vol(V)}{2\Phi_{s,t}^* \min(vol(V_s), vol(V_t))} \tag{26}$$

*Therefore, if the network depth $K$ is instantiated such that $K < \frac{vol(V)}{2Z_{s,t}^*}$, the layer depth is less than the expected number of steps required for the message passing operator to traverse the minimal separating surface $C^*$ in the maximal direction. Because the available steps cannot satisfy the expected hitting time, the corresponding transition probability mass $(P^K)_{ts}$ is structurally constrained. This directly forces the Jacobian sensitivity $\frac{\partial x_t^{(K)}}{\partial x_s^{(0)}}$ toward zero, establishing that a shrinking local Nodal Conductance ($\Phi_{s,t}^* \to 0$) induces localised over-squashing independently of the global graph geometry.*

*Proof Sketch.* In a standard linear Message Passing Neural Network, the layer-wise feature aggregation is defined by the matrix system $X^{(K)} = P^K X^{(0)}$, where the operator $P = D^{-1}A$ is identical to the transition matrix of a discrete random walk. Consequently, the exact Jacobian sensitivity of a target node's final state to a source node's initial state, $\frac{\partial x_t^{(K)}}{\partial x_s^{(0)}}$, evaluates to the scalar entry $(P^K)_{ts}$.

For this entry to accumulate significant probability mass (visualised in Figure 6a), the architectural depth $K$ must reach the expected hitting time required for a structural signal to cross the network. By applying the algebraic property of averages to the commute time bound established in Theorem 4.10, the maximum expected directional hitting time is bounded below by $\frac{\text{vol}(V)}{2Z_{s,t}^*}$. Substituting the local Nodal Conductance $\Phi_{s,t}^*$ isolates the geometric bottleneck threshold. If a fixed hyperparameter depth $K$ satisfies $K < \frac{\text{vol}(V)}{2Z_{s,t}^*}$, the network terminates its forward pass before the expected traversal time of the minimal surface $C^*$. This

structural deficit forces the Jacobian sensitivity to converge toward zero, algebraically proving the mechanism of localised over-squashing (Figure 6b). (See Appendix B.10 for the linear system derivation). □

Because Corollary 4.11 establishes that local geometric capacities restrict the Jacobian sensitivity in MPNNs, it becomes necessary to determine how structurally sensitive these specific bottlenecks are to topological modifications or noise. In Section 4.4, we transition from static geometric limits to continuous sensitivity analysis, evaluating the polyhedral differentiability of the Nodal Tension LP and its Lipschitz continuity under $L_\infty$ perturbations to the edge weights.

## 4.4 Polyhedral Differentiability and Lipschitz Continuity

Because Corollary 4.11 establishes that local geometric capacities restrict the Jacobian sensitivity in Message Passing Neural Networks, it becomes necessary to determine how structurally sensitive these specific bottlenecks are to topological modifications or noise. In both environmental modelling and graph representation learning, edge weights are rarely static; they are frequently subject to observational uncertainty, adversarial perturbations, or active algorithmic updating during training.

While discrete algorithms compute cut capacities efficiently (Goldberg & Tarjan, 1988; Kleinberg & Tardos, 2006; Cormen et al., 2022), they yield combinatorial outputs that lack the continuous structure required for analytical sensitivity analysis or gradient-based backpropagation. Conversely, while global spectral metrics (such as the algebraic connectivity $\lambda_2$ (Fiedler, 1973)) are continuous and differentiable, their gradients are inherently dense. A structural perturbation anywhere within a dense graph shifts the global spectrum, causing spectral metrics to absorb irrelevant, out-of-distribution noise from regions far removed from the $s - t$ pathway.

To formalise a metric that is both differentiable and robust, we leverage the continuous linear programming formulation of Nodal Tension, which allows us to examine the polyhedral structure of the optimal solution. In this subsection, we transition from static geometric limits to continuous sensitivity analysis, evaluating the polyhedral differentiability of the Nodal Tension LP and its Lipschitz continuity under $L_\infty$ perturbations to the edge weights. By deriving the exact Clarke subdifferential of the $L_1$ bottleneck capacity, we prove that the continuous Nodal Tension subgradients are sparse and topologically localised to the minimal surface. This property establishes the Nodal Tension LP as a stable, differentiable layer that is structurally immune to noise occurring outside the weakest link.

**Lemma 4.12** (Marginal Sensitivity of Unique Bottlenecks). *Let $G = (V, E, w)$ be a weighted graph possessing a unique optimal Nodal Tension minimum $s - t$ cut, denoted $C^*$. Let $Z^*_{s,t}(w)$ be the optimal continuous objective value of the Nodal Tension LP as a function of the edge weight vector $w$. The partial derivative of the bottleneck capacity with respect to any individual continuous edge weight $w_{uv}$ is equal to the optimal integer tension variable $d^*_{uv}$:*

$$\frac{\partial Z^*_{s,t}}{\partial w_{uv}} = d^*_{uv} \tag{27}$$

*Consequently, by the integrality of the node potentials, the marginal sensitivity of the objective is binary and topologically localised exclusively to the minimal separating surface:*

$$\frac{\partial Z^*_{s,t}}{\partial w_{uv}} = \begin{cases} 1 & \text{if } (u, v) \in C^* \\ 0 & \text{if } (u, v) \notin C^* \end{cases} \tag{28}$$

*Proof Sketch.* The continuous Nodal Tension LP evaluates the objective min $\sum w_{ij} d_{ij}$ over a feasible region defined by the graph topology and boundary conditions, independent of the edge weights. To evaluate the partial derivative of the optimal capacity $Z^*_{s,t}$ from first principles, we apply a positive infinitesimal perturbation $\epsilon$ to a specific edge weight $w_{uv}$. Because the feasible region remains unchanged, evaluating the perturbed objective at the original optimal vertex establishes a strict upper bound on the rate of change. Symmetrically, evaluating the original objective at the perturbed optimal vertex establishes a strict lower bound. By applying the Squeeze Theorem as $\epsilon \to 0$, the continuous partial derivative evaluates to the corresponding optimal tension variable: $\frac{\partial Z^*_{s,t}}{\partial w_{uv}} = d^*_{uv}$.

By the established integrality of the Nodal Tension LP, the node potentials partition the graph into binary source and target components. To minimise the objective, the tension $d_{uv}^* = |p_u^* - p_v^*|$ is algebraically forced to evaluate to 1 if the edge spans this partition ($(u, v) \in C^*$), and to 0 if both endpoints reside within the same component ($(u, v) \notin C^*$). Substituting these exact binary states yields the piecewise derivative, proving algebraically that the capacity of a unique bottleneck possesses zero marginal sensitivity to continuous weight perturbations occurring anywhere outside the minimal separating surface. (See Appendix B.11 for the detailed proof). □

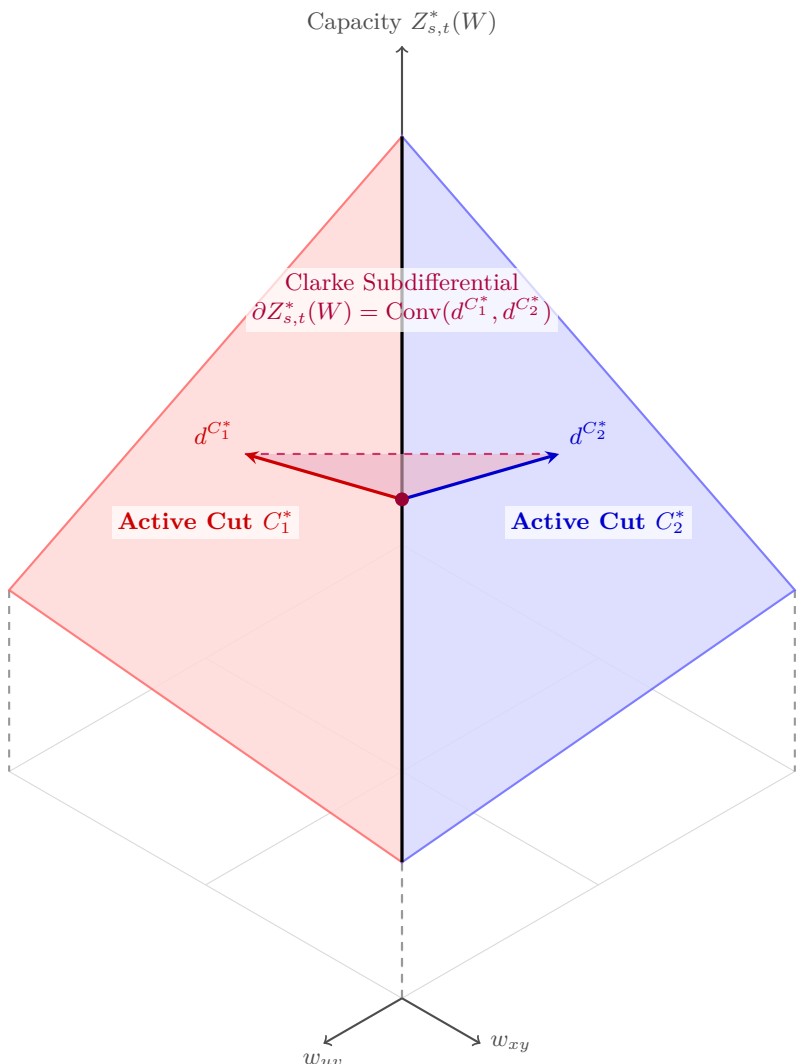

Figure 7: Visualisation of the exact polyhedral differentiability of the Nodal Tension capacity (Theorem 4.13). The continuous bottleneck capacity $Z_{s,t}^*(W)$ forms a globally concave, piecewise linear surface over the continuous edge weight parameters. The red and blue facets represent distinct parameter domains where a specific minimum cut ($C_1^*$ or $C_2^*$) is uniquely optimal. At the non-differentiable intersection ridge (topological degeneracy), both cuts actively define the capacity. While combinatorial algorithms fail at these intersections by discontinuously snapping between states, the exact Clarke subdifferential natively evaluates to the continuous convex hull of their binary tension vectors (the purple shaded fan). This unbroken polyhedral geometry provides the sparse, stable, and theoretically bounded backpropagation necessary for integrating local resilience into end-to-end Graph Structure Learning (Corollary 4.14).

**Theorem 4.13** (Exact Polyhedral Differentiability)**.** *Let $G = (V, E, w)$ be a weighted graph. The optimal Nodal Tension capacity $Z_{s,t}^*(w)$ is a globally concave, piecewise linear function with respect to the continuous edge weight vector $w \in \mathbb{R}_{>0}^{|E|}$. At points of topological degeneracy where multiple distinct minimum $s - t$ cuts exist with identical capacity, the objective function is non-differentiable. To evaluate the sensitivity at these degenerate intersections, we employ the Clarke subdifferential (Clarke, 1990). For a concave, piecewise linear function, the Clarke subdifferential at a non-differentiable point is defined as the convex hull of all limit gradients obtained by approaching the point from surrounding domains where the function is differentiable (illustrated in Figure 7). Let $\mathcal{C}^*$ denote the set of all valid minimum $s - t$ cuts actively defining the capacity at this parameterisation $w$. By Lemma 4.12, the exact limit gradient approaching from a domain where a cut $C^* \in \mathcal{C}^*$ is uniquely optimal is its binary tension vector. Therefore, the exact Clarke subdifferential of the capacity, denoted $\partial Z_{s,t}^*(w)$, evaluates to the convex hull of the binary optimal tension vectors corresponding to every active minimum cut in $\mathcal{C}^*$:*

$$\partial Z_{s,t}^*(w) = Conv\Big( \big\{ d^{C^*} \in \{0,1\}^{|E|} \mid C^* \in \mathcal{C}^* \big\} \Big) \tag{29}$$

*where $d^{C^*}$ is the binary indicator vector of the specific cut $C^*$, such that $d_{uv}^{C^*} = 1$ if $(u,v) \in C^*$, and $0$ otherwise. Consequently, any valid subgradient $g \in \partial Z_{s,t}^*(w)$ is bounded ($g_{uv} \in [0,1]$) and structurally sparse, assigning non-zero sensitivity exclusively to edges that actively participate in at least one optimal separating surface.*

*Proof Sketch.* The continuous Nodal Tension objective is formulated as a pointwise infimum over the finite extreme points (vertices) of the invariant feasible polyhedron $\mathcal{X}$: $Z_{s,t}^*(w) = \min_{v \in \mathcal{V}(\mathcal{X})} w^T d^{(v)}$. Because it is the minimum of a finite set of linear affine functions, the capacity is structurally guaranteed to be a globally concave, piecewise linear function of the edge weights.

At points of topological degeneracy, the minimum is achieved simultaneously by an active subset of vertices $\mathcal{V}^*(w)$. By the integrality property of the Nodal Tension LP (Theorem 3.2), this active vertex set maps bijectively to the set of active minimum cuts $\mathcal{C}^*$, where each active tension vector $d^{(v)}$ is the binary indicator vector $d^{C^*}$.

Because the standard gradient is undefined at the intersection of these active planes, we apply Danskin's Theorem (Danskin, 1966; Bertsekas, 1997) and the definition of the Clarke subdifferential for the pointwise minimum of affine functions. The exact generalised subgradient evaluates to the convex hull of the standard gradients of the active linear functions. Evaluating the standard gradient yields $\nabla_w(w^T d^{(v)}) = d^{(v)} \equiv d^{C^*}$. Therefore, the subdifferential is the convex hull of the active binary cut vectors.

By the algebraic definition of a convex combination, any subgradient extracted from this hull must be bounded ($g_{uv} \in [0,1]$). Furthermore, if an edge is disjoint from all active cuts in $\mathcal{C}^*$, its tension variable is $0$ across all spanning vectors, forcing the final subgradient to be zero and guaranteeing strict topological sparsity. (See Appendix B.12 for the complete polyhedral derivation). $\qquad \square$

**Corollary 4.14** (Sparse Backpropagation in Parametric Graph Models)**.** *Let $G = (V, E, W_\theta)$ be a graph where the edge weights are generated by a continuously differentiable mapping $W_\theta : \mathbb{R}^d \to \mathbb{R}_{>0}^{|E|}$ with parameters $\theta$. Let $J_W(\theta) \in \mathbb{R}^{|E| \times d}$ denote the Jacobian matrix of the edge generation function. By the chain rule for Clarke subdifferentials, the generalised gradient of the Nodal Tension capacity with respect to the model parameters is $\partial_\theta Z_{s,t}^*(W_\theta) = \{ J_W(\theta)^T g \mid g \in \partial_W Z_{s,t}^* \}$.*

*Let $E^* = \bigcup_{C^* \in \mathcal{C}^*} C^*$ define the support set of all actively optimal minimum cuts. For any valid parameter update $h \in \partial_\theta Z_{s,t}^*$, the vector computation collapses to:*

$$h = \sum_{e \in E^*} g_e \nabla_\theta W_e(\theta) \tag{30}$$

*where $g_e \in [0,1]$. Because the $L_0$ norm of the topological subgradient is bounded by $\|g\|_0 \leq |E^*|$, the computational complexity of the backward pass is independent of the global edge count $|E|$.*

*Proof.* Let $\theta \in \mathbb{R}^d$ dictate the edge weight vector $W_\theta$. We evaluate the sensitivity of the objective $Z_{s,t}^*$ with respect to $\theta$. By applying the standard chain rule for generalised gradients, any subgradient $h \in \partial_\theta Z_{s,t}^*$ is formed by the matrix-vector product of the transposed Jacobian $J_W(\theta)^T$ and a topological subgradient $g \in \partial_W Z_{s,t}^*$.

Expanding this matrix-vector product into a sum over the edge set yields:

$$h = \sum_{e \in E} g_e \nabla_\theta W_e(\theta) \tag{31}$$

By Theorem 4.13, the topological subgradient $g$ is defined as a convex combination of binary indicator vectors $d^{C^*}$ for $C^* \in \mathcal{C}^*$. Consider any edge $e \notin E^*$, meaning $e$ is structurally disjoint from all active optimal bottlenecks. By definition, its corresponding entry $d_e^{C^*} = 0$ for every spanning vector in the convex hull. Consequently, the extracted scalar evaluates to $g_e = 0$.

We substitute $g_e = 0$ for all $e \notin E^*$ into the expanded chain rule summation. This algebraic substitution annihilates all terms corresponding to the network complement $E \setminus E^*$, collapsing the global sum:

$$h = \sum_{e \in E^*} g_e \nabla_\theta W_e(\theta) + \sum_{e \notin E^*} 0 \cdot \nabla_\theta W_e(\theta) = \sum_{e \in E^*} g_e \nabla_\theta W_e(\theta) \tag{32}$$

The support of the topological gradient vector $g$ is bounded by the union of the active cuts, yielding $\|g\|_0 \leq |E^*|$. This completes the proof, algebraically guaranteeing that optimising the continuous Nodal Tension LP structurally truncates the parameter gradient $h$, restricting the model sensitivity to the localised Jacobian of the structural bottleneck. $\square$

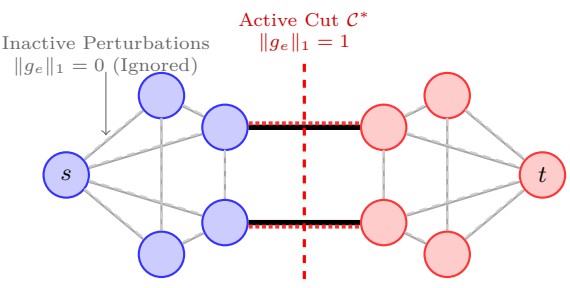

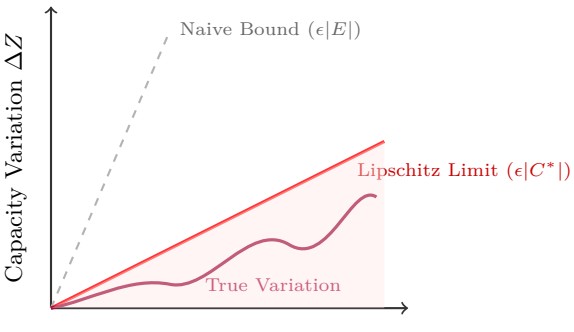

**(a)** The Global Noise Filter
Subgradient $g$ isolates the minimal surface

**(b)** The Lipschitz Bounding Cone
Variation is independent of global graph size

Figure 8: Visualisation of localised Lipschitz continuity and adversarial robustness under $L_\infty$ edge weight perturbations (Corollary 4.15). **(a)** The global noise filter. A uniform noise matrix $\Delta W$ perturbs every edge in the graph (represented by dashed overlays). However, evaluating the Clarke subgradient $g$ reveals that $\|g_e\|_1 = 0$ for all edges interior to the topological partitions. The Nodal Tension LP algebraically annihilates the vast majority of the noise, rendering the continuous objective structurally sensitive only to perturbations directly striking the minimal surface edges ($|C^*| = 2$). **(b)** The tightness of the analytical bound. A naive global metric scaling with the total edge count $|E|$ produces a large vulnerability cone (steep grey dashed line). Conversely, the Hölder inequality (Hardy et al., 1952) bound establishes a tight local Lipschitz envelope scaled strictly by the cut cardinality $|C^*|$ (red line). The true capacity variation $\Delta Z$ (purple curve) resides within this restricted feasible region (shaded red). This geometric constraint guarantees that Message Passing bottlenecks identified by the LP remain structurally stable against widespread, chaotic global noise.

**Corollary 4.15** (Lipschitz Continuity under $L_\infty$ Perturbations)**.** *Let $G = (V, E, W)$ be a weighted graph. Let $\Delta W \in \mathbb{R}^{|E|}$ be an additive edge weight perturbation matrix bounded by the $L_\infty$ norm: $\|\Delta W\|_\infty \leq \epsilon$. Let*

$C^*$ *define the set of active minimum $s - t$ cuts for the unperturbed weights $W$. The Nodal Tension capacity $Z_{s,t}^*$ is locally Lipschitz continuous. The absolute variation of the optimal objective under the continuous perturbation $\Delta W$ is bounded by:*

$$|Z_{s,t}^*(W + \Delta W) - Z_{s,t}^*(W)| \leq \epsilon \max_{C^* \in \mathcal{C}^*} |C^*| \tag{33}$$

*This establishes that the local Lipschitz constant with respect to the $L_\infty$ norm is the cardinality of the maximal active separating surface. The metric's deviation is structurally independent of the global edge count $|E|$ (visually demonstrated by the bounding cone in Figure 8b).*

*Proof.* Let $Z(W) = Z_{s,t}^*(W)$ denote the optimal objective value function. We evaluate the absolute difference $|Z(W + \Delta W) - Z(W)|$ subject to the perturbation bound $\|\Delta W\|_\infty \leq \epsilon$.

By the Lebourg Mean Value Theorem for locally Lipschitz continuous functions (Clarke, 1990), there exists a scalar $t \in (0, 1)$ and a generalised Clarke subgradient $g \in \partial Z(W + t\Delta W)$ such that the exact objective difference evaluates to the inner product:

$$Z(W + \Delta W) - Z(W) = g^T \Delta W \tag{34}$$

To establish the strict upper bound on this inner product, we apply Hölder's Inequality (Hardy et al., 1952):

$$|g^T \Delta W| \leq \|g\|_1 \|\Delta W\|_\infty \leq \epsilon \|g\|_1 \tag{35}$$

We must now evaluate the strict algebraic upper bound for the $L_1$ norm of the subgradient, $\|g\|_1$. By Theorem 4.13, for an infinitesimally bounded local perturbation ($\epsilon \to 0$), the subgradient $g$ is defined as a convex combination of the active binary cut vectors $d^{C^*}$ for $C^* \in \mathcal{C}^*$:

$$g = \sum_{C^* \in \mathcal{C}^*} \alpha_{C^*} d^{C^*}, \quad \text{subject to } \sum \alpha_{C^*} = 1, \quad \alpha_{C^*} \geq 0 \tag{36}$$

Because every vector $d^{C^*}$ is binary ($d_e^{C^*} \in \{0, 1\}$), the $L_1$ norm of any specific basis vector evaluates to its set cardinality: $\|d^{C^*}\|_1 = |C^*|$. Applying the triangle inequality to the convex combination yields the strict maximum bound for the subgradient norm:

$$\|g\|_1 = \Big\| \sum_{C^* \in \mathcal{C}^*} \alpha_{C^*} d^{C^*} \Big\|_1 \leq \sum_{C^* \in \mathcal{C}^*} \alpha_{C^*} \|d^{C^*}\|_1 = \sum_{C^* \in \mathcal{C}^*} \alpha_{C^*} |C^*| \leq \max_{C^* \in \mathcal{C}^*} |C^*| \tag{37}$$

Substituting this exact topological bound back into the Hölder inequality chain isolates the final absolute limit:

$$|Z(W + \Delta W) - Z(W)| \leq \epsilon \max_{C^* \in \mathcal{C}^*} |C^*| \tag{38}$$

This completes the proof, establishing that the metric's absolute vulnerability to continuous uniform noise is scaled by the local edge cardinality of the bottleneck, rejecting all perturbation mass $\Delta w_e$ located outside the active surfaces (as depicted by the structural noise filter in Figure 8a). $\square$

**Conclusion of the Theoretical Analysis:** The progression from Section 3 through Section 4 connects classical operations research to continuous graph geometry and representation learning. The motivating objective of this work is to quantify local network resilience in environmental corridors (Urban & Keitt, 2001), a setting where isolating the specific topological weakest link between two habitats is empirically necessary. Section 3 formulated this local vulnerability through the continuous Nodal Tension LP, contrasting its exact $s - t$ combinatorial boundaries (Theorems 3.1 and 3.2) with the global, dense evaluation characteristic of spectral algebraic connectivity (Fiedler, 1973).

Section 4 validates the necessity of this localised approach. By establishing the structural redundancy gap between $L_1$ tension and average-case $L_2$ effective resistance (Lemma 4.1), and mapping the optimal

Table 2: Quantitative validation of the Nodal Tension model's correctness. This table compares the optimal value ($Z^*$) of our linear program against the minimum $s - t$ cut value computed by the standard Edmonds-Karp algorithm. The comparison is performed across multiple graph topologies: Erdős-Rényi (GNP) random graphs, Barabási-Albert (BA) scale-free networks, regular grid graphs, and a real-world network (Karate Club). The tests cover three different sizes (Small, Medium, Large) and both unweighted and weighted scenarios. The final column confirms a perfect match between the two methods in all test cases, providing empirical support for Theorem 3.1.

| Graph Type | Size | Weighted | Min-Cut (NetworkX) | Optimal LP Value (Z*) | Match? |
|---|---|---|---|---|---|
| Random (GNP) | Small | No | 3.00 | 3.00 | ✓ |
| Random (GNP) | Small | Yes | 15.00 | 15.00 | ✓ |
| Scale-Free (BA) | Small | No | 3.00 | 3.00 | ✓ |
| Scale-Free (BA) | Small | Yes | 17.00 | 17.00 | ✓ |
| Grid | Small | No | 2.00 | 2.00 | ✓ |
| Grid | Small | Yes | 5.00 | 5.00 | ✓ |
| Random (GNP) | Medium | No | 10.00 | 10.00 | ✓ |
| Random (GNP) | Medium | Yes | 39.00 | 39.00 | ✓ |
| Scale-Free (BA) | Medium | No | 3.00 | 3.00 | ✓ |
| Scale-Free (BA) | Medium | Yes | 28.00 | 28.00 | ✓ |
| Grid | Medium | No | 2.00 | 2.00 | ✓ |
| Grid | Medium | Yes | 3.00 | 3.00 | ✓ |
| Random (GNP) | Large | No | 21.00 | 21.00 | ✓ |
| Random (GNP) | Large | Yes | 113.00 | 113.00 | ✓ |
| Scale-Free (BA) | Large | No | 3.00 | 3.00 | ✓ |
| Scale-Free (BA) | Large | Yes | 20.00 | 20.00 | ✓ |
| Grid | Large | No | 2.00 | 2.00 | ✓ |
| Grid | Large | Yes | 5.00 | 5.00 | ✓ |
| Karate Club | Real-World | No | 10.00 | 10.00 | ✓ |
| Karate Club | Real-World | Yes | 53.00 | 53.00 | ✓ |

LP capacity to negatively curved discrete manifolds (Theorem 4.6), we geometrically isolate the structural bottleneck. Theorem 4.10 subsequently restricts random walk commute times using this exact local capacity, which Corollary 4.11 translates into a strict attenuation bound on the Jacobian sensitivity of Message Passing Neural Networks (Topping et al., 2022). This sequence proves that the local Nodal Tension bottleneck governs dynamic graph flow independent of the global network scale.

Finally, Section 4.4 justifies the continuous LP framework over standard discrete combinatorial algorithms (Goldberg & Tarjan, 1988) by establishing its continuous sensitivity properties. By extracting the exact Clarke subdifferential from the polyhedral intersection (Theorem 4.13), we demonstrate that the metric natively yields structurally sparse parameter gradients (Corollary 4.14) and local Lipschitz continuity bounded by the cut cardinality (Corollary 4.15). While the immediate empirical evaluation of this metric remains focused on static environmental connectivity analysis (Section 5), the exact differentiability and $L_\infty$ robustness established herein provide a bounded basis for integrating local resilience into future end-to-end Graph Structure Learning architectures.

## 5 Experimental Validation

In this section, we present the experimental validation of our Nodal Tension model. We begin in Sections 5.1 and 5.2 by establishing the computational correctness, integrality, and spectral relationships of our model against standard benchmark algorithms. Next, Sections 5.3 through 5.6 provide empirical validation for the theoretical bounds derived in Section 4, evaluating the model's behaviour on redundancy, curvature filtering, MPNN over-squashing, and differentiable robustness. Finally, we apply the validated model to a large-scale environmental problem in Section 5.7.

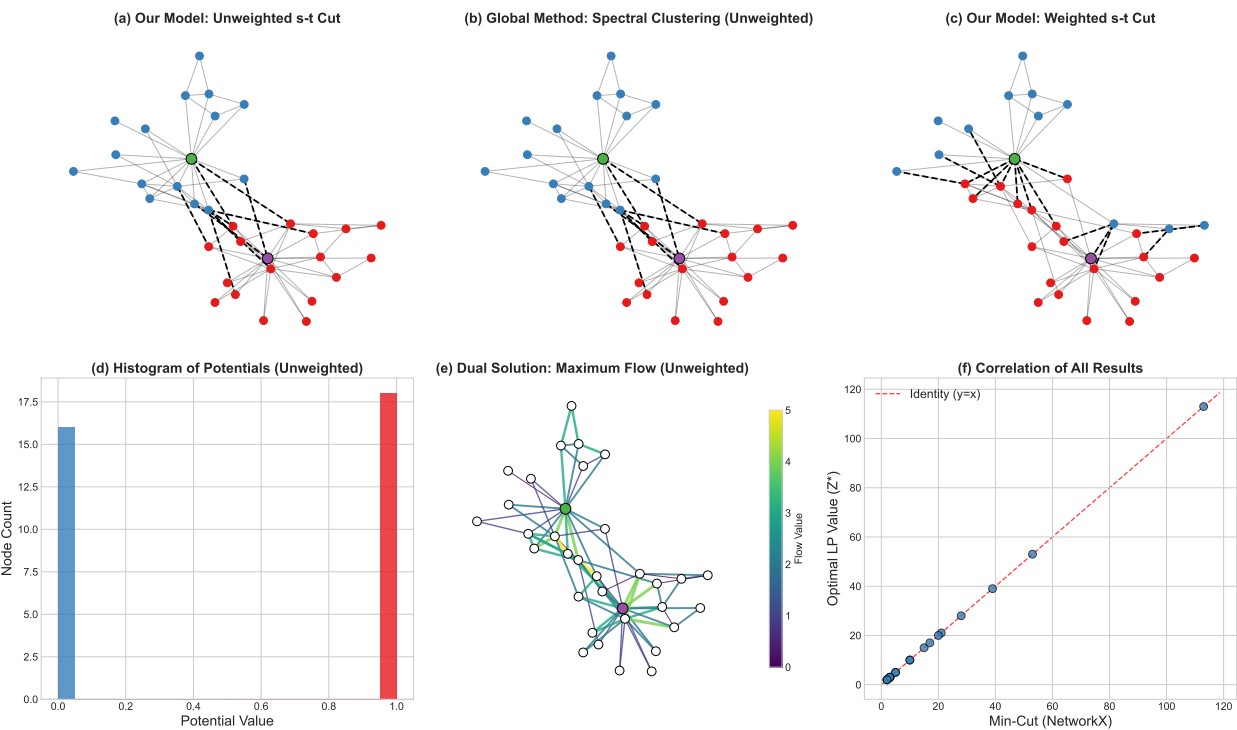

Figure 9: Validation of the Nodal Tension model. **(a)** Our model correctly finds the minimum $s-t$ cut in the unweighted Zachary's Karate Club graph, separating the two faction leaders (nodes 0 and 33). **(b)** A global partitioning method, Spectral Clustering, finds a nearly identical partition, validating that our local model correctly identifies the dominant global structure in this well-known graph. **(c)** On a weighted version of the graph where inter-faction edges are made expensive, our model correctly refines the cut based on edge costs. **(d)** A histogram of the optimal potentials from the unweighted solution shows two sharp peaks at 0 and 1, providing computational support for our Integrality Theorem 3.2. **(e)** A visualisation of the dual solution shows that the maximum flow paths as thick, bright "arteries" that converge on and cross the exact edges of the minimum cut. **(f)** A correlation plot of our LP's optimal value against a standard NetworkX algorithm for all experiments in Table 2, showing perfect agreement.

## 5.1 Validation of Model Correctness

To validate the correctness of our model, we compared its output against the standard Edmonds-Karp max-flow algorithm implemented in the NetworkX library. We performed this comparison across test cases including Erdős-Rényi (GNP) random graphs, Barabási-Albert (BA) scale-free networks, regular grid graphs, and the Zachary's Karate Club graph. Each topology was tested at three scales (Small, Medium, Large) and in both unweighted and weighted configurations.

The quantitative results, summarised in Table 2, show a match between the optimal value $Z^*$ of our LP and the minimum $s-t$ cut value computed by the NetworkX algorithm in every test case. This provides empirical evidence that our model correctly solves the minimum $s-t$ cut problem, validating Theorem 3.1.

Figure 9 provides visual confirmation of these results. The top row shows the model's output on diverse unweighted topologies. The bottom row demonstrates that the model adapts to engineered edge weights, finding a new minimum cut that minimises cost. Panel (f) provides quantitative evidence, showing correlation between our model's output and the standard algorithm across all experiments.

## 5.2 Validation of Theoretical Properties

Table 3: Quantitative validation of the integrality theorem and analysis of cut balance across diverse network topologies. The 'Is Integral?' column provides computational verification of Theorem 3.2, showing that a binary, integer-valued potential solution was found in every test case. The 'Variance of Potential' column quantifies the balance of the resulting s-t cut. A variance near 0.25 indicates a balanced partition into two large sets of nodes, characteristic of the community structure in the Karate Club graph. In contrast, a variance near 0 indicates a highly unbalanced or "trivial" cut, where the model isolates an endpoint to sever the connection in the unstructured networks.

| Graph Type | Size | Weighted | Is Integral? | Variance of Potentials |
|---|---|---|---|---|
| Random (GNP) | Small | No | ✓ | 0.010 |
| Random (GNP) | Small | Yes | ✓ | 0.010 |
| Scale-Free (BA) | Small | No | ✓ | 0.010 |
| Scale-Free (BA) | Small | Yes | ✓ | 0.010 |
| Grid | Small | No | ✓ | 0.010 |
| Grid | Small | Yes | ✓ | 0.010 |
| Random (GNP) | Medium | No | ✓ | 0.004 |
| Random (GNP) | Medium | Yes | ✓ | 0.004 |
| Scale-Free (BA) | Medium | No | ✓ | 0.004 |
| Scale-Free (BA) | Medium | Yes | ✓ | 0.004 |
| Grid | Medium | No | ✓ | 0.004 |
| Grid | Medium | Yes | ✓ | 0.004 |
| Random (GNP) | Large | No | ✓ | 0.002 |
| Random (GNP) | Large | Yes | ✓ | 0.002 |
| Scale-Free (BA) | Large | No | ✓ | 0.002 |
| Scale-Free (BA) | Large | Yes | ✓ | 0.002 |
| Grid | Large | No | ✓ | 0.002 |
| Grid | Large | Yes | ✓ | 0.004 |
| Karate Club | Real-World | No | ✓ | 0.249 |
| Karate Club | Real-World | Yes | ✓ | 0.242 |

We conducted two experiments to computationally validate the theoretical properties of our model's solutions regarding integrality and spectral relationships.

### 5.2.1 Integrality and Cut Balance

First, we evaluated our integrality theorem (Theorem 3.2) on the established suite of graphs. The results in Table 3 show that an integer-valued potential solution was found in every test run, providing empirical support for the theoretical proof. The table uses the variance of the potentials as a quantitative measure of cut balance. The structured Karate Club graph yields a high-variance, balanced cut (variance $\approx 0.25$), whereas the unstructured networks (Grid, GNP, BA) result in low-variance cuts where isolating an endpoint is optimal.

Figure 10 illustrates these findings. The histograms of potentials for the Grid, GNP, and BA graphs are skewed, corresponding to the low-variance unbalanced cuts in the insets. The histogram for the Karate Club graph is bimodal, reflecting the partition into two distinct communities. This indicates that the distribution of integer solutions corresponds to structural differences between networks.

### 5.2.2 Relationship to Spectral Connectivity

Second, we performed an experiment to evaluate the theoretical bound connecting our local resilience measure $Z^*$ to the global algebraic connectivity $\lambda_2$ (Theorem 3.3). We constructed a graph with two dense cliques connected by a tunable bottleneck of capacity $k$.

Figure 11 visualises the result. The top row shows that our local model identifies the cheapest path between $s$ and $t$, finding the cut on the bridge until its cost exceeds an alternative cut ($k = 15$). The bottom row

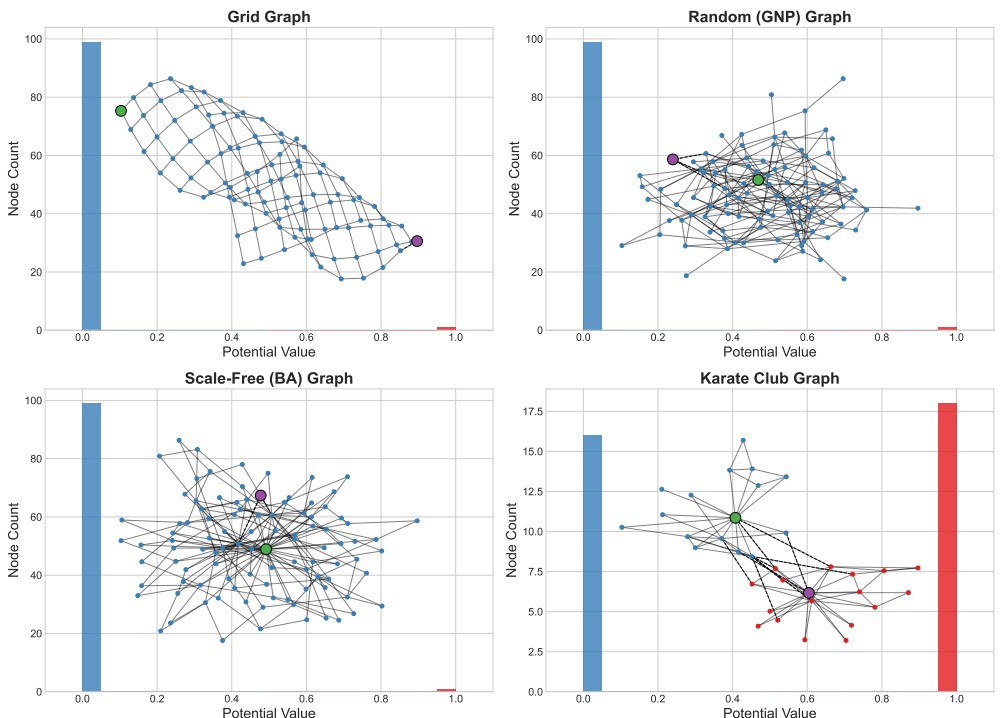

Figure 10: Computational validation of the integrality theorem and visualisation of cut balance. Each panel displays a histogram of the optimal node potentials for a representative unweighted graph, with an inset showing the corresponding minimum $s-t$ cut. **(a, b, c)** For the Grid, Random (GNP), and Scale-Free (BA) graphs, the histograms are highly skewed. This reflects the result that the minimum cut in these topologies isolates an endpoint. This visual evidence explains the low potential variance reported in Table 3. **(d)** In contrast, for the Karate Club graph, which possesses a community structure, the histogram shows two well-balanced peaks at 0 and 1. This corresponds to the balanced partition found by the model. The binary nature of the potentials across all graph types provides empirical support for our integrality theorem.

shows that the global spectral method continues to identify the bridge as the main bottleneck even when $k = 15$ because it optimises for global sparsity. Figure 12 supports this, showing the linear growth of $Z^*$ versus the sub-linear growth of $\lambda_2$.

We evaluated the tightness of our spectral bound using a Monte Carlo search on the Karate Club graph. As shown in Figure 13, the search identifies a pair for which our local model's bound equals the Cheeger bound, demonstrating that the local model can recover the globally sparsest cut given specific endpoints.

## 5.3 The Redundancy Gap: $L_1$ vs $L_2$ Pathways

Having validated the model's fundamental properties and its relation to global spectral connectivity in Section 5.1 and 5.2, we now empirically evaluate the redundancy gap formalised in Section 4.1. This experiment contrasts the worst-case structural isolation identified by the $L_1$ Nodal Tension against the average-case connectivity modelled by $L_2$ effective resistance (Circuit Theory (Chandra et al., 1989; Spielman & Srivastava, 2008)). We accomplish this through two controlled experiments: one evaluating spatial routing behaviour, and another evaluating distance decay.

**Spatial Routing Behaviour.** We first generated two spatial grid networks designed to force a divergence between global, average-case, and worst-case routing. The first topology is an Asymmetric Hourglass, comprising a $20 \times 15$ node grid connected to a $5 \times 5$ grid via a narrow 2-edge bridge. The second topology is a U-Trap, comprising a $30 \times 20$ grid where the target node is enclosed by a structural wall, leaving only a 2-edge opening. In both networks, the source and target nodes are heavily connected to their immediate

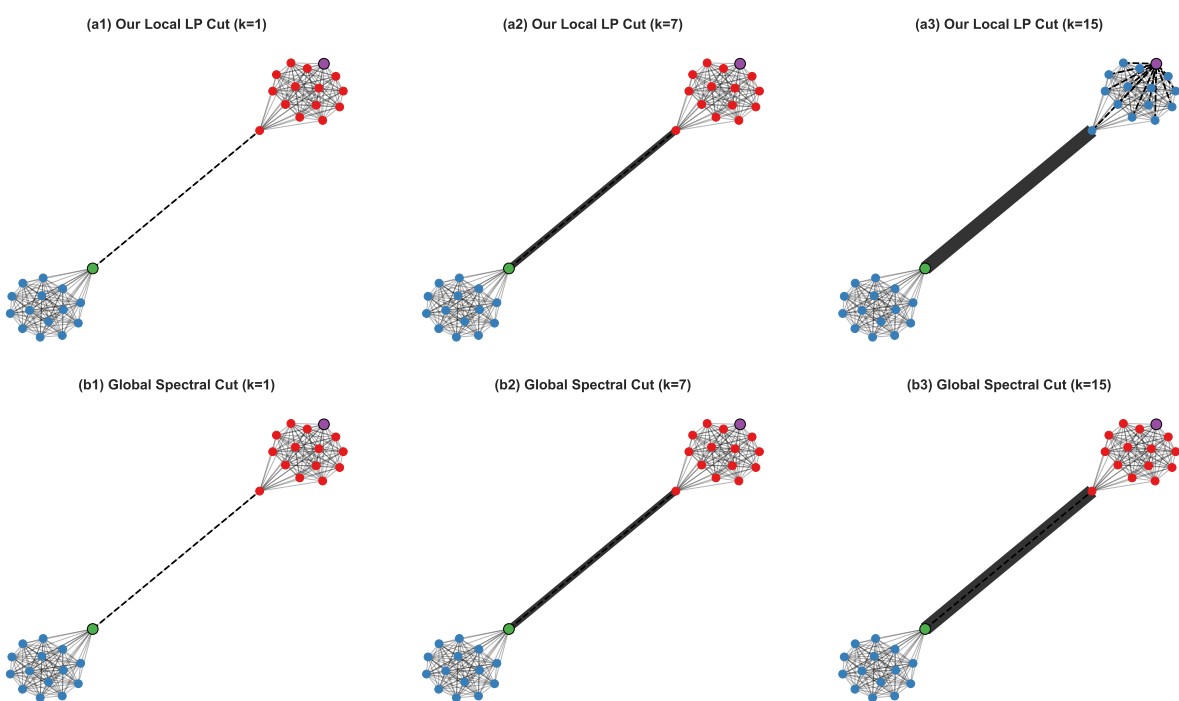

Figure 11: Visual comparison of our local s-t cut (top row) versus the global spectral cut (bottom row) on a network with a tunable bottleneck of capacity $k$. **(a1, a2, b1, b2)** For small bottleneck capacities ($k = 1, 7$), both our local model and the global spectral method identify the bridge as the minimum cut. **(a3)** For a large capacity ($k = 15$), our local model determines that it is cheaper to sever the 14 edges connected to the target node (a cut of cost 14) than to cut the bridge (cost 15). **(b3)** In contrast, the global spectral method continues to identify the bridge as the global partition, as cutting it separates the graph into two large, balanced sets. This demonstrates that our model evaluates local resilience distinctly from global structural analysis.

neighbourhoods to prevent the LP from identifying trivial cuts. This configuration guarantees that the local bottleneck capacity is $Z_{s,t}^* = 2.00$ at the defects.

The spatial routing results are visualised in Figure 14. The global spectral cut (Column 1) partitions the graphs based on structural volume, separating the main bodies rather than identifying the targeted $s - t$ bottlenecks. The $L_2$ circuit theory potentials (Column 2) exhibit a continuous spatial gradient, bleeding around the U-Trap and spreading across all redundant pathways. Because $L_2$ conductance is reduced by path length (resistors in series), the effective conductances ($C_{s,t} = 0.29$ and $0.26$) underestimate the local capacity, empirically demonstrating the upper bound $C_{s,t} \leq Z_{s,t}^*$ established in Lemma 4.1. In contrast, the $L_1$ Nodal Tension (Column 3) isolates the worst-case structural failure. The optimal potentials adopt strict binary states, fracturing the networks at the 2-edge defects and recovering the capacity of $Z_{s,t}^* = 2.00$.

**Distance Decay and Flow Saturation.** To evaluate how this redundancy gap behaves over physical distance, we constructed an "elastic corridor" topology. This network consists of two dense terminal cliques (representing source and target habitats) connected by a uniform grid corridor of a fixed width ($W = 3$). We iteratively elongated the corridor length $L$ from 1 to 30 edges. At each length, we computed both $C_{s,t}$ and $Z_{s,t}^*$.

The results of this progression are shown in Figure 15. As the corridor elongates, the $L_1$ bottleneck capacity remains invariant at $Z_{s,t}^* = 3.00$, verifying that Nodal Tension evaluates only the absolute structural limitation independent of distance. Conversely, the $L_2$ effective conductance decays inversely with length ($C_{s,t} \propto 1/L$) due to accumulating series resistance. This divergence provides direct empirical validation for

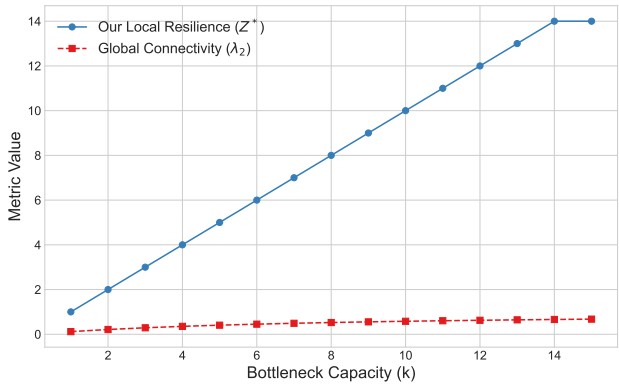

Figure 12: Quantitative comparison of our local resilience score ($Z^*$) versus the global algebraic connectivity ($\lambda_2$) as a function of the graph's bottleneck capacity, $k$. The blue line shows that our local resilience score $Z^*$ increases linearly with the bottleneck capacity and plateaus at a value of 14, where isolating the target node becomes more efficient. The red line shows that the global connectivity $\lambda_2$ increases at a sub-linear rate.

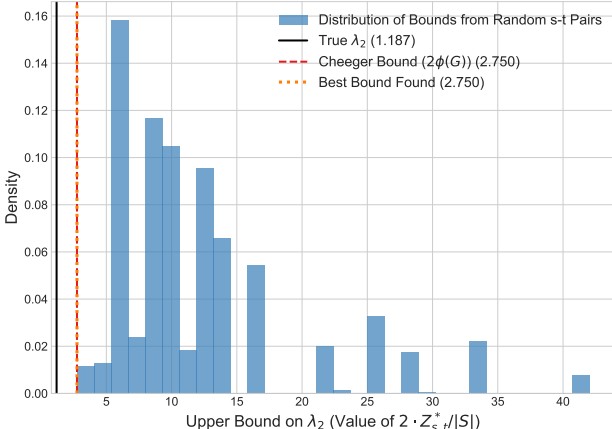

Figure 13: Distribution of upper bounds on the algebraic connectivity ($\lambda_2$) obtained by applying our local resilience model to 1,000 random source-target pairs on the Karate Club graph. The histogram shows the variation in bounds derived from local s-t cuts. The search identifies a pair whose bound (orange dotted line) is equal to the Cheeger bound ($2\phi(G)$, red dashed line). The gap between these bounds and the true $\lambda_2$ (black solid line) illustrates the Cheeger inequality gap.

Corollary 4.2. When $L = 1$, the gap approaches zero ($\Delta_{s,t} = 0.026$), confirming the definition of a strict topological bridge where the unit potential difference drops across the bottleneck. As $L$ increases to 30, the gap widens toward the capacity limit ($\Delta_{s,t} \to Z_{s,t}^*$), transitioning into a diffuse bottleneck. Furthermore, the visual attenuation of the continuous $L_2$ potentials across the extended path (Figure 15c, d) physically demonstrates Corollary 4.3 (Flow Saturation). The $L_2$ flow dissipates across the prolonged corridor, meaning the electrical signal attenuates completely before it can fully utilise the absolute $L_1$ capacity of the structural bottleneck.

Together, these empirical results confirm that the $L_1$ Nodal Tension model isolates the structural capacity of a bottleneck, remaining independent of global graph volume and corridor length. This prevents the metric from underestimating vulnerabilities in extended networks, a known limitation of $L_2$ circuit theory (Von Luxburg et al., 2014; Alon & Yahav, 2021) and spectral methods (Guattery & Miller, 1998). Having established that the model successfully ignores macro-level structural redundancy and distance, we must also evaluate how it handles micro-level geometric irregularities, such as topological dead-ends or spurious

edges. In the following subsection, we test this local filtering property by comparing Nodal Tension against discrete Forman-Ricci curvature on standard machine learning datasets.

Figure 14: Comparison of network partitioning and routing objectives across two spatial networks: an asymmetric hourglass (top row) and a U-trap (bottom row). **(a, d)** The global spectral cut partitions the graphs based on global structural volume ($\lambda_2 = 0.0177$ and $0.0086$), separating the main bodies rather than the targeted $s - t$ bottlenecks. **(b, e)** $L_2$ circuit theory evaluates average-case connectivity. The continuous potentials explore redundant pathways, and the resulting effective conductances ($C_{s,t} = 0.29$ and $0.26$) are reduced by path length (resistors in series). **(c, f)** $L_1$ Nodal Tension evaluates worst-case local connectivity. The optimal potentials adopt strict binary states, fracturing the networks at the 2-edge topological defects. This identifies the local bottleneck capacity ($Z_{s,t}^* = 2.00$), demonstrating the metric's independence from global graph volume and redundant path length.

## 5.4 Geometric Filtering and Spurious Bottleneck Rejection

While Section 5.3 demonstrated that Nodal Tension is invariant to macroscopic path length and global volume, we must also empirically validate its response to microscopic geometric irregularities. To validate the spurious bottleneck rejection property derived in Section 4.2, we evaluate the discrete Forman-Ricci curvature against continuous Nodal Tension on both adversarial topologies and a standard machine learning benchmark dataset. The objective is to compute the intersection between edges flagged by negative differential curvature and edges actively carrying Nodal Tension, empirically demonstrating the LP's ability to filter out non-constraining structural dead-ends.

**Microscopic Filtering on Geometric Decoys.** We first created two targeted networks designed to expose the known vulnerabilities of local curvature metrics. Both networks feature a true macroscopic bottleneck (a 1-edge bridge connecting two dense cliques). To trick the local metric, we attached geometric "decoys" to the source clique. The first topology (Figure 16, top row) features a spurious tree, representing a branching dead-end path. The second topology (Figure 16, bottom row) features a spurious hub, representing a high-degree star node connected to isolated leaves.

As visualised in Figure 16(a, c), the discrete Forman-Ricci curvature fails to identify the global structural restriction. The true bridge connects two moderate-degree nodes but forms no shared triangles, resulting in a neutral curvature of $\mathbf{F} = 0$. Conversely, the local geometry of the decoys drives their curvature heavily

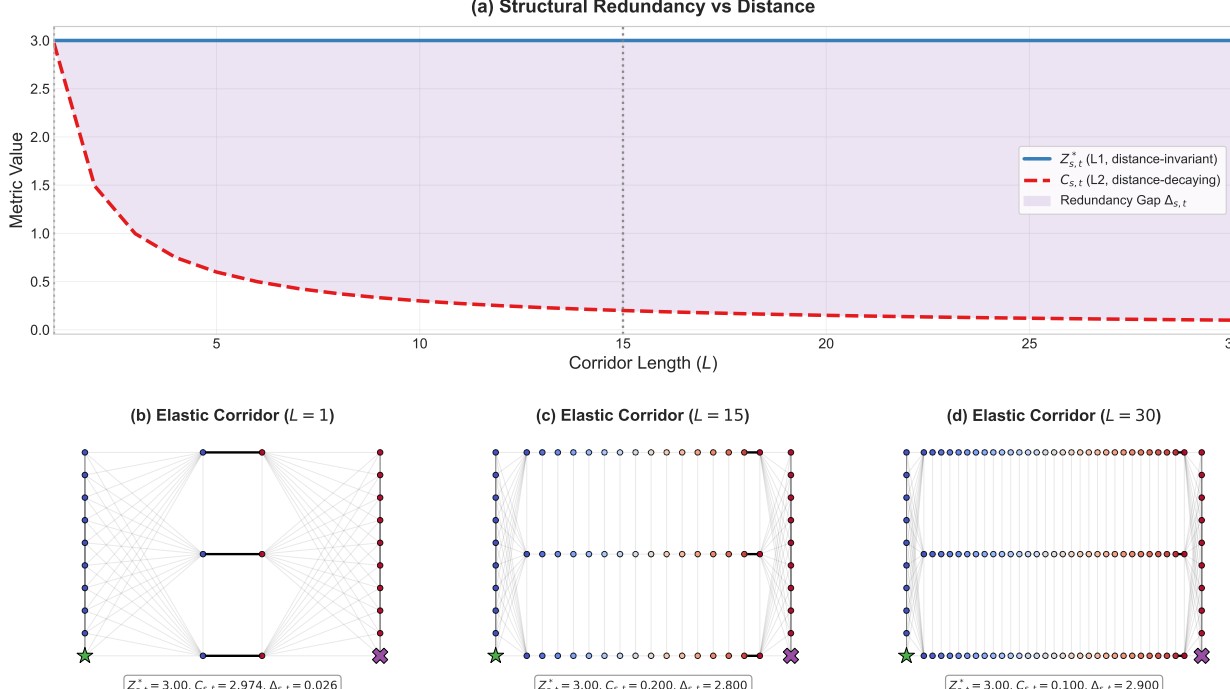

Figure 15: Empirical evaluation of distance decay and the redundancy gap ($\Delta_{s,t}$) on an elastic corridor topology. **(a)** Quantitative comparison as the corridor length ($L$) increases. The $L_1$ bottleneck capacity ($Z_{s,t}^* = 3.00$, solid line) remains invariant to distance, evaluating only the local structural limitation. In contrast, the $L_2$ effective conductance ($C_{s,t}$, dashed line) decays inversely with length due to series resistance, causing the redundancy gap (shaded region) to widen. **(b)** At $L = 1$, the network forms a strict topological bridge ($\Delta_{s,t} = 0.026$); the $L_2$ potential drops sharply across the bottleneck. **(c, d)** As the corridor elongates to $L = 15$ and $L = 30$, it transitions into a diffuse bottleneck ($\Delta_{s,t} \to Z_{s,t}^*$). The continuous $L_2$ potentials (visualised by node colour) attenuate gradually across the extended path. This confirms Corollary 4.3, demonstrating that $L_2$ flow dissipates across the prolonged corridor rather than utilising the absolute $L_1$ capacity of the bottleneck.

negative ($\mathbf{F} = -4$ for the tree, $\mathbf{F} = -12$ for the hub), falsely flagging them as severe bottlenecks. Figure 16(b, d) demonstrates the corrective filtering of the continuous Nodal Tension. The LP assigns strict zero tension ($\tau = 0$) to the highly negatively curved decoys, analytically routing around the local geometric noise to isolate the true structural bridge ($\tau = 1$). This provides a controlled, empirical proof of Corollary 4.7.

**Macroscopic Filtering on the Cora Benchmark.** To verify that these geometric bounds scale to real-world, highly irregular network data, we evaluated both metrics on the largest connected component of the standard Cora citation network (Sen et al., 2008). We selected a source and target separated by a significant shortest-path distance, forcing the continuous LP to evaluate the complete spatial manifold spanning the core of the graph.

The results on the Cora network are presented in Figure 17. The discrete curvature map (Figure 17a) exhibits diagnostic ambiguity, reporting thousands of edges with highly negative curvature across the citation graph. In contrast, the continuous Nodal Tension (Figure 17b) isolates the active $s-t$ manifold, analytically filtering out the surrounding topological noise. The edge count histogram (Figure 17d) confirms this behaviour at scale: the LP assigns zero tension to thousands of spurious edges, even those with curvature descending to $\mathbf{F} \approx -150$, validating the Spurious Bottleneck Rejection condition (Corollary 4.7) on dense data.

Crucially, this experiment verifies the explicit bounds linking the continuous tension variables to the discrete geometry. As shown in Figure 17e, every active edge satisfies the shared-triangle upper bound ($|\triangle_{uv}| \le$

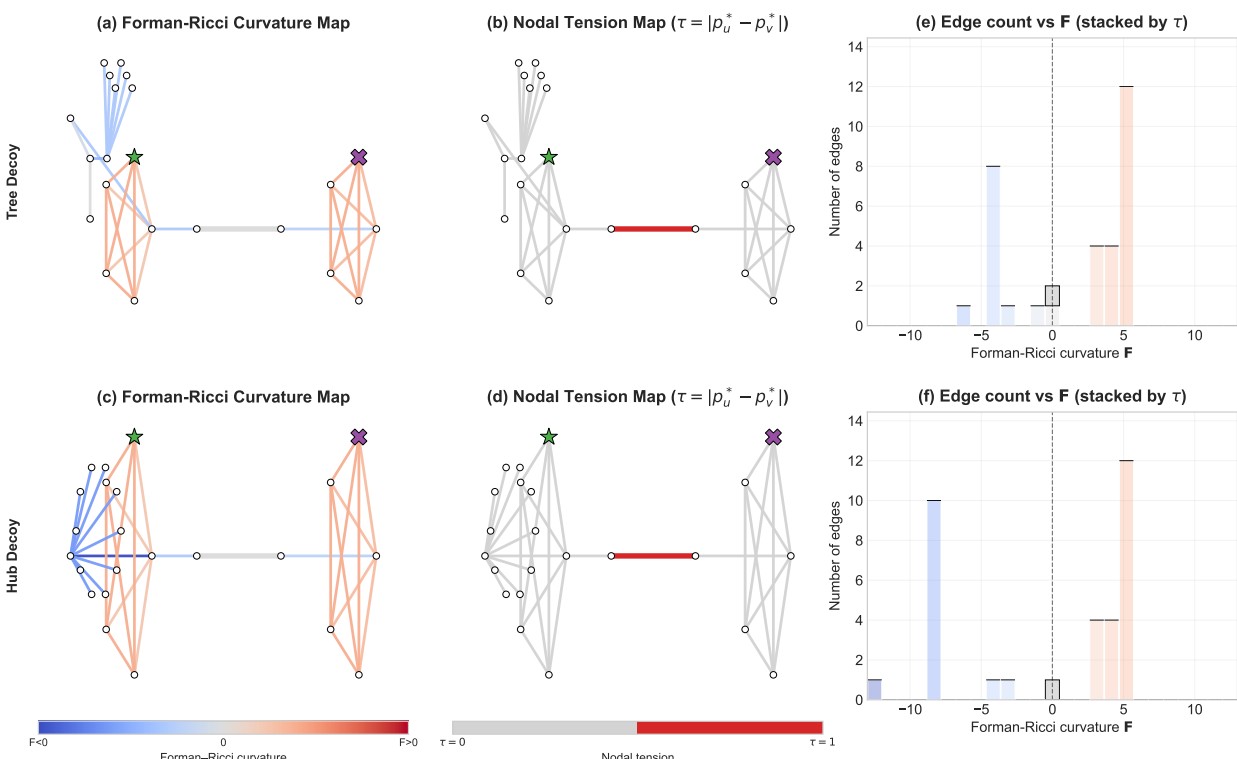

Figure 16: Empirical validation of spurious bottleneck rejection (Corollary 4.7) on networks with geometric decoys: a spurious tree (top row) and a spurious hub (bottom row). **(a, c)** Forman-Ricci curvature maps evaluate local neighbourhood dispersion. In both topologies, the decoy structures exhibit highly negative curvature ($\mathbf{F} < 0$), mimicking the geometric profile of topological bridges. **(b, d)** Nodal Tension maps ($\tau = |p_u^* - p_v^*|$) evaluate the global $s - t$ partition. The continuous potentials assign zero tension ($\tau = 0$) to the geometrically irregular decoys, analytically filtering them out to isolate the true structural bridge ($\tau = 1$). **(e, f)** Histograms of edge counts distributed by local curvature and stacked by assigned tension. The true macroscopic bottleneck is correctly identified despite possessing a neutral curvature ($\mathbf{F} = 0$). Conversely, the edges exhibiting the most severe negative curvature ($\mathbf{F} \ll 0$) are rejected by the LP, demonstrating that Nodal Tension is invariant to isolated local differential geometry.

$Z_{s,t}^* - 1$) established in Lemma 4.4. Furthermore, Figure 17f verifies the Geometric Localisation limit established in Theorem 4.6; the aggregate discrete curvature of the isolated active manifold is bounded by the theoretical limit derived from the global capacity and local node degrees, satisfying the inequality with a quantitative slack of 24.00.

Together, these results confirm that Nodal Tension acts as a global filter for local differential geometry. Because the metric correctly identifies true structural bottlenecks while rejecting spurious dead-ends and isolated hubs, it resolves the diagnostic ambiguity that limits local curvature measures. Having established this topological filtering capability, we can now apply the $L_1$ bottleneck capacity to diagnose functional failures in graph learning algorithms. In the following subsection, we utilise Nodal Tension to analytically predict the onset of over-squashing in Message Passing Neural Networks (MPNNs).

## 5.5 Predicting MPNN Over-Squashing

To validate the dynamical bounds governing information flow detailed in Section 4.3, we simulate a linear Message Passing Neural Network (MPNN) on structurally constrained synthetic graphs. Because Corollary 4.11 establishes a strict algebraic threshold for Jacobian sensitivity, our objective is to empirically verify that

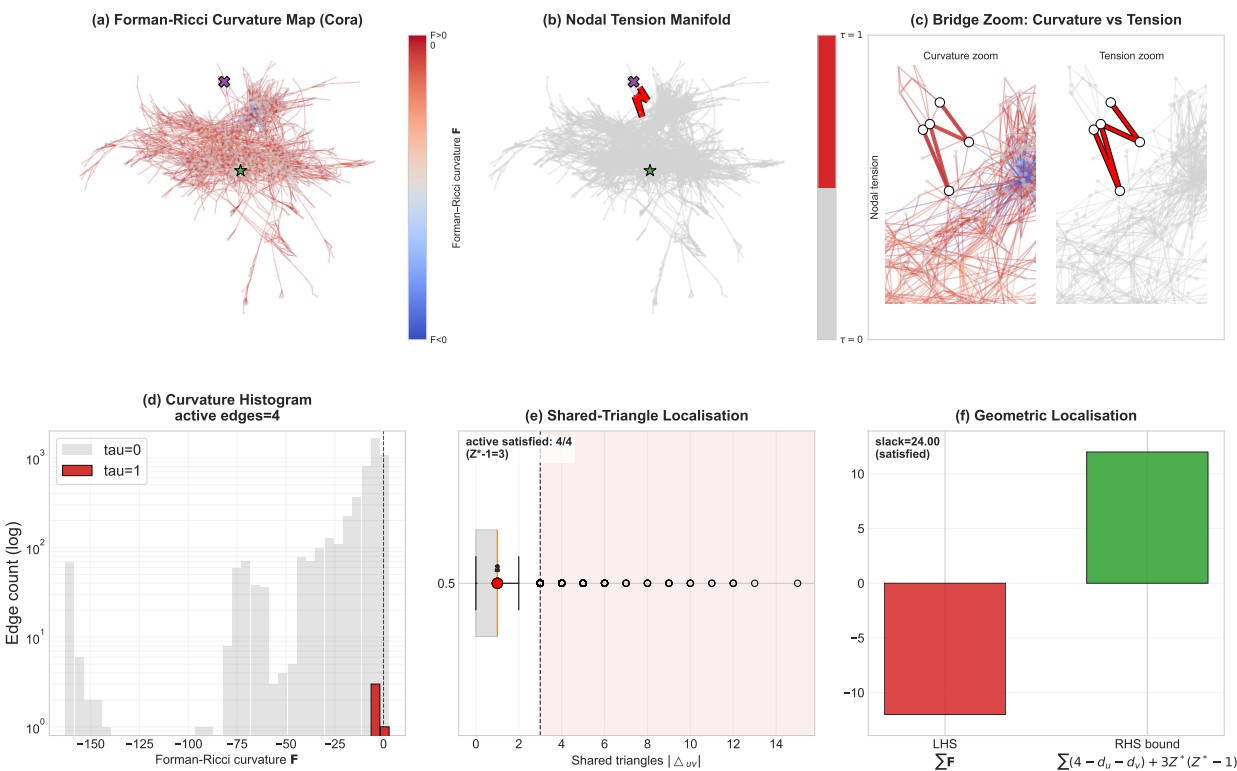

Figure 17: Empirical validation of geometric filtering on the Cora citation network (Sen et al., 2008). **(a)** The discrete Forman-Ricci curvature map exhibits severe diagnostic ambiguity, flagging thousands of edges with negative curvature across the network. **(b)** The continuous Nodal Tension manifold filters this geometric noise, isolating the exact $s - t$ structural bottleneck. **(c)** A microscopic zoom of the separating manifold, contrasting the local curvature noise against the binary tension gradient. **(d)** Edge count distribution (log scale) by Forman-Ricci curvature. The LP assigns zero tension ($\tau = 0$) to thousands of spurious edges exhibiting extreme negative curvature (down to $\mathbf{F} \approx -150$), physically validating Corollary 4.7. **(e)** Validation of Lemma 4.4; all 4 active cut edges satisfy the shared-triangle upper bound of $|\triangle_{uv}| \leq Z_{s,t}^* - 1$ (where $Z_{s,t}^* - 1 = 3$). **(f)** Validation of Theorem 4.6 (Geometric Localisation); the aggregate discrete curvature of the active manifold (LHS) is bounded by the theoretical limit derived from the global capacity and local degrees (RHS), satisfying the inequality with a slack of 24.00.

the continuous $L_1$ bottleneck capacity ($Z_{s,t}^*$) dictates the layer depth ($K$) at which gradients vanish, and to evaluate how standard topological interventions interact with these geometric limits.

**The MPNN Phase Transition.** We first created a tunable topological bottleneck consisting of two uniform cliques (a source and target component) connected by the specified $Z_{s,t}^*$ bridging edges. By incrementally adjusting this specific $L_1$ capacity while keeping the total graph volume ($\text{vol}(V)$) approximately constant, we simulated the $K$-step transition probability matrix of a standard linear MPNN and extracted the target sensitivity ($P^K)_{ts}$.

As visualised in Figure 18, the empirical forward pass is governed by the expected hitting time bounds derived in Theorem 4.10. The spatial distributions (Figure 18a–c) demonstrate that a tight capacity ($Z_{s,t}^* = 1$) physically traps the probability mass within the source component, whereas an expanded capacity ($Z_{s,t}^* = 5$) permits transmission. Crucially, the analytical boundary $K = \frac{\text{vol}(V)}{2Z_{s,t}^*}$ correctly delineates the hyperparameter threshold at which the sensitivity completes its rapid accumulation and enters a traversable state (Figure 18d, e). For any fixed network depth (Figure 18f), the structural survival of the gradient is dependent on satisfying this local isoperimetric constraint.

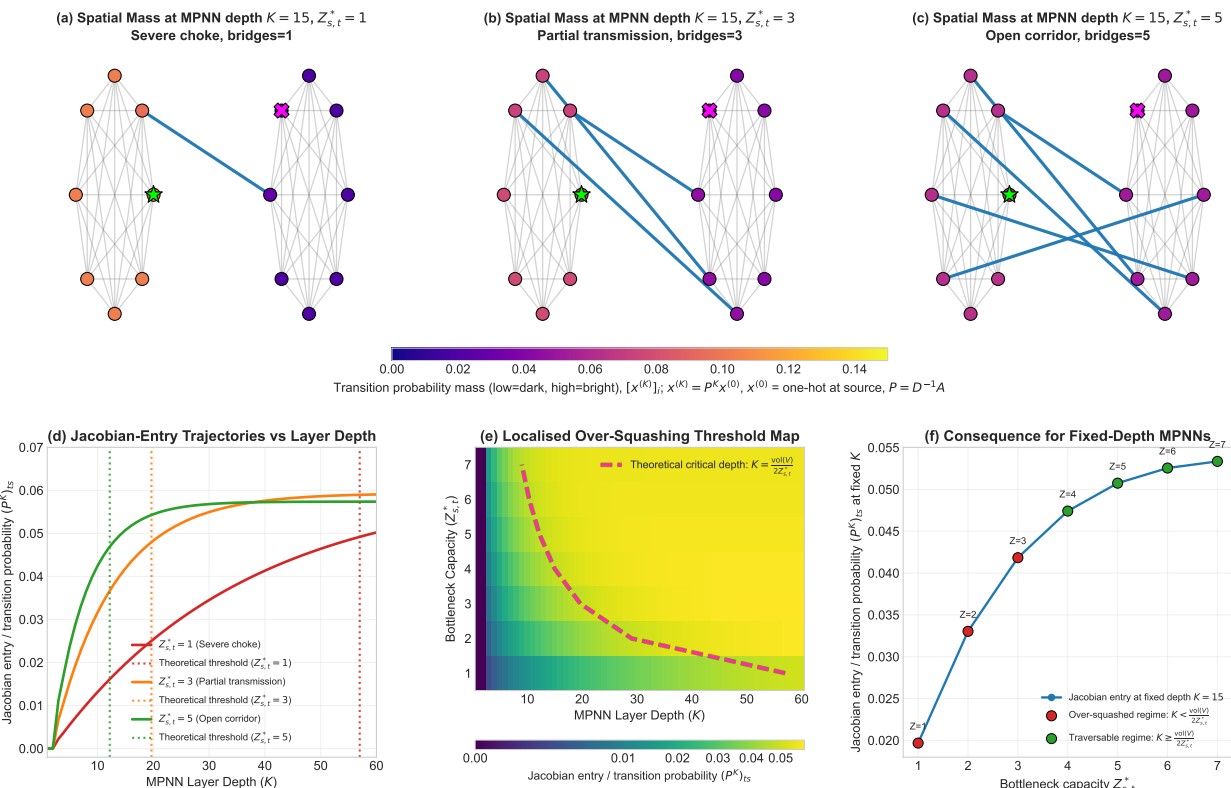

Figure 18: Empirical validation of localised over-squashing (Corollary 4.11) governed by the local bottleneck capacity $(Z_{s,t}^*)$. **(a–c)** Spatial distribution of the transition probability mass $(P^K)$ at a fixed layer depth of $K = 15$. A severe structural choke $(Z_{s,t}^* = 1)$ traps the signal mass within the source component, whereas an open corridor $(Z_{s,t}^* = 5)$ permits transmission, illustrating the expected hitting time restrictions derived in Theorem 4.10. **(d)** Layer-wise trajectories of the Jacobian sensitivity $(P^K)_{ts}$. The sensitivity accumulates rapidly during the initial forward pass and gradually levels off once the MPNN depth $K$ reaches the theoretical expected hitting time threshold $(\frac{\text{vol}(V)}{2Z_{s,t}^*})$. A restricted capacity severely reduces this growth rate and delays the threshold, leaving the network structurally constrained at earlier depths. **(e)** A 2D heatmap mapping sensitivity across varying layer depths and bottleneck capacities. The theoretical expected depth bound (dashed curve) traces the geometric boundary where the sensitivity completes its rapid growth and enters the saturated traversable regime. **(f)** The consequence for fixed-depth architectures (e.g., $K = 15$). For a static hyperparameter $K$, the magnitude of the Jacobian entry is directly dictated by the local capacity. If the $s - t$ capacity is too low, the slow growth rate forces the sensitivity toward zero, demonstrating the necessity of local Nodal Conductance over global spectral metrics (Lemma 4.9).

**The Rewiring Paradox.** To investigate the practical implications of these bounds for graph topology modification, we evaluated a "rewiring paradox" scenario. We constructed a base topology comprising three sequential clusters (Figure 19a), embedding a target $s - t$ bottleneck locally while positioning the absolute global minimum cut elsewhere. We then applied two distinct edge-addition strategies, holding the total number of added edges constant (+10 edges): a heuristic intervention contained within the dense source cluster (Figure 19b), and a surgical intervention using Nodal Tension to cross the $s - t$ minimal surface $C^*$ (Figure 19c).

The empirical results in Figure 19 validate the limits of Corollary 4.11. The heuristic rewiring, despite increasing local density, degrades the Jacobian sensitivity relative to the base graph (Figure 19d). By Theorem 4.10, injecting edges internally increases the total graph volume in the numerator $(\text{vol}(V))$ without expanding the bridging capacity in the denominator $(Z_{s,t}^*)$. This raises the required expected depth

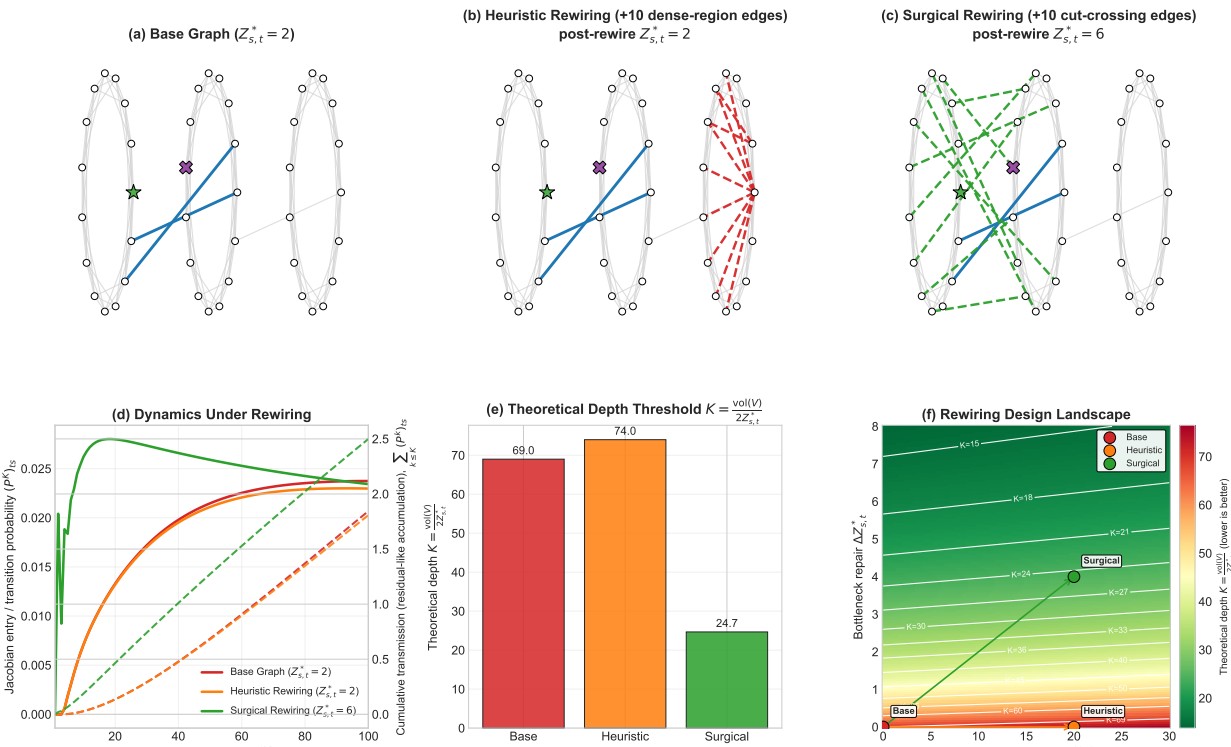

Figure 19: Empirical validation of the graph rewiring paradox, governed by the local bottleneck bounds established in Theorem 4.10 and Corollary 4.11. **(a)** Base graph topology exhibiting a moderate structural bottleneck ($Z_{s,t}^* = 2$). **(b)** Heuristic rewiring injects 10 edges within dense regions. This intervention increases the total graph volume $\mathrm{vol}(V)$ while leaving the local $s - t$ capacity unchanged. **(c)** Surgical rewiring uses Nodal Tension to add 10 edges across the isolated minimal surface $C^*$, explicitly expanding the bottleneck capacity ($Z_{s,t}^* = 6$). **(d)** MPNN forward-pass dynamics. Heuristically adding edges degrades the Jacobian sensitivity relative to the base graph, physically acting as a probability sink that delays signal propagation. **(e)** Evaluation of the theoretical depth threshold ($K = \frac{\mathrm{vol}(V)}{2Z_{s,t}^*}$). The heuristic volume injection raises the required traversal depth, correctly predicting the suppressed sensitivity observed in panel (d). **(f)** The rewiring design landscape mapping volume injection against bottleneck repair. The isoclines represent the theoretical expected depth threshold. This confirms that topological interventions must improve the local Nodal Conductance (Lemma 4.9) rather than solely increasing graph density; otherwise, the added edges will exacerbate localised over-squashing.

threshold, causing the added edges to act as a probability sink that traps the random walk and exacerbates over-squashing (Figure 19e). Conversely, surgical rewiring explicitly expands the bridging capacity in the denominator, successfully restoring gradient flow. The rewiring design landscape (Figure 19f) maps this trade-off, plotting the theoretical expected depth threshold as continuous contour lines over the intervention space. Because adding any edge inevitably increases the total graph volume, topological interventions must yield a disproportionate improvement to the local $s-t$ capacity to cross into a lower depth isocline. This validates the necessity of local Nodal Conductance (Lemma 4.9); algorithms that merely inject density without targeting the exact $L_1$ minimal surface $C^*$ will traverse the wrong contour and exacerbate over-squashing.

Because the local $L_1$ capacity ($Z_{s,t}^*$) provides the explicit threshold distinguishing successful signal transmission from geometric gradient extinction, identifying and optimising this capacity becomes a crucial objective for robust graph representation learning. Having established both the static precision and dynamical necessity of Nodal Tension, we now transition from discrete structural analysis to continuous sensitivity analysis. As established in Section 4.4, evaluating the polyhedral differentiability and Lipschitz continuity of the LP is necessary to integrate this minimal surface metric into end-to-end differentiable optimisation pipelines.

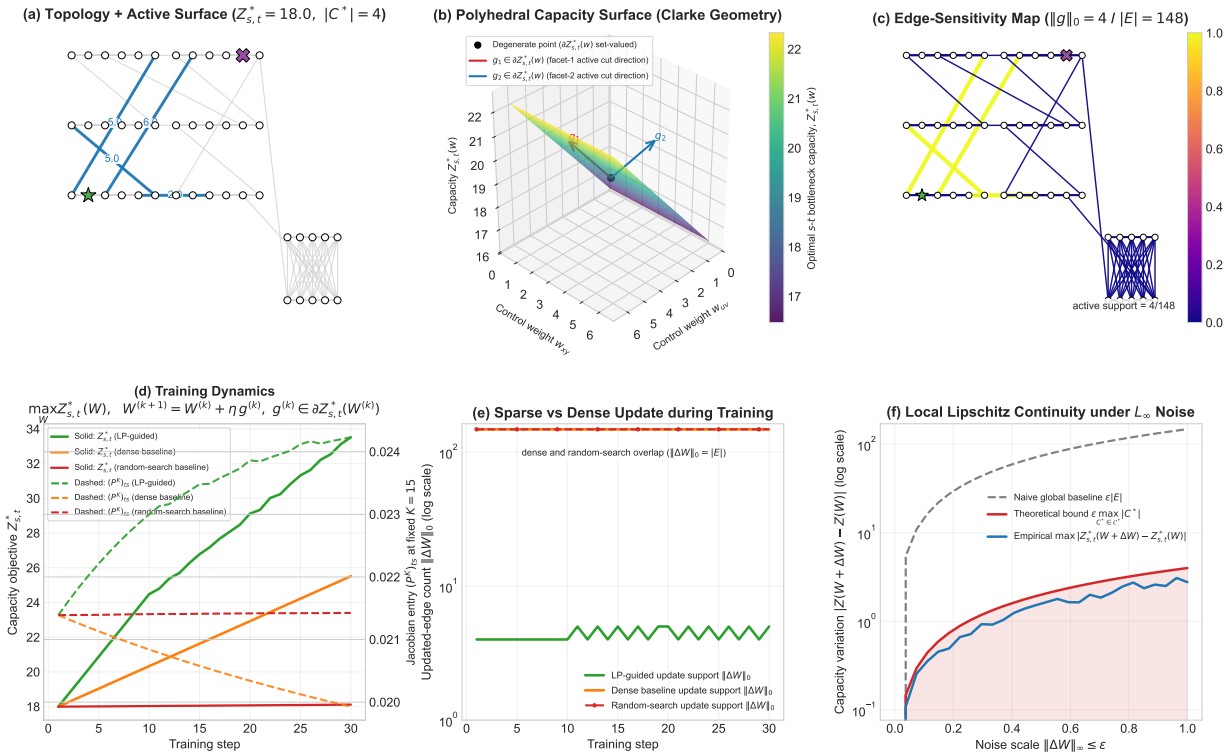

Figure 20: Empirical validation of the polyhedral differentiability, sparse backpropagation, and local Lipschitz robustness of the continuous Nodal Tension objective. **(a)** The evaluation topology, highlighting the active minimal surface ($|C^*| = 4$) that structurally restricts $s - t$ flow. **(b)** The continuous bottleneck capacity evaluated across two parameterised edge weights. The objective forms a globally concave, piecewise linear surface. At points of topological degeneracy (the ridge), the Clarke subdifferential evaluates to the set-valued convex hull of the active cut directions (Theorem 4.13). **(c)** The edge-level sensitivity map. The subgradient is topologically sparse ($\|g\|_0 = |C^*|$), allocating non-zero evaluations exclusively to the active bottleneck (Corollary 4.14). **(d)** Training dynamics comparing LP-guided gradient ascent against dense and random-search baselines. The exact topological gradients efficiently maximise the bottleneck capacity, directly restoring the MPNN Jacobian transmission. **(e)** The updated-edge count during training. The LP-guided backward pass restricts parameter updates to the sparse active support, whereas standard baselines densely perturb the entire graph geometry. **(f)** Evaluation of local Lipschitz continuity under adversarial $L_\infty$ noise. The empirical capacity variation respects the tight theoretical bounding envelope scaled by the maximal cut cardinality (Corollary 4.15), confirming that the continuous objective is structurally insulated from uniform perturbations occurring outside the minimal surface.

## 5.6 End-to-End Differentiable Bottleneck Optimisation

To empirically validate the polyhedral differentiability and Lipschitz robustness of the continuous LP derived in Section 4.4, we integrated the exact Clarke subgradient extraction into a continuous evaluation and training loop. Our objective is to demonstrate that the $L_1$ bottleneck capacity ($Z_{s,t}^*$) produces a navigable, piecewise linear loss surface with sparse gradients, and to verify that its empirical capacity variation ($\Delta Z_{s,t}^*$) respects the theoretical local Lipschitz bounding envelope under adversarial $L_\infty$ perturbations.

**Polyhedral Geometry and Sparse Backpropagation.** We created a structurally complex multi-community graph featuring parallel parameterised $s - t$ bottlenecks and a dense distractor appendage (Figure 20a). This topology ensures a high global edge count ($|E|$) while effectively constraining the active minimal surface ($|C^*|$).

As visualised in Figure 20b, mapping the continuous Nodal Tension capacity across the parameterised bottleneck weights reveals a globally concave, piecewise linear surface. At points of topological degeneracy (the geometric ridge), the Clarke subdifferential natively evaluates the set-valued intersection of the active cut directions, confirming the unbroken polyhedral geometry established in Theorem 4.13. Furthermore, evaluating this subgradient over the entire network produces a sparse edge-sensitivity map (Figure 20c). The gradient isolates the minimal surface ($\|g\|_0 = |C^*|$), allocating zero sensitivity to the vast majority of the dense graph, directly validating the sparse backpropagation bounds of Corollary 4.14.

**Training Dynamics and Adversarial Robustness.** To test this differentiability in practice, we deployed the subgradient extraction within a gradient ascent loop tasked with alleviating the structural $s-t$ bottleneck. For the LP-guided optimiser, the continuous edge weights were updated via $W^{(k+1)} = W^{(k)} + \eta g^{(k)}$, where $\eta$ is the learning rate and $g^{(k)} \in \partial Z^*_{s,t}(W^{(k)})$ is the sparse Clarke subgradient. We compared this analytical backpropagation against two standard algorithmic baselines: a dense heuristic update and a zeroth-order random search. The dense baseline evaluates a global update direction $h \in \mathbb{R}^{|E|}$ proportional to the product of incident node degrees ($h_{uv} \propto d_u d_v$), applying the normalised update $W^{(k+1)} = W^{(k)} + \eta h$. This represents a standard topological assumption that uniformly augmenting high-degree hubs improves flow. The random-search baseline acts as a zeroth-order empirical optimiser; at each step, it samples multiple isotropic perturbations $\delta \in \mathbb{R}^{|E|}$ from a standard normal distribution, scales them to the step size, and greedily applies the specific perturbation that yields the maximal empirical capacity gain, $Z^*_{s,t}(W^{(k)} + \delta) - Z^*_{s,t}(W^{(k)})$.

As shown in Figure 20e, the LP-guided optimiser restricted its parameter updates entirely to the sparse active support ($\|\Delta W\|_0 = |C^*|$), whereas the standard dense and random-search baselines naively perturbed the entire network geometry ($\|\Delta W\|_0 = |E|$). By surgically targeting the active cut, the continuous LP efficiently maximised the objective capacity (Figure 20d), which consequently restored the MPNN Jacobian transmission ($P^K)_{ts}$. This confirms that the dynamical principles of localised over-squashing (Corollary 4.11) can be resolved using end-to-end differentiable structural updates without requiring computationally expensive or noisy dense graph modifications.

Finally, we evaluated the metric's stability under dense uniform noise by subjecting all $|E|$ edges to adversarial $L_\infty$ perturbations. Figure 20f plots the maximum absolute capacity variation against the noise scale $\epsilon$. While a naive global bound scales vulnerably with the total edge count ($\epsilon|E|$), the true empirical variation respects the tight theoretical envelope scaled by the maximal active cut cardinality ($\epsilon \max |C^*|$). This validates Corollary 4.15; the continuous objective structurally filters out dense noise occurring outside the minimal surface, ensuring robust optimisation in highly irregular or uncertain topologies.

### 5.7 Real-World Case Study

Having validated the model's correctness and theoretical properties on synthetic and benchmark topologies (Sections 5.3–5.6), we now apply it to a large-scale empirical network: identifying critical choke points in a habitat corridor for grizzly bears (*Ursus arctos horribilis*) in the Canadian Rocky Mountains (Proctor et al., 2015). This case study serves to demonstrate how the geometric bounds, expected commute time limits, and Clarke subgradients derived in Section 3 and 4 translate into physical ecological diagnostics (McRae et al., 2012), addressing the limitations of standard average-case flow models (McRae et al., 2008). The primary goal is to assess the resilience of the connection between two major protected ecosystems and to contrast our targeted $L_1$ bottleneck analysis against both global spectral methods and $L_2$ effective resistance, validating the practical necessity of local network bounds in complex empirical systems.

#### 5.7.1 Data Acquisition and Network Construction

To model the landscape, we required two primary types of geospatial data: a habitat resistance surface and the boundaries of core protected areas to serve as our source ($s$) and target ($t$) nodes. All data were acquired from publicly available, authoritative sources.

**Habitat Resistance Surface.** The foundation of our analysis is a raster surface where each pixel's value represents the "cost" or "resistance" for a grizzly bear to traverse that area. We derived this surface from the Grizzly Bear Habitat Capability Ratings dataset provided by the Government of British Columbia (available

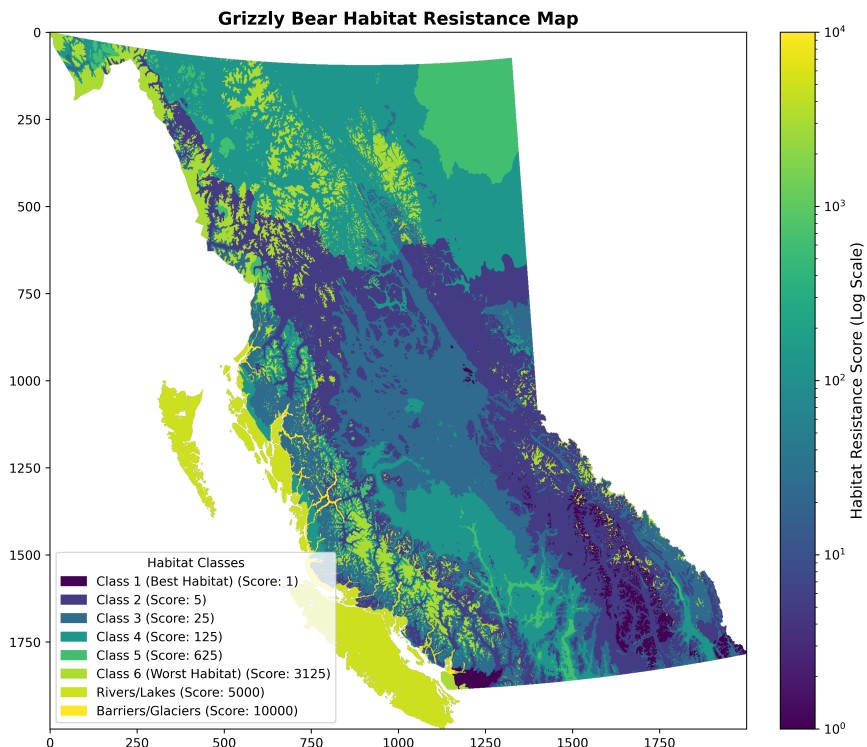

Figure 21: Habitat resistance surface (original resolution) for the grizzly bear case study. The resistance values are derived from the Grizzly Bear Habitat Capability Ratings for British Columbia. Habitat classes were converted to numerical resistance scores on an exponential scale, where Class 1 corresponds to a resistance of 1, and barriers correspond to a resistance of 10,000. The map is visualised on a logarithmic colour scale, where dark blue areas represent low-cost habitat, and bright yellow areas represent high-cost barriers. This surface forms the basis for the weighted network graph.

via the B.C. Data Catalogue[1]). The original vector data was rasterised to a $2000 \times 2000$ grid (Figure 21), where habitat classes were converted to numerical resistance scores on an exponential scale, matching established least-cost modelling practices (Spear et al., 2010). Specifically, capability classes 1 through 6 were mapped to resistance values $1, 5, 25, 125, 625,$ and $3,125,$ respectively, water bodies to $5,000,$ and impassable barriers such as glaciers and bare rock correspond to a resistance of 10,000.

**Source and Target Nodes.** The selection of appropriate source and target nodes is critical for a meaningful connectivity analysis. For this study, we chose two key protected areas within the internationally recognised Yellowstone to Yukon (Y2Y) corridor, a region identified as a global priority for large carnivore conservation (Proctor et al., 2012). All protected area boundaries were acquired from the World Database on Protected Areas (WDPA) managed by UNEP-WCMC and IUCN.

Our source ($s$) is the **Akamina-Kishinena Provincial Park** (WDPA ID: 21193[2]). This park, though modest in size, is of exceptional strategic importance. It is situated directly on the Canada-US border, forming a critical transboundary linkage zone with Glacier National Park in Montana (Proctor et al., 2005). This area, often referred to as the Crown of the Continent, serves as a vital stepping-stone habitat that connects the grizzly bear populations of the northern United States with the larger populations in Canada. Its selection as the source node allows us to model the resilience of the connection from this crucial, and potentially vulnerable, international gateway.

---

[1]https://catalogue.data.gov.bc.ca/dataset/dba6c78a-1bc1-4d4f-b75c-96b5b0e7fd30
[2]https://www.protectedplanet.net/21193

Table 4: Theoretical Diagnostics Dashboard: Quantitative validation of our $L_1$ Nodal Tension framework on the Grizzly Bear habitat corridor. By computing the mathematical variables derived in Section 3 and 4 on the 18,183-node empirical landscape (Figure 21), this dashboard translates LP and graph bounds into applied ecological diagnostics. The redundancy gap ($\Delta_{s,t}$) classifies the Rocky Mountains as a diffuse bottleneck lacking alternative pathways (McRae et al., 2008), while the extreme expected commute time bound ($H(s,t)$) validates migration isolation. Crucially, the Clarke subgradient ($\|g\|_0$) isolates the absolute weakest link down to just 2 specific edges out of 35,856, providing a hyper-sparse, effective blueprint for spatial conservation prioritisation (McRae et al., 2012).

| Metric | Value | Theoretical Link | Ecological Interpretation |
|---|---|---|---|
| **Macroscopic Graph Properties** | | | |
| Graph Size (Nodes) | 18,183 | – | Landscape spatial extent |
| Graph Size (Edges) | 35,856 | – | Allowed movement transitions |
| Algebraic Connectivity ($\lambda_2$) | 0.0217 | Global Cheeger (Thm 4.8) | Overall landscape permeability |
| **$L_1$ Nodal Tension Outputs (Weakest Link)** | | | |
| Resilience Score ($Z_{s,t}^*$) | 3.0650 | Min-cut Capacity (Thm 3.1) | Severity of the absolute structural "pinch point" (Proctor et al., 2015) |
| Cut Balance ($\min(\text{vol}(V_s), \text{vol}(V_t))$) | 3.0650 | Denominator in Thm 4.10 | Indicates a highly asymmetric, perimeter-level severance |
| **Redundancy & Local Isoperimetry ($L_2$ vs $L_1$)** | | | |
| $L_2$ Effective Conductance ($C_{s,t}$) | 1.8004 | Capacity Bound (Lem 4.1) | Average-case flow restricted by landscape resistance (McRae et al., 2008) |
| Redundancy Gap ($\Delta_{s,t}$) | 1.2646 | Bottleneck Typology (Cor 4.2) | Identifies a diffuse barrier lacking alternative pathways (McRae et al., 2008) |
| Local Nodal Conductance ($\Phi_{s,t}^*$) | 1.0000 | $\Phi_{s,t}^* \geq h_G$ (Lem 4.9) | Localised permeability of the specific migration corridor (Rosenberg et al., 1997; Chetkiewicz et al., 2006) |
| **Geometric & Dynamical Limits** | | | |
| Aggregate Cut Curvature ($\sum_{(u,v) \in C^*} \mathbf{F}(u,v)$) | −1 | Localisation (Thm 4.6) | Deepest topological valley restricting biological movement (Adriaensen et al., 2003; Urban & Keitt, 2001) |
| Expected Commute Bound ($H(s,t)$) | $1.58 \times 10^7$ | Hitting Time (Thm 4.10) | Severe dispersal delay causing functional isolation (Ricketts, 2001; Proctor et al., 2012) |
| **Differentiable Optimisation (Sparse Backprop)** | | | |
| Clarke Support ($\|g\|_0 = |C^*|$) | 2 edges | Sparsity (Cor 4.14) | Targeted blueprint identifying where to restore connectivity (McRae et al., 2012) |

Our target ($t$) is the **Purcell Wilderness Conservancy Provincial Park** (WDPA ID: 167308[3]). This is a vast, remote, and ecologically intact core habitat area located further north in the Purcell Mountains. The Purcell range, along with the adjacent Selkirk mountains, contains one of the most important and threatened grizzly bear populations in southern Canada (Proctor et al., 2005). By selecting this large, stable core habitat as our target, we can effectively model the challenge of maintaining connectivity from the transboundary linkage zone to the heart of the Canadian grizzly bear range. The corridor between these two specific areas represents a key pathway for maintaining the long-term genetic health and demographic stability of grizzly bear populations across this critical portion of the Y2Y ecoregion.

**Network Construction.** We converted this geospatial data into a weighted, undirected graph. A grid was overlaid on the resistance raster, with each grid cell corresponding to a node in the graph. Nodes falling within impassable barriers (resistance = 10,000) were excluded. Edges were added between adjacent nodes, with the edge weight calculated as the average resistance of the two connected nodes. This process resulted in a graph with several disconnected components; we selected the largest connected component for our analysis to ensure all subsequent algorithms would run correctly. This final graph, consisting of 18,183 nodes and 35,856 edges, represents the contiguous habitat network available to grizzly bears in the region.

---

[3]https://www.protectedplanet.net/167308

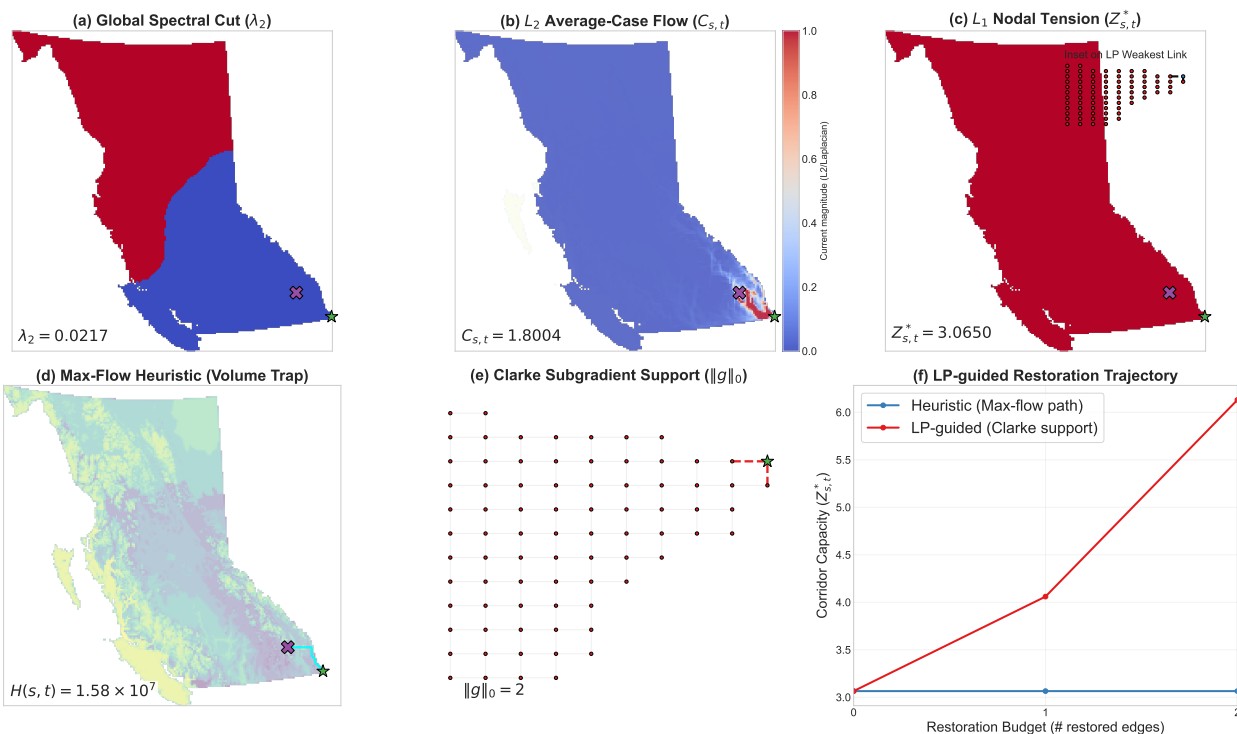

Figure 22: Analysis of the grizzly bear habitat corridor between the Akamina-Kishinena Park (source) and the Purcell Wilderness Conservancy (target). **(a)** The global spectral cut ($\lambda_2$) partitions the landscape based on total structural volume, failing to isolate the specific pathway restriction (Rosenberg et al., 1997) (Theorem 4.8). **(b)** The continuous $L_2$ effective conductance (McRae et al., 2008) reveals a diffuse bottleneck; average-case flow is choked by prolonged series resistance across the mountains (Lemma 4.1). **(c)** The continuous $L_1$ Nodal Tension explicitly isolates the absolute weakest link ($Z_{s,t}^*$), identifying a severe perimeter-level severance (Proctor et al., 2015) (Theorem 3.1). **(d)** The max-flow optimal path identifies a deep interior corridor. Due to the boundary severance in (c), protecting this interior path acts as a volume trap; increasing interior density without expanding the boundary capacity worsens the expected commute time bound ($H_{\text{bound}}$), driving functional isolation (Proctor et al., 2012) (Theorem 4.10). **(e)** The Clarke subgradient ($\|g\|_0$) provides a hyper-sparse, localised evaluation of structural sensitivity (Corollary 4.14). **(f)** Simulated habitat restoration trajectories. Heuristic intervention along the max-flow path fails to improve the local $s-t$ capacity. Conversely, differentiable optimisation using the Clarke subgradients directly targets the minimal surface, maximising conservation return-on-investment (McRae et al., 2012).

### 5.7.2    Results: The Weakest Link vs. the Best Path

Our analysis reveals a structurally counter-intuitive feature of this landscape's connectivity, best understood through the lens of the max-flow min-cut duality and the geometric bounds derived in Section 4. The two analyses shown in Figure 22—the continuous $L_1$ minimum cut partition from our Nodal Tension model and the maximum flow path—are not independent results; they are the primal and dual solutions to the same optimisation problem. One identifies the network's absolute weakest link, while the other identifies its optimal routing path. The key finding of this case study is that for this transboundary landscape, these two critical features are *not* spatially collocated.

Figure 22(d) shows the maximum flow path, a single, clear interior corridor representing the route of least cumulative resistance between the two protected areas. In spatial ecology, interventions frequently target these optimal interior routes (Rosenberg et al., 1997). In contrast, Figure 22(c) shows the $L_1$ minimum cut. It is not located along this optimal interior path, but is instead a highly unbalanced partition that severs the local perimeter of the source node from its immediate neighbours. The quantitative results in Table 4 confirm

this: the Resilience Score $(Z_{s,t}^*)$ is exceptionally low (3.0650), and the Cut Balance $(\min(\text{vol}(V_s), \text{vol}(V_t)))$ evaluates to the exact same value (3.0650). This algebraic equivalence proves that there are zero internal edges within the isolated boundary component.

This divergence between the interior path and the perimeter weakest link serves as a *null signal* for the existence of a single, remote choke point within the Rocky Mountains. We formalise this by evaluating the redundancy gap (Lemma 4.1). The continuous $L_2$ effective conductance $(C_{s,t} = 1.8004$, Figure 22b) is choked by prolonged series resistance across the rugged terrain, yielding a large gap $(\Delta_{s,t} = 1.2646)$. This restricts average-case flow and classifies the corridor as a diffuse bottleneck lacking alternative pathways (McRae et al., 2008) (Corollary 4.2). The landscape is so saturated with interior resistance that the cost to sever the optimal corridor remotely is still higher than the cost of a local severance at the source. Consequently, the targeted $s - t$ signal experiences extreme functional isolation; the highly restricted $L_1$ capacity dictates a severe expected commute time bound $(H(s, t) = 1.58 \times 10^7$ steps), predicting the demographic fragmentation observed in empirical transboundary grizzly populations (Proctor et al., 2012) (Theorem 4.10).

This finding stands in contrast to the output of the global spectral method (Figure 22(a)), which identifies a balanced, large-scale structural division $(\lambda_2 = 0.0217)$ that is irrelevant to the specific $s - t$ corridor. It also exposes the functional limits of standard conservation heuristics. As demonstrated in Figure 22(f), restoring habitat exclusively along the interior max-flow path acts as a "volume trap"; it increases interior network density without expanding the restricted boundary capacity, failing to improve the $s - t$ resilience. Instead, extracting the Clarke subgradient $(\|g\|_0 = 2$, Figure 22e) isolates the specific 2 segments on the source perimeter dictating the capacity (Corollary 4.14). This suggests that for certain diffuse landscapes, ecological conservation management efforts should prioritise the creation of robust buffer zones around existing parks over the protection of interior corridors. Our differentiable Nodal Tension model provides a formal, quantitative blueprint for identifying these spatial limits, maximising conservation return-on-investment (McRae et al., 2012).

# 6 Discussion and Conclusion

In this work, we introduced Nodal Tension, a linear programming framework for quantifying the local resilience of connections in a network. We proved its theoretical properties, including its duality with the maximum flow problem and its relationship to global spectral bounds, and established a theoretical core linking local $L_1$ capacity to geometric curvature, expected commute times, and polyhedral differentiability for optimisation (Section 4). We then demonstrated its practical utility and scalability in a real-world case study on grizzly bear habitat connectivity.

## 6.1 Implications for Scientific Discovery

The primary empirical result of our work is the quantitative ecological insight that emerged from the grizzly bear case study. The prevailing paradigm in conservation connectivity has long focused on the identification and protection of singular, critical corridors or choke points that link core habitat areas (Rosenberg et al., 1997; Hilty et al., 2020). Methods such as least-cost path analysis and circuit theory ($L_2$ effective resistance) are often employed to highlight these specific routes. Our analysis, however, presents a mathematical counterpoint.

Our Nodal Tension model, by solving for the minimum $s - t$ cut, revealed that the most vulnerable point in the connection between our two protected areas was not a remote bottleneck, but the local perimeter of the source park itself. This "trivial" cut is, in fact, a structural finding: it represents a null signal for the existence of a single, well-defined interior choke point. By formalising the redundancy gap (Lemma 4.1), we demonstrated that the landscape is saturated with diffuse series resistance. This acts as a volume trap that worsens the expected commute time for dispersing species (Theorem 4.10), proving that the cost to sever the best path remotely is still higher than the cost of a local severance at the source.

This result does not suggest that corridors are unimportant. Rather, it provides a quantitative method for identifying *when* and *where* the concept of a single corridor is the appropriate model. For vast, complex landscapes with high degrees of redundancy, our findings suggest that the conservation paradigm may need to

shift. Instead of relying on heuristic interventions that fail to improve worst-case capacity, practitioners can utilise the Clarke subgradients (Corollary 4.14) derived from our continuous formulation to surgically target optimal restoration sites (McRae et al., 2012). This implies that strategies focused on creating robust buffer zones and mitigating threats at the park boundaries are frequently the mathematically optimal interventions for maintaining the resilience of the entire network.

## 6.2 Limitations and Future Work

We acknowledge several limitations of our empirical application that provide avenues for future research.

**Static Landscape Model.** Our analysis is based on a static habitat resistance surface. Real-world landscapes are dynamic; resource availability, snowpack, and human activity patterns change seasonally and annually. Future work should aim to incorporate dynamic or time-varying edge weights into the network model, moving from a static graph to a temporal network representation (Holme & Saramäki, 2012). This would allow for an analysis of how corridor resilience changes over time and could identify critical temporal bottlenecks (e.g., a corridor that is only viable in the summer).

**Single-Species Focus.** The resistance surface and analysis were tailored specifically to grizzly bears. However, ecosystems are multi-species systems. A landscape that is permeable for a grizzly bear may be a barrier for a pronghorn or a wolverine. An extension of this work would be to develop a multi-commodity flow version of the Nodal Tension model, a well-established generalisation of the standard max-flow problem (Salimifard & Bigharaz, 2022). In this formulation, each "commodity" would represent a different species with its own unique resistance map, allowing for a more holistic analysis of ecosystem-level resilience.

**Graph Model Abstraction.** The conversion of a continuous landscape into a discrete, grid-based graph is a necessary abstraction with theoretical implications. Our empirical case study modelled the landscape as an undirected graph with isotropic resistance, meaning that the cost of traversing an edge is the same in both directions. However, real-world movement is often anisotropic; for example, it is more energetically costly for a bear to move uphill against a steep elevation gradient than downhill. Because the total unimodularity of the constraint matrix holds natively for directed node-edge incidence matrices (Section 3), the Nodal Tension framework inherently supports directed graphs with asymmetric edge weights. Future empirical applications should incorporate such anisotropic effects by modelling the landscape as a directed graph. This approach is well-established in landscape ecology for creating more physically realistic cost surfaces (Van Etten, 2017), and it would provide a more nuanced basis for the resilience calculation.

## 6.3 Conclusion

This paper introduced the Nodal Tension model, a continuous optimisation framing for local network resilience. We demonstrated that this formulation is the linear programming dual of the maximum flow problem, thereby inheriting the theoretical guarantees of the max-flow min-cut theorem. Building on this foundation, we derived a comprehensive geometric toolkit, proving strict relationships between local $L_1$ bottlenecks, structural redundancy, discrete curvature, and polyhedral subgradient sparsity. Our theoretical and empirical experiments confirmed the model's correctness and explored its advantages over both global spectral measures and average-case circuit theory.

The practical contribution of this work is the translation of these mathematical bounds into actionable, physical diagnostics, derived from the application of our local resilience measure to a real-world problem. Our case study on grizzly bear habitat connectivity did not simply identify a corridor; it demonstrated that for some complex landscapes, the very concept of a single, remote choke point may not apply. By revealing a diffuse bottleneck and proving that the weakest link was the local perimeter of the protected area, our work provides a quantitative, highly localised and targeted understanding of network vulnerability. It offers network analysts and environmental conservation planners a differentiable computational tool to determine when to focus on protecting singular corridors, and when to prioritise the creation of robust buffer zones at a habitat's own doorstep.

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

# A    Detailed Proofs of Fundamental Properties in Section 3.3

In this appendix, we provide the detailed proofs for the fundamental LP properties presented in Section 3.3. These proofs supply the operations research arguments that underpin our Nodal Tension formulation, including the equivalence of our model to the minimum $s - t$ cut, the integrality of the optimal potentials, and the derivation of the global spectral bound discussed at the end of Section 3.3.

## A.1    Proof of Theorem 3.1

**Theorem** (Restatement of Theorem 3.1). *Let $G = (V, E, w)$ be a weighted, undirected graph with edge weights $w_{ij} > 0$ interpreted as capacities. For any two distinct nodes $s, t \in V$, the optimal value, $Z_{s,t}^*$, of the Nodal Tension LP (Equations 1-5) is exactly equal to the capacity of the minimum weighted $s - t$ cut.*

*Proof.* As noted in the main text, this result is a direct consequence of standard network flow duality. The proof is presented in two parts. First, we show that the optimal LP value is less than or equal to the capacity of the minimum $s - t$ cut by constructing a feasible solution. Second, we show the reverse inequality by analysing the dual of the LP and invoking the max-flow min-cut theorem.

**Part 1:** $Z_{s,t}^* \leq \textbf{min-cut}(s, t)$

Let $(S^*, V \setminus S^*)$ be a minimum $s - t$ cut of the graph $G$, where $s \in S^*$ and $t \in V \setminus S^*$. Let $C^*$ be the set of edges crossing this partition, i.e., $C^* = \{(i, j) \in E \mid i \in S^*, j \in V \setminus S^* \text{ or } j \in S^*, i \in V \setminus S^*\}$. The capacity of this minimum cut is min-cut$(s, t) = \sum_{(i,j) \in C^*} w_{ij}$.

We construct a specific, feasible solution $(\hat{p}, \hat{d})$ for the Nodal Tension LP as follows:

- For each node $i \in V$, set the potential $\hat{p}_i = 0$ if $i \in S^*$ and $\hat{p}_i = 1$ if $i \in V \setminus S^*$.

- For each edge $(i, j) \in E$, set the tension $\hat{d}_{ij} = |\hat{p}_i - \hat{p}_j|$.

This solution is feasible. The separation constraints are met, as $\hat{p}_s = 0$ and $\hat{p}_t = 1$. The tension constraints are met by definition, since $\hat{d}_{ij} = |\hat{p}_i - \hat{p}_j|$ implies $\hat{d}_{ij} \geq \hat{p}_i - \hat{p}_j$ and $\hat{d}_{ij} \geq \hat{p}_j - \hat{p}_i$.

Now, we calculate the objective value for this feasible solution:

$$Z(\hat{p}, \hat{d}) = \sum_{(i,j) \in E} w_{ij} \hat{d}_{ij} = \sum_{(i,j) \in E} w_{ij} |\hat{p}_i - \hat{p}_j| \tag{39}$$

The term $|\hat{p}_i - \hat{p}_j|$ is non-zero only if the edge $(i, j)$ crosses the cut $(S^*, V \setminus S^*)$ (i.e., $(i, j) \in C^*$), in which case its value is $|0 - 1| = 1$. If the edge does not cross the cut, both endpoints are in the same partition, and $|\hat{p}_i - \hat{p}_j|$ is either $|0 - 0| = 0$ or $|1 - 1| = 0$. Therefore, the sum reduces to:

$$Z(\hat{p}, \hat{d}) = \sum_{(i,j) \in C^*} w_{ij} \cdot 1 = \text{min-cut}(s, t) \tag{40}$$

We have constructed a feasible solution with an objective value equal to the capacity of the minimum $s - t$ cut. Since the optimal value $Z_{s,t}^*$ is the minimum value over all feasible solutions, it must be less than or equal to the value of this particular solution. Thus, $Z_{s,t}^* \leq \text{min-cut}(s, t)$.

**Part 2:** $Z_{s,t}^* \geq \textbf{min-cut}(s, t)$

This inequality is established by analysing the dual of the Nodal Tension LP. The Nodal Tension LP can be written as:

$$\text{Minimise} \quad \sum_{(i,j) \in E} w_{ij} d_{ij} \tag{41}$$

$$\text{s.t.} \quad d_{ij} - p_i + p_j \geq 0 \quad \forall (i, j) \in E \tag{42}$$

$$d_{ij} + p_i - p_j \geq 0 \quad \forall (i, j) \in E \tag{43}$$

$$p_s = 0, \quad p_t = 1 \tag{44}$$

The dual of this LP is a maximum flow problem. By associating dual variables with each constraint, one can show via standard linear programming duality (Schrijver, 1998) that the dual problem is equivalent to finding the maximum flow from source $s$ to sink $t$ in a network where each edge $(i, j)$ has a capacity of $w_{ij}$.

By the strong duality theorem of linear programming, the optimal value of a primal LP is equal to the optimal value of its dual LP, provided a feasible solution exists. Since our problem is feasible, we have:

$$Z^*_{s,t} = \text{max-flow}(s, t) \tag{45}$$

Furthermore, the fundamental max-flow min-cut theorem states that the value of the maximum flow from a source $s$ to a sink $t$ in a network is equal to the capacity of the minimum $s - t$ cut (Ford Jr & Fulkerson, 1956; Kleinberg & Tardos, 2006).

$$\text{max-flow}(s, t) = \text{min-cut}(s, t) \tag{46}$$

Combining these two equalities, we get $Z^*_{s,t} = \text{min-cut}(s, t)$. Thus, $Z^*_{s,t} \geq \text{min-cut}(s, t)$.

Since we have shown both $Z^*_{s,t} \leq \text{min-cut}(s, t)$ and $Z^*_{s,t} \geq \text{min-cut}(s, t)$, we conclude that $Z^*_{s,t} = \text{min-cut}(s, t)$. □

## A.2 Proof of Theorem 3.2

**Theorem** (Restatement of Theorem 3.2). *For a graph with arbitrary positive edge weights $w_{ij}$ and integer separation constraints ($p_s = 0, p_t = 1$), an optimal vertex solution to the Nodal Tension LP exists where every node potential $p_i \in \{0, 1\}$.*

*Proof.* The proof is based on the theory of totally unimodular matrices. A matrix is totally unimodular (TUM) if the determinant of every square submatrix is either 0, +1, or -1. A fundamental theorem in polyhedral combinatorics states that if the constraint matrix $A$ of a linear program is totally unimodular and the right-hand-side vector $b$ is integer-valued, then every basic feasible solution (i.e., every vertex of the feasible polyhedron) is integer, regardless of whether the objective function coefficients are integers. Our proof proceeds by showing that our Nodal Tension LP satisfies these conditions.

**Step 1: Expressing the LP in Standard Form**

First, we write the constraints of the Nodal Tension LP as a system of linear inequalities $Ax \geq b$. Let the combined vector of primal variables be $(\ldots, p_i, \ldots, d_{ij}, \ldots)^T$. The constraints are:

$$d_{ij} - p_i + p_j \geq 0 \quad \forall (i, j) \in E \tag{47}$$
$$d_{ij} + p_i - p_j \geq 0 \quad \forall (i, j) \in E \tag{48}$$
$$p_s \geq 0 \tag{49}$$
$$-p_s \geq 0 \tag{50}$$
$$p_t \geq 1 \tag{51}$$
$$-p_t \geq -1 \tag{52}$$

The right-hand-side vector $b$ is composed of 0s, 1s, and -1s, and is therefore an integer vector.

**Step 2: Analysing the Constraint Matrix A**

The constraint matrix $A$ exhibits a specific block structure derived from the graph's topology. Let $B$ be the $|V| \times |E|$ node-edge incidence matrix of an arbitrarily directed version of the graph $G$, where $B_{v,e} = 1$ if edge $e$ enters node $v$, $-1$ if it leaves, and 0 otherwise. The tension constraints $d_{ij} \geq p_i - p_j$ and $d_{ij} \geq p_j - p_i$ can be expressed in matrix block form as $Id - B^T p \geq 0$ and $Id + B^T p \geq 0$, where $I$ is the identity matrix.

It is a foundational result in polyhedral combinatorics (Schrijver, 1998) that the directed node-edge incidence matrix $B$ (and thus its transpose $B^T$) is totally unimodular. Furthermore, standard operations that preserve total unimodularity include appending identity matrices and duplicating matrix blocks with sign inversions. Therefore, the complete composed constraint matrix $A$ is totally unimodular.

Crucially, if the environmental landscape is instead modelled as a directed graph to capture asymmetric edge weights (anisotropic movement), the constraint matrix inherently retains this exact directed node-edge incidence structure. Thus, the TUM property and strict integrality guarantees firmly hold for directed formulations as well.

**Step 3: Concluding Integrality of Potentials**

Since the constraint matrix $A$ is totally unimodular and the right-hand-side vector $b$ is integer, we conclude that all basic feasible solutions of the LP are integer. This means that for any optimal solution found by a vertex-following algorithm (such as the Simplex method), all variables, including the potentials $\{p_i\}$, will be integer-valued.

**Step 4: Showing Potentials are in {0, 1}**

We have established that an optimal integer solution $\{p_i^*\}$ exists. Now we show that these integers must be either 0 or 1. Assume for contradiction that there exists an optimal solution where some potential $p_k^*$ is an integer not in $\{0, 1\}$.

- Case 1: $p_k^* \geq 2$. Consider a new solution where we set $p_k' = p_k^* - 1$ and keep all other potentials the same. This new potential $p_k'$ is still an integer and is closer to the fixed potentials $p_s = 0$ and $p_t = 1$. For any edge $(k, j)$, the term $|p_k' - p_j^*|$ will be less than or equal to $|p_k^* - p_j^*|$. Therefore, the total objective value $\sum w_{ij}|p_i - p_j|$ for this new solution can only decrease or stay the same. This contradicts the assumption that the original solution was optimal (or shows that a better or equivalent solution exists with a smaller potential).

- Case 2: $p_k^* \leq -1$. A similar argument holds by setting $p_k' = p_k^* + 1$.

Given that all edge weights $w_{ij}$ are positive, the objective function $\sum w_{ij}|p_i - p_j|$ can only be minimised if the integer potentials are forced to lie within the range of the fixed boundary potentials. Since the boundary potentials are $p_s = 0$ and $p_t = 1$, all optimal integer potentials must be in the set $\{0, 1\}$.

This completes the proof that an optimal solution exists where the potentials directly partition the graph's nodes into two sets corresponding to the minimum $s - t$ cut. □

## A.3 Proof of Theorem 3.3

**Theorem** (Restatement of Theorem 3.3). *Let $Z_{s,t}^*$ be the optimal value of the Nodal Tension LP for a given source $s$ and target $t$ on a weighted graph. Let $(S, V \setminus S)$ be the corresponding minimum $s - t$ cut, with $vol(S) \leq vol(V)/2$. The algebraic connectivity of the graph, $\lambda_2$, is bounded above by:*

$$\lambda_2 \leq 2\frac{Z_{s,t}^*}{vol(S)}$$

*Proof.* The proof connects our local resilience measure, the $L_1$ capacity $Z_{s,t}^*$, to the global algebraic connectivity, $\lambda_2$, by leveraging the weighted version of the Cheeger inequality (Chung, 1997).

**Step 1: The Weighted Cheeger Inequality**

For any weighted, undirected graph, the algebraic connectivity $\lambda_2$ is bounded above by the weighted Cheeger constant $\phi(G)$:

$$\lambda_2 \leq 2\phi(G) \tag{53}$$

The weighted Cheeger constant is defined as the minimum ratio of the total weight of edges in a cut to the volume of the smaller partition. The volume of a set of vertices $U$, denoted $vol(U)$, is the sum of the weighted degrees of the nodes in $U$.

$$\phi(G) = \min_{U \subset V, vol(U) \leq vol(V)/2} \frac{w(U, V \setminus U)}{vol(U)} \tag{54}$$

where $w(U, V \setminus U)$ is the sum of weights of edges crossing the cut between partition $U$ and its complement.

**Step 2: Relating the LP Value to the Cheeger Constant**

From Theorem 3.1, we know that the optimal value of our LP, $Z_{s,t}^*$, is equal to the capacity of the minimum weighted $s - t$ cut. Let the partition corresponding to this minimum $s - t$ cut be $(S, V \setminus S)$. Then:

$$Z_{s,t}^* = w(S, V \setminus S) \tag{55}$$

As stated in the theorem conditions, we assume $\text{vol}(S) \leq \text{vol}(V \setminus S)$, which means $\text{vol}(S) \leq \text{vol}(V)/2$. (If the target partition $V \setminus S$ is smaller, the logic symmetrically applies by using $\text{vol}(V \setminus S)$ as the denominator).

The sparsity of this specific cut $(S, V \setminus S)$ is $\frac{w(S, V \setminus S)}{\text{vol}(S)} = \frac{Z_{s,t}^*}{\text{vol}(S)}$. The Cheeger constant $\phi(G)$ is the minimum sparsity over *all possible* cuts in the graph that satisfy the volume constraint. Since our specific $s - t$ cut is just one of these possible cuts, its sparsity must be greater than or equal to the minimum possible sparsity ($\phi(G)$). Therefore:

$$\phi(G) \leq \frac{Z_{s,t}^*}{\text{vol}(S)} \tag{56}$$

**Step 3: Combining the Inequalities and Analysing the Bound Limitation**

We can now substitute the result from Step 2 into the Cheeger inequality from Step 1:

$$\lambda_2 \leq 2\phi(G) \leq 2\frac{Z_{s,t}^*}{\text{vol}(S)} \tag{57}$$

This establishes the desired upper bound for the weighted case and completes the algebraic proof.

As discussed in the main text, while sound, this derivation explicitly exposes the inherent limitation of using global spectral bounds for local resilience. Because $Z_{s,t}^*$ is divided by $\text{vol}(S)$, any highly unbalanced local bottleneck forces the upper bound toward infinity as $\text{vol}(S)$ becomes small relative to the network. This slack necessitates the derivation of the tight, local dynamical bounds (such as the Local Cheeger Inequality for commute times) presented in Section 4. $\qquad\square$

# B  Detailed Proofs for the Geometry and Learning of Local Resilience in Section 4

In this appendix, we provide the proofs for the geometric, dynamical, and algorithmic properties of the Nodal Tension model introduced in Section 4. These proofs rely on spectral graph theory, differential geometry, and continuous optimisation, maintaining a deliberate separation from the foundational operations research proofs presented in Appendix A.

## B.1  Proof of Lemma 4.1

**Lemma** (Restatement of Lemma 4.1)**.** *For any weighted, undirected graph $G = (V, E, w)$ and distinct nodes $s, t \in V$, the $L_2$ effective conductance $C_{s,t}$ is bounded above by the $L_1$ optimal Nodal Tension $Z_{s,t}^*$:*

$$C_{s,t} \leq Z_{s,t}^* \tag{58}$$

*Furthermore, the magnitude of the redundancy gap, $\Delta_{s,t} = Z_{s,t}^* - C_{s,t}$, is non-negative and quantifies the structural redundancy of the pathway: $\Delta_{s,t}$ grows as the capacity of parallel, edge-disjoint routes bypassing the primary bottleneck increases.*

*Proof.* The proof proceeds by explicitly mapping the discrete $L_1$ optimal solution into the continuous $L_2$ variational space.

**Step 1: Variational Formulations**

We begin with the variational definitions of both metrics. By Dirichlet's Principle, the effective conductance $C_{s,t}$ is the absolute minimum of the $L_2$ Dirichlet energy over all continuous node potential vectors $x \in \mathbb{R}^{|V|}$ satisfying the boundary conditions $x_s = 0$ and $x_t = 1$:

$$C_{s,t} = \inf_{\substack{x \in \mathbb{R}^{|V|} \\ x_s=0, x_t=1}} \sum_{(i,j) \in E} w_{ij}(x_i - x_j)^2 \tag{59}$$

Conversely, by Theorem 3.1, the optimal Nodal Tension $Z_{s,t}^*$ is the capacity of the minimum $s-t$ cut, which can be expressed as the minimum of the $L_1$ total variation over node potentials $p \in \mathbb{R}^{|V|}$ subject to the identical boundary conditions:

$$Z_{s,t}^* = \min_{\substack{p \in \mathbb{R}^{|V|} \\ p_s=0, p_t=1}} \sum_{(i,j) \in E} w_{ij}|p_i - p_j| \tag{60}$$

**Step 2: Feasibility and Evaluation of the $L_1$ Optimum**

Let $p^*$ denote an optimal vertex solution to the $L_1$ Nodal Tension LP, such that $\sum w_{ij}|p_i^* - p_j^*| = Z_{s,t}^*$. By Theorem 3.2 (Total Unimodularity), we are guaranteed that $p^*$ is binary, meaning $p_i^* \in \{0, 1\}$ for all $i \in V$.

Because $p^*$ is a real-valued vector ($p^* \in \mathbb{R}^{|V|}$) and satisfies the boundary constraints $p_s^* = 0$ and $p_t^* = 1$, it is a valid, feasible candidate solution for the continuous $L_2$ Dirichlet energy minimisation problem.

We now evaluate the $L_2$ Dirichlet energy at the point $p^*$. Since both $p_i^*$ and $p_j^*$ belong to $\{0, 1\}$, their difference must be an integer: $(p_i^* - p_j^*) \in \{-1, 0, 1\}$. For any value $y \in \{-1, 0, 1\}$, the algebraic identity $y^2 = |y|$ holds true. Therefore, for every individual edge $(i, j)$ in the graph:

$$(p_i^* - p_j^*)^2 = |p_i^* - p_j^*| \tag{61}$$

Summing this equivalence over all edges, weighted by $w_{ij}$, yields:

$$\sum_{(i,j) \in E} w_{ij}(p_i^* - p_j^*)^2 = \sum_{(i,j) \in E} w_{ij}|p_i^* - p_j^*| = Z_{s,t}^* \tag{62}$$

**Step 3: Establishing the Upper Bound**

The definition of a global infimum requires that the optimal value $C_{s,t}$ must be less than or equal to the objective value evaluated at any arbitrary feasible point in the domain. Since $p^*$ is a feasible point, we trivially obtain:

$$C_{s,t} \leq \sum_{(i,j) \in E} w_{ij}(p_i^* - p_j^*)^2 \tag{63}$$

Substituting the result from Step 2 directly produces the bound:

$$C_{s,t} \leq Z_{s,t}^* \tag{64}$$

**Step 4: Algebraic Quantification of the Gap**

Given $C_{s,t} \leq Z_{s,t}^*$, we define the redundancy gap as $\Delta_{s,t} = Z_{s,t}^* - C_{s,t} \geq 0$. We now prove what structural property this gap quantifies by deriving its algebraic form.

Let $C^* \subset E$ be the set of edges constituting the minimum $s-t$ cut, such that $Z_{s,t}^* = \sum_{(u,v) \in C^*} w_{uv}$. Let $\phi^*$ be the unique continuous $L_2$ minimizer. By standard electrical network theory, the effective conductance $C_{s,t}$ is equal to the total electrical flow from $s$ to $t$ under a unit boundary potential difference.

By Kirchhoff's Current Law (Doyle & Snell, 1984; Bollobás, 2011), the net flow across any topological $s-t$ partition, including the minimum cut $C^*$, must equal the total network flow $C_{s,t}$. The flow across a specific directed edge $(u, v) \in C^*$ is defined by Ohm's Law as $f_{uv} = w_{uv}(\phi_u^* - \phi_v^*)$. Therefore, evaluating the total flow across the minimum cut yields an exact equality for the conductance:

$$C_{s,t} = \sum_{(u,v) \in C^*} w_{uv}(\phi_u^* - \phi_v^*) \tag{65}$$

Furthermore, because $\phi_s^* = 0$, $\phi_t^* = 1$, and $\phi^*$ satisfies the harmonic maximum principle, the continuous potential difference across any single edge is bounded: $\phi_u^* - \phi_v^* \leq 1$.

We can now express the gap $\Delta_{s,t}$ algebraically by substituting this flow equivalence:

$$\Delta_{s,t} = Z_{s,t}^* - C_{s,t} = \sum_{(u,v) \in C^*} w_{uv} - \sum_{(u,v) \in C^*} w_{uv}(\phi_u^* - \phi_v^*) = \sum_{(u,v) \in C^*} w_{uv}\big(1 - (\phi_u^* - \phi_v^*)\big) \qquad (66)$$

Because $\phi_u^* - \phi_v^* \leq 1$ and edge weights $w_{uv} > 0$, every term in the summation is non-negative, confirming $\Delta_{s,t} \geq 0$.

Crucially, this exact algebraic form reveals the physical meaning of the gap. The gap $\Delta_{s,t} = 0$ if and only if $\phi_u^* - \phi_v^* = 1$ for every edge in the minimum cut $C^*$. Since the global potential difference is fixed at 1 ($\phi_t^* - \phi_s^* = 1$), this requires that no potential drop occurs anywhere else in the network; the $L_2$ minimizer must be constant on the source-side component and constant on the target-side component.

Physically, this means the gap is zero if and only if the network possesses infinite conductance (zero resistance) everywhere except directly across the $L_1$ bottleneck. Conversely, any positive gap ($\Delta_{s,t} > 0$) isolates and quantifies the aggregate series resistance (the required structural potential drops) contributed by the remainder of the network outside the primary minimum cut. Thus, $\Delta_{s,t}$ is a precise algebraic measure of how "diffuse" the topological bottleneck is. $\qquad \square$

## B.2  Proof of Corollary 4.2

**Corollary** (Restatement of Corollary 4.2)**.** *For a given $L_1$ local bottleneck capacity $Z_{s,t}^*$, the magnitude of the redundancy gap $\Delta_{s,t} = Z_{s,t}^* - C_{s,t}$ uniquely classifies the structural topology of the vulnerability into two functional extremes:*

1. ***Strict Topological Bridge ($\Delta_{s,t} \to 0$)**: The effective conductance approaches the minimum cut capacity ($C_{s,t} \approx Z_{s,t}^*$). The network resistance is strictly concentrated at a singular structural interface.*

2. ***Diffuse Bottleneck ($\Delta_{s,t} \to Z_{s,t}^*$)**: The effective conductance is fractionally smaller than the minimum cut ($C_{s,t} \ll Z_{s,t}^*$). The network resistance is distributed across a prolonged, fragmented sequence of edges outside the primary cut.*

*Proof.* Let $\phi^* \in \mathbb{R}^{|V|}$ be the unique harmonic extension minimising the $L_2$ Dirichlet energy subject to $\phi_s^* = 0$ and $\phi_t^* = 1$. Let $C^* \subset E$ denote the minimum $s - t$ cut set, partitioning the vertices into a source set $V_s$ and a target set $V_t$. By Lemma 4.1, the algebraic gap is exactly:

$$\Delta_{s,t} = \sum_{(u,v) \in C^*} w_{uv}\big(1 - (\phi_u^* - \phi_v^*)\big) \qquad (67)$$

By the Maximum Principle for harmonic functions on graphs (Doyle & Snell, 1984; Levin & Peres, 2017), $0 \leq \phi_i^* \leq 1$ for all $i \in V$, and therefore $0 \leq (\phi_u^* - \phi_v^*) \leq 1$ for all directed edges $(u,v)$ where $u \in V_t, v \in V_s$. Let $\mathcal{P}$ be the set of all simple paths from $s$ to $t$. For any path $P \in \mathcal{P}$, the telescopic sum of potential differences along the edges of $P$ must equal the global boundary condition:

$$\sum_{(i,j) \in P} (\phi_i^* - \phi_j^*) = \phi_t^* - \phi_s^* = 1 \qquad (68)$$

**Case 1: Strict Topological Bridge ($\Delta_{s,t} \to 0$)**

Assume $\Delta_{s,t} \to 0$. Since $w_{uv} > 0$ and $\big(1 - (\phi_u^* - \phi_v^*)\big) \geq 0$, the sum converges to zero if and only if $(\phi_u^* - \phi_v^*) \to 1$ for every edge $(u,v) \in C^*$.

Consider any path $P \in \mathcal{P}$. Since $C^*$ is a valid $s-t$ cut, $P$ must contain at least one forward edge $(u,v) \in C^*$. Substituting $(\phi_u^* - \phi_v^*) \to 1$ into the telescopic path sum yields:

$$1 + \sum_{(i,j) \in P \setminus C^*} (\phi_i^* - \phi_j*) \to 1 \implies \sum_{(i,j) \in P \setminus C^*} (\phi_i^* - \phi_j^*) \to 0 \tag{69}$$

Because no potential difference can be negative along the direction of net flow, this requires $(\phi_i^* - \phi_j^*) \to 0$ for all edges $(i,j) \notin C^*$. By Ohm's Law, the effective resistance of any edge is given by $R_{ij} = (\phi_i^* - \phi_j^*)/f_{ij}$. For a finite non-zero flow $f_{ij}$ to exist under a zero potential gradient, the resistance of the subgraphs induced by $V_s$ and $V_t$ must approach zero. Thus, the topology consists of two conductive (infinitely dense) components connected exclusively by $C^*$.

**Case 2: Diffuse Bottleneck ($\Delta_{s,t} \to Z_{s,t}^*$)**

Assume $\Delta_{s,t} \to Z_{s,t}^*$. By the definition of the gap $\Delta_{s,t} = Z_{s,t}^* - C_{s,t}$, this condition is algebraically equivalent to $C_{s,t} \to 0$.

Substituting $C_{s,t} = \sum_{C^*} w_{uv}(\phi_u^* - \phi_v^*) \to 0$, and given $w_{uv} > 0$, it follows that the potential drop across the cut edges vanishes: $(\phi_u^* - \phi_v^*) \to 0$ for all $(u,v) \in C^*$.

Returning to the telescopic path sum for any path $P \in \mathcal{P}$, we substitute the vanishing cut potential:

$$0 + \sum_{(i,j) \in P \setminus C^*} (\phi_i^* - \phi_j^*) \to 1 \tag{70}$$

This establishes that the entire unit potential drop $\phi_t^* - \phi_s^* = 1$ is consumed by the edges in $E \setminus C^*$. Because $\phi^*$ minimises the $L_2$ Dirichlet energy, this extreme distribution of potential differences physically requires a prolonged sequence of low-capacity, un-bypassed edges in the sets $V_s$ and $V_t$. The resistance is proven to be distributed across the path length rather than localised at the $L_1$ partition, fulfilling the definition of a diffuse bottleneck. $\square$

## B.3 Proof of Corollary 4.3

**Corollary** (Restatement of Corollary 4.3). *For any $L_1$ minimum $s-t$ cut set $C^*$, the redundancy gap $\Delta_{s,t}$ is equivalent to the aggregate unutilised structural capacity of the cut edges under $L_2$ optimal flow. Consequently, in a diffuse bottleneck ($\Delta_{s,t} \to Z_{s,t}^*$), the absolute weakest link $C^*$ is severely flow-starved; its maximum capacity is unutilised due to the overriding series resistance of the surrounding network.*

*Proof.* Let $C^* \subset E$ be the minimum $s-t$ cut, defining the $L_1$ bottleneck capacity $Z_{s,t}^* = \sum_{(u,v) \in C^*} w_{uv}$. Let $\phi^* \in \mathbb{R}^{|V|}$ be the harmonic potential vector minimising the $L_2$ Dirichlet energy subject to $\phi_s^* = 0$ and $\phi_t^* = 1$.

By standard electrical network theory (Doyle & Snell, 1984; Levin & Peres, 2017), the total effective conductance $C_{s,t}$ is the aggregate electrical flow crossing any valid $s-t$ partition. Evaluating this flow specifically across the minimum cut $C^*$, and applying Ohm's Law $f_{uv}^* = w_{uv}(\phi_u^* - \phi_v^*)$ to each directed edge $(u,v) \in C^*$, yields:

$$C_{s,t} = \sum_{(u,v) \in C^*} f_{uv}^* = \sum_{(u,v) \in C^*} w_{uv}(\phi_u^* - \phi_v^*) \tag{71}$$

We substitute this flow equivalence directly into the definition of the redundancy gap, $\Delta_{s,t} = Z_{s,t}^* - C_{s,t}$:

$$\Delta_{s,t} = \sum_{(u,v) \in C^*} w_{uv} - \sum_{(u,v) \in C^*} f_{uv}^* = \sum_{(u,v) \in C^*} (w_{uv} - f_{uv}^*) \tag{72}$$

By the Maximum Principle for harmonic functions (Doyle & Snell, 1984; Levin & Peres, 2017), the potential difference across any directed edge between the source and target components is bounded: $(\phi_u^* - \phi_v^*) \leq 1$.

Multiplying by the positive edge weights $w_{uv} > 0$ yields $w_{uv}(\phi_u^* - \phi_v^*) \leq w_{uv}$, which algebraically proves that $f_{uv}^* \leq w_{uv}$.

Therefore, every term in the summation $(w_{uv} - f_{uv}^*)$ is non-negative. This establishes that the gap $\Delta_{s,t}$ is the algebraic sum of the unutilised capacities (the residual bandwidth) of the edges in the minimum cut under optimal $L_2$ flow.

To prove the limit behaviour for a diffuse bottleneck, we evaluate the condition $\Delta_{s,t} \to Z_{s,t}^*$. Substituting the gap and capacity definitions:

$$\sum_{(u,v) \in C^*} (w_{uv} - f_{uv}^*) \to \sum_{(u,v) \in C^*} w_{uv} \tag{73}$$

This condition can only hold if the aggregate flow crossing the cut approaches zero: $\sum_{C^*} f_{uv}^* \to 0$. As proven in the limit analysis of Corollary 4.2, this reduction in global conductance forces the local potential difference across the cut to vanish, meaning $(\phi_u^* - \phi_v^*) \to 0$ for every individual edge $(u, v) \in C^*$.

Consequently, $f_{uv}^* \to 0$ for every edge in the bottleneck. Since the capacity $w_{uv}$ remains fixed and positive, the ratio of utilised flow to structural capacity, $f_{uv}^*/w_{uv}$, approaches zero. This proves that the theoretical $L_1$ bottleneck capacity is rendered unutilised (flow-starved) due to the lack of electrical potential gradient reaching the cut interface, a direct consequence of the series resistance of the surrounding diffuse network. $\quad\square$

### B.4 Proof of Lemma 4.4

**Lemma** (Restatement of Lemma 4.4). *Let $p^*$ be the optimal continuous solution to the Nodal Tension LP, and let $Z_{s,t}^*$ denote its optimal objective value (the $L_1$ bottleneck capacity). For an unweighted graph, if an edge $e = (u, v)$ sustains maximal tension such that $|p_u^* - p_v^*| = 1$, the geometric severance of the partition limits its shared triangles to $|\triangle_{uv}| \leq Z_{s,t}^* - 1$. Consequently, its discrete Forman-Ricci curvature is explicitly bounded from above by the global LP capacity and the local degrees:*

$$\mathbf{F}(u, v) \leq 4 - d_u - d_v + 3(Z_{s,t}^* - 1) \tag{74}$$

*Proof.* Let $G = (V, E)$ be an unweighted graph where $w_{ij} = 1$ for all $(i, j) \in E$. Let $p^* \in \mathbb{R}^{|V|}$ be the optimal solution to the Nodal Tension LP. By the integrality established in Theorem 3.2, the optimal potentials partition the vertex set into two disjoint components: a source set $V_s = \{i \in V \mid p_i^* = 0\}$ and a target set $V_t = \{i \in V \mid p_i^* = 1\}$.

Let $C^* = \{(x, y) \in E \mid x \in V_s, y \in V_t\}$ denote the corresponding minimum $s - t$ cut set. For an unweighted graph, the LP objective value $Z_{s,t}^*$ is equivalent to the cardinality of this cut set: $Z_{s,t}^* = \sum_{(x,y) \in C^*} 1 = |C^*|$.

Assume there exists an edge $e = (u, v) \in E$ that sustains maximal tension, $|p_u^* - p_v^*| = 1$. This condition necessitates that one endpoint belongs to $V_s$ and the other to $V_t$. Without loss of generality, let $u \in V_s$ and $v \in V_t$. Therefore, $(u, v) \in C^*$.

We now evaluate the set of shared triangles incident to $(u, v)$, denoted $\triangle_{uv}$. A shared triangle is defined by the existence of a common neighbour $w \in V$ such that both $(u, w) \in E$ and $(w, v) \in E$.

Because $V_s \cup V_t = V$ forms a collectively exhaustive and mutually exclusive partition of the vertices, the common neighbour $w$ must reside within either $V_s$ or $V_t$. This yields two disjoint cases for any triangle formed by $w$:

1. **Case 1 ($w \in V_s$):** Since $w \in V_s$ and $v \in V_t$, the edge $(w, v)$ spans the partition. By definition, $(w, v) \in C^*$.

2. **Case 2 ($w \in V_t$):** Since $u \in V_s$ and $w \in V_t$, the edge $(u, w)$ spans the partition. By definition, $(u, w) \in C^*$.

To formalise the cardinality bound, we construct a mapping $f : \triangle_{uv} \to C^* \setminus \{(u, v)\}$. For each triangle defined by a common neighbour $w$, we map it to an edge in the cut set as follows:

$$f(w) = \begin{cases} (w, v) & \text{if } w \in V_s \\ (u, w) & \text{if } w \in V_t \end{cases} \tag{75}$$

Because each common neighbour $w$ is a distinct vertex, the edges $(w, v)$ and $(u, w)$ are distinct for different triangles. Therefore, the mapping $f$ is injective.

By the principles of discrete mathematics, the existence of an injective mapping from domain $A$ to codomain $B$ guarantees that $|A| \leq |B|$. Thus, the number of shared triangles is bounded by the cardinality of the codomain:

$$|\triangle_{uv}| \leq \left| C^* \setminus \{(u, v)\} \right| \tag{76}$$

Since $(u, v) \in C^*$, the cardinality of the residual cut set is $|C^*| - 1$. Substituting $Z_{s,t}^* = |C^*|$, we obtain the strict algebraic bound:

$$|\triangle_{uv}| \leq Z_{s,t}^* - 1 \tag{77}$$

Finally, we apply this constraint to the unweighted Forman-Ricci curvature equation. Substituting Equation 77 into $\mathbf{F}(u, v) = 4 - d_u - d_v + 3|\triangle_{uv}|$ preserves the inequality, yielding:

$$\mathbf{F}(u, v) \leq 4 - d_u - d_v + 3(Z_{s,t}^* - 1) \tag{78}$$

This establishes that the local differential geometry of the edge is explicitly bounded from above by the macroscopic optimal LP capacity, completing the proof. $\square$

### B.5    Proof of Lemma 4.5

**Lemma** (Restatement of Lemma 4.5)**.** *Let $p^*$ be the optimal continuous solution to the Nodal Tension LP, and $Z_{s,t}^*$ the global bottleneck capacity. The optimal tension gradient across any edge $e = (u, v)$ is restricted by its local positive curvature. Specifically, assigning maximal tension across an edge requires simultaneously exhausting the capacity of all its shared triangles, establishing the strict algebraic bound:*

$$|p_u^* - p_v^*| \leq \frac{Z_{s,t}^*}{1 + |\triangle_{uv}|} \tag{79}$$

*Substituting the components of Forman-Ricci curvature (Equation 11), the continuous tension gradient is explicitly inversely bounded by the positive local geometry:*

$$|p_u^* - p_v^*| \leq \frac{3Z_{s,t}^*}{\mathbf{F}(u, v) + d_u + d_v - 1} \tag{80}$$

*Proof.* Let $G = (V, E)$ be an unweighted graph where $w_{ij} = 1$ for all $(i, j) \in E$. Let $p^* \in \mathbb{R}^{|V|}$ be the optimal solution to the Nodal Tension LP, with the optimal objective value $Z_{s,t}^* = \sum_{(i,j) \in E} |p_i^* - p_j^*|$. By Theorem 3.2, the optimal potentials $p_i^*$ are binary, taking values in $\{0, 1\}$. Consequently, the tension gradient across any edge $(u, v) \in E$ is also binary: $|p_u^* - p_v^*| \in \{0, 1\}$.

We evaluate the algebraic bound for both possible states of the tension gradient:

**Case 1 (Zero Tension):** Assume $|p_u^* - p_v^*| = 0$. Because the graph capacity $Z_{s,t}^* \geq 0$ and the number of shared triangles $|\triangle_{uv}| \geq 0$, the denominator $1 + |\triangle_{uv}|$ is positive. Therefore, the inequality $0 \leq \frac{Z_{s,t}^*}{1+|\triangle_{uv}|}$ holds trivially.

**Case 2 (Maximal Tension):** Assume $|p_u^* - p_v^*| = 1$. This dictates that the edge $(u, v)$ belongs to the strict $L_1$ minimum cut set $C^*$, and therefore $Z_{s,t}^* = |C^*|$. As established in the proof of Lemma 4.4 (Appendix

B.4), the existence of a valid bipartite $s - t$ partition creates an injective mapping from the set of shared triangles $\triangle_{uv}$ to the set of residual cut edges $C^* \setminus \{(u, v)\}$.

Because every shared triangle must injectively map to at least one distinct additional edge within the cut set, the total cardinality of the cut set must be greater than or equal to the edge $(u, v)$ itself (contributing 1 unit of capacity) plus the number of mapped edges from the triangles (contributing $|\triangle_{uv}|$ units of capacity). This imposes the strict geometric minimum on the cut size:

$$|C^*| \geq 1 + |\triangle_{uv}| \tag{81}$$

Substituting the LP capacity $Z_{s,t}^* = |C^*|$ into this inequality yields $Z_{s,t}^* \geq 1 + |\triangle_{uv}|$. Dividing both sides by the positive term $(1 + |\triangle_{uv}|)$ isolates the constant 1:

$$1 \leq \frac{Z_{s,t}^*}{1 + |\triangle_{uv}|} \tag{82}$$

Since we assumed $|p_u^* - p_v^*| = 1$ for this case, we substitute the tension gradient directly into the left side of the inequality, yielding $|p_u^* - p_v^*| \leq \frac{Z_{s,t}^*}{1 + |\triangle_{uv}|}$.

Because the inequality holds for both possible integer states of the optimal continuous variables, the bound is globally valid for the Nodal Tension LP.

To formulate this purely in terms of differential geometry, we substitute the exact definition of Forman-Ricci curvature. By algebraically isolating $|\triangle_{uv}|$ from Equation 11:

$$\mathbf{F}(u, v) = 4 - d_u - d_v + 3|\triangle_{uv}| \implies |\triangle_{uv}| = \frac{\mathbf{F}(u, v) - 4 + d_u + d_v}{3} \tag{83}$$

We substitute this expression into the denominator of our continuous capacity bound:

$$1 + |\triangle_{uv}| = 1 + \frac{\mathbf{F}(u, v) - 4 + d_u + d_v}{3} = \frac{3 + \mathbf{F}(u, v) - 4 + d_u + d_v}{3} = \frac{\mathbf{F}(u, v) + d_u + d_v - 1}{3} \tag{84}$$

Substituting this final denominator back into the tension gradient inequality and simplifying yields:

$$|p_u^* - p_v^*| \leq \frac{Z_{s,t}^*}{\frac{\mathbf{F}(u,v) + d_u + d_v - 1}{3}} = \frac{3 Z_{s,t}^*}{\mathbf{F}(u, v) + d_u + d_v - 1} \tag{85}$$

This completes the proof. The bound guarantees that as the positive curvature of the local manifold increases, the denominator restricts the maximal assignable tension. If the local structural capacity of the triangles exceeds the global bottleneck capacity $(1 + |\triangle_{uv}| > Z_{s,t}^*)$, the optimal continuous tension $|p_u^* - p_v^*|$ is forced to drop to 0. $\qquad \square$

## B.6    Proof of Theorem 4.6

**Theorem** (Restatement of Theorem 4.6). *Let $p^*$ be the optimal continuous solution to the Nodal Tension LP, yielding the minimal $s - t$ separating manifold $C^*$ with capacity $Z_{s,t}^*$. The global minimisation of the $L_1$ objective localises the tension partition within the cross-section of minimal structural cohesion. Consequently, the aggregate discrete Forman-Ricci curvature of the optimal bottleneck is explicitly bounded from above by the local neighbourhood dispersions and the square of the global capacity:*

$$\sum_{(u,v)\in C^*} \mathbf{F}(u, v) \leq \sum_{(u,v)\in C^*} (4 - d_u - d_v) + 3 Z_{s,t}^*(Z_{s,t}^* - 1) \tag{86}$$

*Proof.* Let $G = (V, E)$ be an unweighted graph where $w_{ij} = 1$ for all $(i, j) \in E$. Let $p^* \in \mathbb{R}^{|V|}$ be the optimal solution to the Nodal Tension LP. The optimal potentials induce a strict bipartite $s - t$ partition, defining

the minimum cut set $C^* = \{(u,v) \in E \mid |p_u^* - p_v^*| = 1\}$. For an unweighted graph, the global bottleneck capacity is equal to the cardinality of this cut set: $Z_{s,t}^* = |C^*|$.

As established in Lemma 4.4 (Appendix B.4), the geometric severance of the $s-t$ partition limits the number of shared triangles for every individual cut edge $(u,v) \in C^*$ via an injective mapping to the residual cut edges. This imposes the local algebraic bound on the discrete Forman-Ricci curvature for each edge in the manifold:

$$\mathbf{F}(u,v) \leq 4 - d_u - d_v + 3(Z_{s,t}^* - 1) \quad \forall(u,v) \in C^* \tag{87}$$

To determine the differential geometry of the entire minimal separating surface, we aggregate the discrete curvature over the complete $L_1$ partition. We sum the local inequality over all edges $(u,v) \in C^*$:

$$\sum_{(u,v)\in C^*} \mathbf{F}(u,v) \leq \sum_{(u,v)\in C^*} \left[4 - d_u - d_v + 3(Z_{s,t}^* - 1)\right] \tag{88}$$

By the linearity of summation, we separate the local neighbourhood dispersion terms from the global capacity term:

$$\sum_{(u,v)\in C^*} \mathbf{F}(u,v) \leq \sum_{(u,v)\in C^*} (4 - d_u - d_v) + \sum_{(u,v)\in C^*} 3(Z_{s,t}^* - 1) \tag{89}$$

Crucially, the capacity term $3(Z_{s,t}^* - 1)$ is a global constant with respect to the individual edges in the cut set. Therefore, summing this constant over the exact $|C^*|$ edges of the partition evaluates to the constant multiplied by the cardinality of the set:

$$\sum_{(u,v)\in C^*} 3(Z_{s,t}^* - 1) = |C^*| \cdot 3(Z_{s,t}^* - 1) \tag{90}$$

Substituting the capacity equivalence $|C^*| = Z_{s,t}^*$ yields the exact algebraic expansion:

$$Z_{s,t}^* \cdot 3(Z_{s,t}^* - 1) = 3Z_{s,t}^*(Z_{s,t}^* - 1) \tag{91}$$

Substituting this evaluated constant back into the separated summation yields the final bounded aggregate curvature of the manifold:

$$\sum_{(u,v)\in C^*} \mathbf{F}(u,v) \leq \sum_{(u,v)\in C^*} (4 - d_u - d_v) + 3Z_{s,t}^*(Z_{s,t}^* - 1) \tag{92}$$

This completes the proof. The aggregation guarantees that the global continuous minimisation of the Nodal Tension LP bounds the aggregate discrete curvature of the resulting partition. The optimal manifold is geometrically forced into a state characterised by high negative neighbourhood dispersion (the $-(d_u + d_v)$ summation) and bounded internal topological cohesion (the $O((Z_{s,t}^*)^2)$ term). $\square$

## B.7 Proof of Corollary 4.7

**Corollary** (Restatement of Corollary 4.7). *Let $C^*$ be the optimal $L_1$ separating manifold defining the true $s-t$ topological bottleneck identified by Theorem 4.6. For any locally negatively curved edge $e = (x,y) \in E \backslash C^*$ (where $\mathbf{F}(x,y) < 0$) that is topologically disjoint from this global minimal surface, the optimal continuous tension gradient is forced to zero:*

$$|p_x^* - p_y^*| = 0 \tag{93}$$

*Proof.* Let $p^* \in \{0,1\}^{|V|}$ be the optimal solution to the continuous Nodal Tension LP. By the integrality property established in Theorem 3.2, the variables partition the vertex set $V$ into two disjoint subsets: the source component $V_s = \{v \in V \mid p_v^* = 0\}$ and the target component $V_t = \{v \in V \mid p_v^* = 1\}$.

The optimal cut set $C^*$ is defined as the set of all edges $(u, v) \in E$ that span across these two subsets. For any edge in $C^*$, one endpoint is in $V_s$ and the other is in $V_t$, yielding $|p_u^* - p_v^*| = 1$.

Now consider an arbitrary edge $(x, y) \in E \setminus C^*$. Assume it has negative Forman-Ricci curvature, $\mathbf{F}(x, y) < 0$.

Because $(x, y)$ is not in the cut set $C^*$, it does not span the partition between $V_s$ and $V_t$. This requires that both endpoints $x$ and $y$ reside in the same vertex subset. This leaves two possible cases:

1. Both $x, y \in V_s$. In this case, the optimal potentials are $p_x^* = 0$ and $p_y^* = 0$. The tension gradient evaluates to $|0 - 0| = 0$.

2. Both $x, y \in V_t$. In this case, the optimal potentials are $p_x^* = 1$ and $p_y^* = 1$. The tension gradient evaluates to $|1 - 1| = 0$.

In both cases, the optimal tension gradient across the edge is zero, $|p_x^* - p_y^*| = 0$. This result depends on the topological position of the edge relative to the optimal $s - t$ partition. The local differential geometry of the edge, such as its negative curvature $\mathbf{F}(x, y) < 0$, does not alter the LP capacity constraints. Therefore, any negatively curved edge outside the optimal bottleneck manifold receives zero tension. $\square$

## B.8 Proof of Lemma 4.9

**Lemma** (Restatement of Lemma 4.9). *Let $p^* \in \{0, 1\}^{|V|}$ be the optimal integer solution to the Nodal Tension LP, partitioning the vertex set into a source component $V_s = \{v \in V \mid p_v^* = 0\}$ and a target component $V_t = \{v \in V \mid p_v^* = 1\}$. Let the optimal $s - t$ capacity be $Z_{s,t}^* = |C^*|$. We define the local Nodal Conductance, $\Phi_{s,t}^*$, to evaluate the isoperimetric restriction of the specific $s - t$ pathway:*

$$\Phi_{s,t}^* = \frac{Z_{s,t}^*}{\min(vol(V_s), vol(V_t))} \tag{94}$$

*Because the vertex partitions $V_s$ and $V_t$ constitute a specific, valid subset of the graph where $\min(vol(V_s), vol(V_t)) \leq \frac{1}{2}vol(V)$, the local Nodal Conductance is bounded from below by the absolute global Cheeger constant $h_G$:*

$$h_G \leq \Phi_{s,t}^* \tag{95}$$

*Proof.* Let $G = (V, E)$ be a graph. By the integrality established in Theorem 3.2, the optimal solution $p^*$ defines two subsets: $V_s = \{v \in V \mid p_v^* = 0\}$ and $V_t = \{v \in V \mid p_v^* = 1\}$. Because every vertex $v \in V$ must take one binary state in $\{0, 1\}$, the subsets $V_s$ and $V_t$ are mutually exclusive ($V_s \cap V_t = \emptyset$) and collectively exhaustive ($V_s \cup V_t = V$).

The volume of a set is defined as the sum of the degrees of its contained vertices. Because $V_s$ and $V_t$ partition the total vertex set $V$, the sum of their individual volumes must equal the total volume of the graph:

$$\text{vol}(V_s) + \text{vol}(V_t) = \text{vol}(V) \tag{96}$$

By the algebraic definition of a minimum, if two real numbers sum to a total $T$, the smaller of the two numbers cannot exceed $\frac{T}{2}$. Therefore, it follows that:

$$\min(\text{vol}(V_s), \text{vol}(V_t)) \leq \frac{1}{2}\text{vol}(V) \tag{97}$$

Now, we evaluate this specific partition against the global Cheeger constant. As defined in Theorem 4.8, the global Cheeger constant $h_G$ identifies the absolute minimum isoperimetric ratio over all possible subsets $S \subset V$ that satisfy the volume constraint $\text{vol}(S) \leq \frac{1}{2}\text{vol}(V)$:

$$h_G = \min_{S \subset V, \, \text{vol}(S) \leq \frac{1}{2}\text{vol}(V)} \frac{|E(S, V \setminus S)|}{\text{vol}(S)} \tag{98}$$

Let $S^*$ be the specific subset out of the Nodal Tension partition $\{V_s, V_t\}$ that possesses the smaller volume: $S^* = \arg\min_{S \in \{V_s, V_t\}} \text{vol}(S)$.

Based on Equation 97, $S^*$ satisfies the required volume constraint $\text{vol}(S^*) \leq \frac{1}{2}\text{vol}(V)$. Therefore, $S^*$ is a valid candidate subset within the domain of the global minimisation for $h_G$.

The edges that span $S^*$ and its complement $V \setminus S^*$ are the edges spanning $V_s$ and $V_t$. By definition, this is the optimal $s - t$ minimum cut set $C^*$. For an unweighted graph, the cardinality of this cut set is the Nodal Tension LP capacity: $|E(S^*, V \setminus S^*)| = |C^*| = Z_{s,t}^*$.

Evaluating the isoperimetric ratio specifically for our valid subset $S^*$ yields the definition of local Nodal Conductance:

$$\frac{|E(S^*, V \setminus S^*)|}{\text{vol}(S^*)} = \frac{Z_{s,t}^*}{\min(\text{vol}(V_s), \text{vol}(V_t))} = \Phi_{s,t}^* \tag{99}$$

Because the global Cheeger constant $h_G$ is defined as the absolute minimum over the entire domain of valid subsets, the ratio evaluated at any single valid subset $S^*$ must be greater than or equal to this minimum. This establishes the bound:

$$h_G \leq \Phi_{s,t}^* \tag{100}$$

This completes the proof. The local Nodal Conductance provides a guaranteed upper bound on the absolute graph conductance, correctly evaluating the severity of the specific $s - t$ topological restriction independent of the global spectral minimum. $\qquad\square$

## B.9 Proof of Theorem 4.10

**Theorem** (Restatement of Theorem 4.10). *Let $G = (V, E)$ be an unweighted graph where a discrete-time random walk $(X_k)_{k \geq 0}$ transitions from node $u$ to an adjacent node $v$ with uniform probability $P_{uv} = \frac{1}{d_u}$. The hitting time $h(s, t)$ is explicitly defined as the expected number of steps required for the walk to first reach the target node $t$ given a starting position at the source node $s$:*

$$h(s, t) = \mathbb{E}\left[\min\{k \geq 0 \mid X_k = t\} \mid X_0 = s\right] \tag{101}$$

*The expected commute time $H(s, t)$ is the strict sum of the directional hitting times, representing the total expected duration of the round trip:*

$$H(s, t) = h(s, t) + h(t, s) \tag{102}$$

*Let $p^*$ be the optimal integer solution to the Nodal Tension LP, yielding the minimal $s - t$ separating surface $C^*$ with capacity $Z_{s,t}^*$. The expected commute time is bounded from below by the total graph volume and the inverse of this optimal capacity:*

$$H(s, t) \geq \frac{vol(V)}{Z_{s,t}^*} \tag{103}$$

*By substituting the local Nodal Conductance $\Phi_{s,t}^*$, the expected commute time is bounded by the local isoperimetric ratio and the volume of the smaller partitioned component:*

$$H(s, t) \geq \frac{vol(V)}{\Phi_{s,t}^* \min(vol(V_s), vol(V_t))} \tag{104}$$

*Proof.* Let $G = (V, E)$ be a connected, unweighted graph. The relationship between the expected commute time of a random walk and the effective resistance $R_{s,t}$ between nodes $s$ and $t$ is established by the fundamental identity (Chandra et al., 1989):

$$H(s, t) = \text{vol}(V) R_{s,t} \tag{105}$$

To bound $R_{s,t}$, we utilise its explicit variational definition. The effective conductance between $s$ and $t$, which is the reciprocal of the effective resistance, is defined as the minimum Dirichlet energy over all real-valued

potential functions $\phi : V \to \mathbb{R}$ that satisfy the boundary conditions $\phi_s = 1$ and $\phi_t = 0$:

$$\frac{1}{R_{s,t}} = \min_{\phi \in \mathbb{R}^{|V|}} \left\{ \sum_{(u,v) \in E} (\phi_u - \phi_v)^2 \;\middle|\; \phi_s = 1, \phi_t = 0 \right\} \tag{106}$$

Because this formulation seeks an absolute minimum over all valid functions $\phi$, evaluating the Dirichlet energy functional at any specific valid test function provides a strict upper bound on the effective conductance.

By Theorem 3.2, the Nodal Tension LP provides an optimal integer solution $p^* \in \{0,1\}^{|V|}$, subject to the boundary conditions $p_s^* = 0$ and $p_t^* = 1$. We construct our test function $\phi^*$ by reversing these potentials:

$$\phi_v^* = 1 - p_v^* \quad \forall v \in V \tag{107}$$

We verify the boundary conditions for $\phi^*$: $\phi_s^* = 1 - 0 = 1$ and $\phi_t^* = 1 - 1 = 0$. Therefore, $\phi^*$ is a valid test function for the variational formulation. Evaluating the Dirichlet energy at $\phi^*$ yields:

$$\sum_{(u,v) \in E} (\phi_u^* - \phi_v^*)^2 = \sum_{(u,v) \in E} \left( (1 - p_u^*) - (1 - p_v^*) \right)^2 = \sum_{(u,v) \in E} (p_v^* - p_u^*)^2 \tag{108}$$

Because the LP variables are binary ($p_v^* \in \{0,1\}$), the squared difference across any edge is identical to the absolute difference: $(p_v^* - p_u^*)^2 = |p_v^* - p_u^*|$. Substituting this identity yields:

$$\sum_{(u,v) \in E} (\phi_u^* - \phi_v^*)^2 = \sum_{(u,v) \in E} |p_v^* - p_u^*| \tag{109}$$

By definition of the Nodal Tension LP formulation (Section 3), the sum of absolute potential differences across all edges is the optimal objective value, representing the minimum $s-t$ capacity $Z_{s,t}^*$. Therefore, the Dirichlet energy evaluated at our test function $\phi^*$ is equal to $Z_{s,t}^*$.

Because the minimum Dirichlet energy cannot exceed the energy evaluated at any specific valid test function, we establish the strict upper bound on the effective conductance:

$$\frac{1}{R_{s,t}} \leq Z_{s,t}^* \tag{110}$$

Inverting both sides yields the explicit lower bound on the effective resistance:

$$R_{s,t} \geq \frac{1}{Z_{s,t}^*} \tag{111}$$

Substituting this bound directly into Equation 105 yields the first inequality of the theorem:

$$H(s,t) = \text{vol}(V) R_{s,t} \geq \frac{\text{vol}(V)}{Z_{s,t}^*} \tag{112}$$

To translate this capacity bound into the domain of local graph geometry, we use the exact definition of local Nodal Conductance from Lemma 4.9, isolating the capacity term: $Z_{s,t}^* = \Phi_{s,t}^* \min(\text{vol}(V_s), \text{vol}(V_t))$. Substituting this expression into the denominator completes the proof:

$$H(s,t) \geq \frac{\text{vol}(V)}{\Phi_{s,t}^* \min(\text{vol}(V_s), \text{vol}(V_t))} \tag{113}$$

$\square$

### B.10 Proof of Corollary 4.11

**Corollary** (Restatement of Corollary 4.11). *Let $G = (V, E)$ be processed by a linear Message Passing Neural Network (MPNN) utilising normalised aggregation $X^{(k)} = D^{-1}AX^{(k-1)}$ for $K$ discrete layers. The Jacobian sensitivity of a target node $t$'s final representation to a source node $s$'s initial feature, denoted $\frac{\partial x_t^{(K)}}{\partial x_s^{(0)}}$, is proportional to the $K$-step random walk transition probability $(P^K)_{ts}$. By Theorem 4.10, the expected commute time dictates that the maximum of the directional hitting times is bounded from below by the local bottleneck capacity $Z_{s,t}^*$:*

$$\max(h(s,t), h(t,s)) \geq \frac{vol(V)}{2Z_{s,t}^*} = \frac{vol(V)}{2\Phi_{s,t}^* \min(vol(V_s), vol(V_t))} \tag{114}$$

*Therefore, if the network depth $K$ is instantiated such that $K < \frac{vol(V)}{2Z_{s,t}^*}$, the layer depth is less than the expected number of steps required for the message passing operator to traverse the minimal separating surface $C^*$ in the maximal direction. Because the available steps cannot satisfy the expected hitting time, the corresponding transition probability mass $(P^K)_{ts}$ is structurally constrained. This directly forces the Jacobian sensitivity $\frac{\partial x_t^{(K)}}{\partial x_s^{(0)}}$ toward zero, establishing that a shrinking local Nodal Conductance ($\Phi_{s,t}^* \to 0$) induces localised over-squashing independently of the global graph geometry.*

*Proof.* Let $G = (V, E)$ possess an adjacency matrix $A$ and a diagonal degree matrix $D$. Consider a linear Message Passing Neural Network (MPNN) where the initial node features $X^{(0)} \in \mathbb{R}^{|V| \times d}$ are iteratively updated. Using standard normalised mean aggregation, the feature matrix at layer $k$ is computed as:

$$X^{(k)} = D^{-1}AX^{(k-1)} \tag{115}$$

The linear operator $P = D^{-1}A$ is identical to the transition probability matrix of a discrete-time random walk on $G$, where the transition probability from $u$ to $v$ is $P_{uv} = \frac{1}{d_u}$. Unrolling the network for $K$ discrete layers yields the explicit closed-form relationship:

$$X^{(K)} = P^K X^{(0)} \tag{116}$$

To quantify the information flow from a specific source node $s$ to a target node $t$, we compute the Jacobian sensitivity of the target's final representation $x_t^{(K)}$ with respect to the source's initial features $x_s^{(0)}$. Taking the partial derivative of the linear system $X^{(K)} = P^K X^{(0)}$ isolates the corresponding scalar entry of the $K$-step transition matrix:

$$\frac{\partial x_t^{(K)}}{\partial x_s^{(0)}} = (P^K)_{ts} \cdot I_d \tag{117}$$

By definition of a Markov chain, the scalar $(P^K)_{ts}$ is the exact probability that a random walk originating at $s$ terminates at $t$ after $K$ steps.

For this transition probability $(P^K)_{ts}$ to capture significant mass, the architectural depth $K$ must accommodate the expected number of steps required to simply reach $t$ from $s$, defined as the hitting time $h(s,t) = \mathbb{E}[\min\{k \geq 0 \mid X_k = t\} \mid X_0 = s]$.

By Theorem 4.10, the expected commute time across the optimal Nodal Tension minimal surface $C^*$ establishes a strict lower bound on the sum of the directional hitting times:

$$h(s,t) + h(t,s) \geq \frac{vol(V)}{Z_{s,t}^*} \tag{118}$$

By the algebraic property of the arithmetic mean, the maximum of any two values is bounded below by their average. Applying this inequality directly to the directional hitting times yields:

$$\max(h(s,t), h(t,s)) \geq \frac{h(s,t) + h(t,s)}{2} \geq \frac{vol(V)}{2Z_{s,t}^*} \tag{119}$$

By substituting the isolated capacity $Z_{s,t}^* = \Phi_{s,t}^* \min(\mathrm{vol}(V_s), \mathrm{vol}(V_t))$ established in Lemma 4.9, we obtain the final threshold:

$$\max(h(s,t), h(t,s)) \geq \frac{\mathrm{vol}(V)}{2\Phi_{s,t}^* \min(\mathrm{vol}(V_s), \mathrm{vol}(V_t))} \tag{120}$$

Therefore, if the network hyperparameter $K$ is set such that $K < \frac{\mathrm{vol}(V)}{2Z_{s,t}^*}$, the depth of the MPNN is less than the expected number of steps required for a random walk to traverse the minimal surface $C^*$ in the maximal direction. Because the available layers $K$ cannot satisfy the expected hitting time, the probability mass $(P^K)_{ts}$ is structurally bounded, forcing the Jacobian sensitivity $\frac{\partial x_t^{(K)}}{\partial x_s^{(0)}}$ toward zero. This algebraic derivation isolates low local Nodal Conductance ($\Phi_{s,t}^* \to 0$) as the geometric driver of over-squashing. $\square$

### B.11 Proof of Lemma 4.12

**Lemma** (Restatement of Lemma 4.12). *Let $G = (V, E, w)$ be a weighted graph possessing a unique optimal Nodal Tension minimum $s - t$ cut, denoted $C^*$. Let $Z_{s,t}^*(w)$ be the optimal continuous objective value of the Nodal Tension LP as a function of the edge weight vector $w$. The partial derivative of the bottleneck capacity with respect to any individual continuous edge weight $w_{uv}$ is equal to the optimal integer tension variable $d_{uv}^*$:*

$$\frac{\partial Z_{s,t}^*}{\partial w_{uv}} = d_{uv}^* \tag{121}$$

*Consequently, by the integrality of the node potentials, the marginal sensitivity of the objective is binary and topologically localised exclusively to the minimal separating surface:*

$$\frac{\partial Z_{s,t}^*}{\partial w_{uv}} = \begin{cases} 1 & \text{if } (u,v) \in C^* \\ 0 & \text{if } (u,v) \notin C^* \end{cases} \tag{122}$$

*Proof.* Let the Nodal Tension LP be parameterised by the positive edge weight vector $w \in \mathbb{R}_{>0}^{|E|}$. The continuous optimisation problem evaluates $Z_{s,t}^*(w) = \min_{(p,d) \in \mathcal{X}} \sum_{(i,j) \in E} w_{ij} d_{ij}$, where the feasible region $\mathcal{X}$ is constrained by $d_{ij} \geq p_i - p_j$, $d_{ij} \geq p_j - p_i$, $p_s = 0$, and $p_t = 1$. Crucially, the feasible region $\mathcal{X}$ is independent of the edge weights $w$.

Assume the graph possesses a unique minimum $s - t$ cut $C^*$. By Theorem 3.1, the optimal vertex solution to the LP, denoted $x^* = (p^*, d^*) \in \mathcal{X}$, is therefore unique.

To derive the partial derivative with respect to an arbitrary edge weight $w_{uv}$ from first principles, we apply a positive infinitesimal perturbation $\epsilon > 0$ to $w_{uv}$. This yields a perturbed weight vector $w(\epsilon) = w + \epsilon \mathbf{e}_{uv}$, where $\mathbf{e}_{uv}$ is the standard basis vector for edge $(u,v)$. Let $x(\epsilon) = (p(\epsilon), d(\epsilon)) \in \mathcal{X}$ be the optimal solution under these perturbed weights, yielding the objective value $Z_{s,t}^*(w(\epsilon))$.

We first establish a strict upper bound on the objective's rate of change. Because the feasible region $\mathcal{X}$ is independent of $w$, the original optimal solution $x^*$ remains feasible under the perturbed weights $w(\epsilon)$. Evaluating the objective function at $x^*$ under the new weights $w(\epsilon)$ must yield a value greater than or equal to the true new minimum:

$$Z_{s,t}^*(w(\epsilon)) \leq \sum_{(i,j) \in E} w_{ij}(\epsilon) d_{ij}^* = \sum_{(i,j) \in E} w_{ij} d_{ij}^* + \epsilon d_{uv}^* = Z_{s,t}^*(w) + \epsilon d_{uv}^* \tag{123}$$

Algebraically rearranging this inequality isolates the upper limit for the difference quotient:

$$\frac{Z_{s,t}^*(w(\epsilon)) - Z_{s,t}^*(w)}{\epsilon} \leq d_{uv}^* \tag{124}$$

We next establish the corresponding lower bound. Because $x(\epsilon)$ is optimal for $w(\epsilon)$, evaluating the original objective at this perturbed solution $x(\epsilon)$ must yield a value greater than or equal to the original minimum

$Z^*_{s,t}(w)$:

$$Z^*_{s,t}(w) \leq \sum_{(i,j) \in E} w_{ij} d_{ij}(\epsilon) = \sum_{(i,j) \in E} w_{ij}(\epsilon) d_{ij}(\epsilon) - \epsilon d_{uv}(\epsilon) = Z^*_{s,t}(w(\epsilon)) - \epsilon d_{uv}(\epsilon) \tag{125}$$

Rearranging this inequality isolates the lower limit for the difference quotient:

$$d_{uv}(\epsilon) \leq \frac{Z^*_{s,t}(w(\epsilon)) - Z^*_{s,t}(w)}{\epsilon} \tag{126}$$

Combining Equation 124 and Equation 126 traps the difference quotient between the two primal states:

$$d_{uv}(\epsilon) \leq \frac{Z^*_{s,t}(w(\epsilon)) - Z^*_{s,t}(w)}{\epsilon} \leq d^*_{uv} \tag{127}$$

Because the original optimal vertex $x^*$ is unique, standard LP stability dictates that for an infinitesimally small perturbation $\epsilon \to 0$, the optimal vertex remains unchanged. Therefore, $\lim_{\epsilon \to 0} d_{uv}(\epsilon) = d^*_{uv}$. Applying the Squeeze Theorem as $\epsilon \to 0$ evaluates the exact continuous partial derivative:

$$\frac{\partial Z^*_{s,t}}{\partial w_{uv}} = \lim_{\epsilon \to 0} \frac{Z^*_{s,t}(w(\epsilon)) - Z^*_{s,t}(w)}{\epsilon} = d^*_{uv} \tag{128}$$

Having derived the continuous derivative, we finally map it to the discrete graph topology. By Theorem 3.2, the optimal potentials $p^*$ partition the graph into a source component ($p^* = 0$) and a target component ($p^* = 1$). To minimise the tension across positive edge weights, the inequality constraints must bind tightly, enforcing $d^*_{uv} = |p^*_u - p^*_v|$.

We evaluate this absolute potential difference for the two possible topological states of the edge $(u, v)$:

1. **Cut Edges:** If the edge spans the $s - t$ partition ($(u, v) \in C^*$), its endpoints reside in opposite components. The potential difference evaluates to $|0 - 1| = 1$ (or $|1 - 0| = 1$), yielding $d^*_{uv} = 1$.

2. **Internal Edges:** If the edge does not span the partition ($(u, v) \notin C^*$), both endpoints reside in the same component. The potential difference evaluates to $|0 - 0| = 0$ (or $|1 - 1| = 0$), yielding $d^*_{uv} = 0$.

Substituting these exact algebraic bounds yields the final piecewise derivative:

$$\frac{\partial Z^*_{s,t}}{\partial w_{uv}} = \begin{cases} 1 & \text{if } (u, v) \in C^* \\ 0 & \text{if } (u, v) \notin C^* \end{cases} \tag{129}$$

This first-principles derivation proves that the continuous LP possesses zero marginal sensitivity to weight perturbations occurring anywhere outside the strict structural bottleneck. □

## B.12 Proof of Theorem 4.13

**Theorem** (Restatement of Theorem 4.13). *Let $G = (V, E, w)$ be a weighted graph. The optimal Nodal Tension capacity $Z^*_{s,t}(w)$ is a globally concave, piecewise linear function with respect to the continuous edge weight vector $w \in \mathbb{R}^{|E|}_{>0}$. At points of topological degeneracy where multiple distinct minimum $s - t$ cuts exist with identical capacity, the objective function is non-differentiable. To evaluate the sensitivity at these degenerate intersections, we employ the Clarke subdifferential (Clarke, 1990). For a concave, piecewise linear function, the Clarke subdifferential at a non-differentiable point is defined as the convex hull of all limit gradients obtained by approaching the point from surrounding domains where the function is differentiable. Let $\mathcal{C}^*$ denote the set of all valid minimum $s - t$ cuts actively defining the capacity at this parameterisation $w$. By Lemma 4.12, the exact limit gradient approaching from a domain where a cut $C^* \in \mathcal{C}^*$ is uniquely optimal is its binary tension vector. Therefore, the exact Clarke subdifferential of the capacity, denoted $\partial Z^*_{s,t}(w)$,*

evaluates to the convex hull of the binary optimal tension vectors corresponding to every active minimum cut in $\mathcal{C}^*$:

$$\partial Z_{s,t}^*(w) = Conv\left(\left\{d^{C^*} \in \{0,1\}^{|E|} \mid C^* \in \mathcal{C}^*\right\}\right) \tag{130}$$

where $d^{C^*}$ is the binary indicator vector of the specific cut $C^*$, such that $d_{uv}^{C^*} = 1$ if $(u,v) \in C^*$, and $0$ otherwise. Consequently, any valid subgradient $g \in \partial Z_{s,t}^*(w)$ is bounded ($g_{uv} \in [0,1]$) and structurally sparse, assigning non-zero sensitivity exclusively to edges that actively participate in at least one optimal separating surface.

*Proof.* Let the continuous Nodal Tension LP be parameterised by the edge weight vector $w \in \mathbb{R}_{>0}^{|E|}$. The continuous optimisation problem evaluates:

$$Z_{s,t}^*(w) = \min_{x \in \mathcal{X}} w^T d \tag{131}$$

where $x = (p,d) \in \mathbb{R}^{|V|+|E|}$ is the combined primal variable vector, and $\mathcal{X}$ is the polyhedral feasible region defined by the topological boundary conditions ($p_s = 0, p_t = 1$) and the tension constraints ($d_{ij} \geq p_i - p_j$, $d_{ij} \geq p_j - p_i$). Because the constraints defining $\mathcal{X}$ do not depend on the objective coefficients $w$, the polyhedron $\mathcal{X}$ is fixed and structurally invariant to weight perturbations.

By the fundamental theorem of linear programming, if the optimal value is finite, the minimum is achieved at one or more extreme points (vertices) of the polyhedron $\mathcal{X}$. Let $\mathcal{V}(\mathcal{X}) = \{v_1, v_2, \ldots, v_k\}$ denote the finite set of all extreme points of $\mathcal{X}$. The objective function can thus be reformulated as a finite pointwise infimum over these vertices:

$$Z_{s,t}^*(w) = \min_{v \in \mathcal{V}(\mathcal{X})} w^T d^{(v)} \tag{132}$$

where $d^{(v)}$ is the tension vector corresponding to the specific vertex $v$. Because $Z_{s,t}^*(w)$ is the pointwise minimum of a finite set of affine (linear) functions of $w$, it is analytically guaranteed to be a globally concave and piecewise linear continuous function.

Let $w$ be a specific weight vector corresponding to a point of topological degeneracy. At this point, the optimal minimum is achieved simultaneously by a subset of active vertices. Let $\mathcal{V}^*(w) \subseteq \mathcal{V}(\mathcal{X})$ denote this active set:

$$\mathcal{V}^*(w) = \left\{v \in \mathcal{V}(\mathcal{X}) \mid w^T d^{(v)} = Z_{s,t}^*(w)\right\} \tag{133}$$

By Theorem 3.2, the totally unimodular structure of $\mathcal{X}$ guarantees that every optimal vertex $v \in \mathcal{V}^*(w)$ is integer, corresponding to the binary indicator vector of a valid minimal separating surface. Therefore, the set of active vertices maps bijectively to the set of active optimal minimum $s-t$ cuts, $\mathcal{C}^*$. For every $v \in \mathcal{V}^*(w)$, the corresponding tension vector $d^{(v)}$ is $d^{C^*}$ for some $C^* \in \mathcal{C}^*$.

Because the objective function $Z_{s,t}^*(w) = \min_{v \in \mathcal{V}(\mathcal{X})}(w^T d^{(v)})$ is concave and piecewise linear, it is non-differentiable at $w$ if $|\mathcal{V}^*(w)| > 1$. To evaluate the gradient, we apply Danskin's Theorem and the definition of the Clarke subdifferential for the pointwise infimum of linear functions (Clarke, 1990). The exact generalised subdifferential $\partial Z_{s,t}^*(w)$ evaluates to the convex hull of the standard gradients of the active linear functions:

$$\partial Z_{s,t}^*(w) = Conv\left(\left\{\nabla_w(w^T d^{(v)}) \mid v \in \mathcal{V}^*(w)\right\}\right) \tag{134}$$

Evaluating the standard gradient of the linear form yields the constant tension vector: $\nabla_w(w^T d^{(v)}) = d^{(v)}$. Substituting the bijective mapping $d^{(v)} \equiv d^{C^*}$ into the convex hull yields the exact topological subdifferential:

$$\partial Z_{s,t}^*(w) = Conv\left(\left\{d^{C^*} \in \{0,1\}^{|E|} \mid C^* \in \mathcal{C}^*\right\}\right) \tag{135}$$

By the definition of a convex hull, any valid subgradient $g \in \partial Z_{s,t}^*(w)$ must be expressible as a convex combination of the active extreme points:

$$g = \sum_{C^* \in \mathcal{C}^*} \alpha_{C^*} d^{C^*}, \quad \text{subject to } \sum \alpha_{C^*} = 1, \quad \alpha_{C^*} \geq 0 \tag{136}$$

This algebraically forces two strict structural bounds on the backpropagated subgradient:

1. **Boundedness:** Because every scalar entry $d_{uv}^{C^*} \in \{0, 1\}$, any convex combination $g_{uv}$ must satisfy $g_{uv} \in [0, 1]$. Exploding gradients are impossible.

2. **Sparsity:** Consider an edge $(u, v)$ that is structurally disjoint from all optimal bottlenecks. By definition, $(u, v) \notin C^*$ for all $C^* \in \mathcal{C}^*$. Therefore, the corresponding entry $d_{uv}^{C^*} = 0$ in every active vector. The convex combination must evaluate to $g_{uv} = \sum \alpha_{C^*}(0) = 0$.

Thus, the exact Clarke subdifferential is bounded, piecewise linear, and sparse, algebraically confining all non-zero gradient mass exclusively to the minimal separating surfaces. $\square$

