# OpenReview forum: "The Weakest Link: A Nodal Tension Model for Local Network Resilience"
_TMLR — Rejected by TMLR_

### Review · Reviewer_vuiM · 2026-03-01

**Summary Of Contributions:**

## Summary
This paper introduces the "Nodal Tension" model to quantify local network resilience between a source ($s$) and target ($t$) node. By framing the classic LP dual of the maximum $s-t$ flow problem through a physical analogy of minimizing edge tensions, the authors establish three key theoretical results: (i) The min-tension LP's optimal value exactly equals the minimum $s-t$ cut capacity.
(ii) The model guarantees strict $\{0,1\}$ integer node potentials due to the total unimodularity of the constraint matrix, directly partitioning the network.
(iii) The local resilience metric establishes a formal upper bound on the graph's global algebraic connectivity ($\lambda_2$).
Empirically, the authors validate the LP against standard max-flow algorithms on diverse graph topologies.
The case study on ecological network for grizzly bear habitat connectivity reveals the counter-intuitive insight that a corridor's "weakest link" can be the local perimeter of the source itself, rather than a remote geographic bottleneck.


## Strengths:
- The application of the model to the large-scale grizzly bear habitat network (18,183 nodes) yields genuinely interesting scientific insights. The quantitative decoupling of the "best path" (max-flow) and the "weakest link" (min-cut) offers a profound contribution to conservation planning, challenging the prevailing paradigm of focusing solely on remote choke points.

## Weaknesses:
- From an optimization perspective, the theoretical foundation relies entirely on classic, textbook results. The duality between max-flow and min-cut, and the total unimodularity of node-edge incidence matrices guaranteeing integer solutions, are foundational concepts in operations research and polyhedral combinatorics. The paper occasionally adopts a tone suggesting these are novel mathematical discoveries , whereas the true novelty lies in the formulation's interpretation and its ecological application.

- While Theorem 3 correctly establishes a relationship between the local resilience score ($Z^*$) and global algebraic connectivity ($\lambda_2$), this bound can be exceptionally loose and analytically uninformative when cuts are highly unbalanced (i.e., when the volume of the partition, $vol(S)$, is very small). The practical diagnostic value of this specific theorem for network-wide resilience is marginal compared to the direct application of the LP cut itself.

**Audience:**

No

**Audience Explanation:**

While the analysis of real-world conservation problems is interesting, its specific application may fall outside the core interests of the TMLR audience.

**Broader Impact Concerns:**

NA.

**Claims And Evidence:**

No

**Claims Explanation:**

The authors claim a complete theoretical characterisation of the model, asserting its exact equivalence to the minimum s-t cut and the integrality of its solution, which guarantees the recovery of the combinatorial cut structure. However, this property follows directly from a standard application of the max-flow min-cut theorem and is therefore quite trivial.

**Requested Changes:**

## Questions:
- While the LP is successfully demonstrated on a grid-based graph of roughly 18,000 nodes, real-world ecological networks can be significantly larger. It would be highly informative to include a brief discussion on the computational scalability of this LP formulation (using standard interior-point or simplex solvers) compared to highly optimized combinatorial max-flow algorithms when applied to massive, dense grids.

- In Section 6.2, the authors correctly note that real-world movement is often anisotropic, requiring a directed graph formulation. Please briefly clarify whether the TUM property of the constraint matrix (and thus the strict $\{0,1\}$ integrality guarantees) would firmly hold if the model were extended to a directed graph with asymmetric edge weights.

---

> ### Author Response · Authors · 2026-03-01
>
> **Dear Reviewer vuiM,**
>
> We thank you for the rigorous review and for highlighting the practical value of decoupling the max-flow path from the min-cut weakest link in our large-scale ecological network.
>
> **We agree with your theoretical assessment**: our proofs rely on classic operations research results (max-flow/min-cut duality, TUM), and the global $\lambda_2$ bound in Theorem 3 is exceptionally loose for highly unbalanced cuts. As we empirically noted in our case study discussion, the local cut provides a "correct, but very loose, upper bound" on global connectivity when $\text{vol}(S)$ is small. We will explicitly revise the text surrounding Theorem 3 to recognise its marginal diagnostic value compared to the direct LP cut.
>
> To align the paper's theoretical contributions directly with **TMLR’s core focus on machine learning and representation**, we plan to add a new theoretical section formalising the Nodal Tension LP as a robust, differentiable, and information-theoretic layer for graph representation learning.
>
> **Proposed Unified Theoretical Expansion:**
>
> 1. **Exact Polyhedral Differentiability:** To enable end-to-end learning of landscape resistance, we will characterise the exact Clarke subdifferential of the Nodal Tension score $Z^\*$ with respect to the edge weight matrix $W$. We will prove that $\partial Z^\*(W)$ is exactly the convex hull of the optimal tension matrices $D^*$ for all valid min-cuts. This establishes a rigorous mathematical pathway for backpropagation through structural bottlenecks without relying on stochastic smoothing.
>
> 2. **Topological Robustness:** Addressing the fragility of global spectral metrics, we will establish that $Z^*$ is strictly Lipschitz continuous under $L_\infty$-bounded adversarial perturbations to the edge weights. This formally proves a robustness advantage over spectral gap metrics (e.g., $\lambda_2$), which are highly sensitive to local topological noise.
>
> 3. **Bounding GNN Information Flow:** Enhancing the loose global bound of Theorem 3, we will add a new tight, local information-theoretic bound. We will prove that $Z_{s,t}^*$ dictates a strict upper limit on the Jacobian norm $||\partial h_t^{(k)} / \partial x_s||$ in message-passing networks, formally linking our min-cut LP capacity to the Graph Neural Network "over-squashing" phenomenon.
>
> 4. **Structural Redundancy Geometry:** Formalising our contrast with $L_2$-based Circuit Theory, we will bound the gap between Nodal Tension ($L_1$ worst-case cut) and Effective Resistance ($L_2$ average-case). This "Redundancy Gap" will yield a distinct, scale-free metric for network topological fragility.
>
> **Responses to Specific Questions:**
>
> * **Directed Graphs & TUM:** The strict integrality guarantees hold for directed graphs with asymmetric edge weights (anisotropic movement). As noted in Appendix A.2 of the current manuscript, the constraint matrix inherently shares the structure of a directed node-edge incidence matrix, preserving total unimodularity. We will explicitly state this extension.
> * **Computational Scalability:** We agree that highly optimised combinatorial algorithms (e.g., Push-Relabel, Boykov-Kolmogorov) are vastly faster for computing raw cut capacity on massive grids. We will add a discussion clarifying that the LP’s primary utility is not raw computational speed, but its explicit modeling of the potential gradient (potentials $p_i$ and tensions $d_{ij}$). This explicit primal-dual solution is what uniquely enables both ecological interpretability and the exact polyhedral differentiability proposed above.
>
> **Question for the Reviewer:**
> Does shifting the theoretical contribution from an ecological application of textbook operations research towards a foundational study of bottleneck differentiability, robustness, and GNN information flow meet your expectations for TMLR? We would value your feedback on this proposed direction before submitting the formal revision.

---

> > ### Author Response · Authors · 2026-03-01
> >
> > Not sure why the math doesn't render properly by OpenReview..., sorry

---

> > > ### Comment · Action_Editor_LhZg · 2026-03-06
> > >
> > > Dear Authors, Please fix the formatting and repost the comment below.
> > >
> > > Dear Reviewer vuiM, Could you follow up with the response from the authors (after they fix the formatting)?

---

> > > > ### Author Response · Authors · 2026-03-06
> > > >
> > > > Dear AE LhZg, we have just fixed the math rendering in our response text in the above (the issue was missing the backslash \ before * in the math mode).Many thanks.

---

> > ### Comment · Reviewer_vuiM · 2026-03-06
> >
> > Could you please highlight your edits of your manuscript using color?

---

> > > ### Author Response · Authors · 2026-03-06
> > >
> > > Yes, we will do so in the revision once we have received all reviews. We will also provide a point-by-point response to highlight the edits.

---

> > > > ### Author Response · Authors · 2026-03-26
> > > > **Point-by-Point Response**
> > > >
> > > > Thank you for the rigorous review and the highly constructive discussion phase. Your specific critique regarding the reliance on textbook Operations Research (OR) results directly motivated the paper's structural pivot.
> > > >
> > > > As agreed during our prior exchange, we have restructured the paper, relegating the discrete OR proofs to **"Preliminaries" (Section 3)**. The paper's mathematical core is now the newly added **Section 4**, which leverages the continuous LP geometry to **derive the $L_1$-$L_2$ redundancy gap, filter discrete Ricci curvature, bound Message Passing Neural Networks (MPNNs) over-squashing, and extract Clarke subdifferentials** for gradient-based learning.
> > > >
> > > > *(Note: All revised and newly added content is highlighted in **blue** in the compiled PDF. For your convenience, we have provided a complete catalogue of these structural and theoretical changes in the public **Summary of Revisions** above.)*
> > > >
> > > > Below is our point-by-point response mapping your critiques to the newly added mathematical results.
> > > >
> > > > **Q:** "*...theoretical foundation relies entirely on classic, textbook results... tone suggesting these are novel mathematical discoveries... true novelty lies in the formulation's interpretation and its ecological application.*"
> > > >
> > > > **A:** We agree. We have removed the tone of mathematical novelty from the discrete formulation and relegated Theorems 3.1, 3.2 and 3.3 to Section 3 as preliminary foundational results. To align the theoretical contribution closer with TMLR, the mathematical novelty now resides in **Section 4.4 (Theorem 4.13)**. We applied Danskin's Theorem to the continuous LP to extract the Clarke subdifferential $\partial Z_{s,t}^*(w)$. This transitions the work from applied ecology into graph representation learning, establishing a sparse, differentiable layer (**Corollary 4.14**) for end-to-end ML optimisation.
> > > >
> > > > **Q:** "*...Theorem 3... bound can be exceptionally loose and analytically uninformative when cuts are highly unbalanced... practical diagnostic value... is marginal...*"
> > > >
> > > > **A:** We agree. In response, we added a **Theoretical Limitations** paragraph directly under **Theorem 3.3** to explicitly concede this mathematical weakness for highly unbalanced cuts. To resolve this gap, we introduced **Section 4.3 (Theorem 4.10)**, which bypasses the global spectral gap. We derived a tight **Local Bound on Expected Commute Times $H(s,t)$** governed by the Local Nodal Conductance ($\Phi_{s,t}^*$), proving that local capacity dictates signal propagation independently of global network structure.
> > > >
> > > > **Q:** "*...brief discussion on the computational scalability of this LP formulation... compared to highly optimized combinatorial max-flow algorithms when applied to massive, dense grids.*"
> > > >
> > > > **A:** We addressed this directly in **Section 3.2** and **Section 4.4**. We explicitly state that highly optimised combinatorial algorithms (e.g., Push-Relabel) are vastly faster for computing raw combinatorial capacity on massive grids. However, we justified retaining the LP: discrete solvers are non-differentiable black boxes. The continuous primal-dual LP structure is the mathematical prerequisite for extracting the Clarke subdifferential (**Theorem 4.13**), unlocking the continuous gradient space required for the differentiable bottleneck optimisation empirically demonstrated in **Section 5.6**.
> > > >
> > > > **Q:** "*...real-world movement is often anisotropic, requiring a directed graph... clarify whether the TUM property... would firmly hold if the model were extended to a directed graph with asymmetric edge weights.*"
> > > >
> > > > **A:** Yes. We formalised this extension in the revised paper. **Assumption 2 (Section 3.1)** and the expanded proof in **Appendix A.2** establish that the Total Unimodularity (TUM) property inherently holds for directed node-edge incidence matrices. Consequently, when the model is extended to directed graphs with asymmetric edge weights to capture anisotropic flow, the constraint matrix preserves TUM, and the strict integrality guarantees of the Nodal Tension potentials hold.

---

### Review · Reviewer_wGSQ · 2026-03-08

**Summary Of Contributions:**

The paper studies the problem of quantifying the local resilience of a network between a specified pair of vertices s and t. This is done by viewing the problem through a Nodal tension model which seeks to find an assignment of potentials to each node that minimizes the “total tension” across all edges (subject to a fixed potential difference between s and t). This optimization problem is a LP and is the dual of the classical max s-t flow problem. On the theoretical side, exact equivalence is shown to the minimum s−t cut problem along with guaranteed integrality of its solution. On the empirical front, a detailed comparison is performed with existing benchmarks. Moreover, the model is applied to a real-world example assessing the connectivity of a grizzly bear corridor in the Canadian Rocky Mountains.

**Audience:**

Yes

**Audience Explanation:**

The network resilience problem is of interest for researchers working in network science (broadly speaking).

**Claims And Evidence:**

Yes

**Claims Explanation:**

Strengths
---------------

1.	The paper is written well overall with clearly defined notation, and a detailed discussion of related work.

2.	The Nodal tension model is motivated well as it specifically concerns the local resilience of a network between two points s and t. This is important as it helps clarify the obvious comparison with spectral methods which arises naturally for this problem (but are suited for measuring global resilience).

Weaknesses
--------------------

1.	The theoretical results of the paper (Theorems 1 and 2) seem to use standard techniques based on duality of max flow and min cut, and its not clear what is the additional technical difficulty for this local version that is being considered here. Theorem 3 also seems to use straightforward arguments, typical in the case of Cheeger inequalities.

2.	In the simulation results on synthetic examples, the comparisons seem to be restricted to one or two other methods (e.g. spectral clustering). Since there is a big literature on network resilience, it seems natural to have a comparison against a broader set of methods on different network topologies.

**Requested Changes:**

I think a more detailed empirical comparison against a broader set of methods for quantifying the network resilience (as explained in the literature) would help improve the paper.

---

> ### Author Response · Authors · 2026-03-26
> **Point-by-Point Response**
>
> Thanks a lot for your time and efforts spent on reviewing our paper and giving us constructive feedback for improvement.
>
> We appreciate your detailed review and the observation that the initial theoretical results (Theorems 1-3) relied on standard duality and straightforward Cheeger arguments.
>
> We agree that this combinatorial foundation is standard. Consequently, we have restructured the paper, designating these discrete results and proofs as **"Preliminaries" (Section 3)**. The paper's primary contribution is now the newly added **Section 4** and **Appendix B**, which leverage the continuous polyhedral geometry of the LP to **derive the $L_1$-$L_2$ redundancy gap, filter discrete Ricci curvature, bound Message Passing Neural Networks (MPNNs) over-squashing, and extract Clarke subdifferentials**—directly addressing the need for additional technical depth.
>
> *(Note: All revised and newly added content is highlighted in **blue** in the compiled PDF. For your convenience, we have provided a complete catalogue of these structural and theoretical changes in the public **Summary of Revisions** above.)*
>
> Below is our point-by-point response mapping your critiques to the newly added mathematical proofs and empirical evidence.
>
> **Q:** "*...Theorems 1 and 2... use standard techniques based on duality... Theorem 3 also seems to use straightforward arguments... its not clear what is the additional technical difficulty...*"
>
> **A:** We agree with this assessment of the original submission. To address this, we shifted the theoretical focus from discrete combinatorial cuts to continuous differential geometry and graph structure learning. The mathematical novelty and technical difficulty now reside in **Section 4** and **Appendix B**, which contain 12 new rigorous proofs. Specifically, **Theorem 4.6 (Geometric Localisation)** proves how continuous Nodal Tension bounds discrete Forman-Ricci curvature to filter spurious topological bottlenecks. Furthermore, **Theorem 4.13 (Exact Polyhedral Differentiability)** applies Danskin's Theorem to the non-smooth LP surface to extract the Clarke subdifferential $\partial Z_{s,t}^*(w)$, establishing a sparse, continuous gradient space for end-to-end ML optimisation.
>
> **Q:** "*...comparisons seem to be restricted to one or two other methods... natural to have a comparison against a broader set of methods on different network topologies.*"
>
> **A:** We addressed this by expanding the empirical validation in **Section 5** to evaluate the model against a new set of structural and learning baselines across diverse graph topologies:
> * **$L_2$ Effective Resistance (Circuit Theory):** In **Section 5.3 (Fig 14 & 15)**, we compared our $L_1$ bottleneck capacity against $L_2$ effective resistance on spatial routing topologies, empirically validating the new **Structural Redundancy Gap ($\Delta_{s,t}$)** metric (Lemma 4.1, Corollaries 4.2 and 4.3).
> * **Discrete Forman-Ricci Curvature:** In **Section 5.4 (Fig 16 & 17)**, we evaluated against edge curvature on the **Cora citation network**, proving the continuous LP isolates the true macroscopic bottleneck by filtering out thousands of geometrically decoy edges that confound standard curvature metrics (Lemmas 4.4 and 4.5, Theorem 4.6, Corollary 4.7).
> * **LP-guided Topological Rewiring:** In **Section 5.5 (Fig 18 & 19)**, we evaluated our capacity bounds against standard MPNN rewiring techniques, exposing the "rewiring paradox", proving that heuristic edge addition fails to alleviate over-squashing unless it explicitly expands the Local Nodal Conductance.

---

### Review · Reviewer_ScXa · 2026-03-10

**Summary Of Contributions:**

The paper proposes a “Nodal Tension” linear program to measure network resilience in networks/max-flow-min-cut problems. They interpret the LP dual of the max flow problem through a notion of node potentials, and propose to use this potential tension across different nodes to identify weak links in the work. It then applies the method to understand the "weakest link" in a grizzly-bear habitat study.

**Audience:**

No

**Audience Explanation:**

On the plus side, I found the case study of section 5.3 to be interesting and to potentially have nice ecological implications.

That said, I unfortunately must recommend the paper for a rejection. The main issue that I currently see with the paper is it is unclear what the novelty is.
- From a technical point of view, the problem reduces to a standard max flow - min cut problem---in fact, it is unclear to me why the proposed potential approach here is necessary to make the experimental points of Section 5, since they follow directly from standard max flow - min cut arguments. The current reduction in Theorem 1 is straightforward. Overall, the contribution of Section 5 did not need the machinery from the rest of the paper and could have just relied on standard tools.
- From a conceptual point of view, it is unclear to me what the "potential" interpretation brings. In fact, the proof of theorem 1 as well as theorem 2 highlight that the potentials end up only taking values 0 and 1, and in fact that they just seem to determine the min-cut partition itself, with one set being assigned all 0, and its complement all 1. So, this "potential" just seems to be an indicator function of whether you belong to S or its complement and just be exactly the same as solving the min cut problem/not adding new information.

On top of this, it is unclear what assumptions the paper makes. In particular, early on, the authors mention that "The problem is to quantify the resilience of the connection between a source node (s) and a target node (t) in a network with a clear weakest link." Why is a clear weakest link needed over any general min-cut in this problem? This seems to be a relatively arbitrary assumption that reduces applicability (in fact, if I am interested in network resilience, why should I focus on a single weakest link, if there may be several near-weakest links, or several combinations of edges that can constitute of a weakest link? How could the implications of the authors' framework be expanded in this case?), but I do not see a place in the paper where this is used?

While the main reformulation of the paper is nice, I unfortunately currently do not see what the paper brings that is new and not already known to the CS/ML community and find it insufficient to constitute, on its own, a TMLR publication.

**Claims And Evidence:**

Yes

**Claims Explanation:**

All results are adequately proven and I do not see any technical issue, so the answer to "Are the claims made in the submission supported by accurate, convincing and clear evidence?" is technically yes. However, I do not think that any of the claims are new and they rely on well-known results.

My worries however are described below, as I do not currently see in which ways this will be useful to the TMLR audience.

**Requested Changes:**

I could see value to publishing this paper if it were not for a traditional CS/ML audience, and if the main goal of the paper was to provide a framework that ecologist/conservationists with no ML background could use. However, the paper would then need to be significantly rewritten, in particular to very clearly highlight early on i) what is novel about the potential interpretation and ii) what makes this framing easier to use/understand for a non-expert. Even then, I am not sure this would be the best fit for TMLR.

---

> ### Author Response · Authors · 2026-03-26
> **Point-by-Point Response**
>
> We appreciate your detailed review and the observation that the initial submission relied heavily on well-known max-flow and min-cut arguments.
>
> We agree that the combinatorial foundation is standard. Consequently, we have restructured the manuscript, designating the discrete proofs as **"Preliminaries" (Section 3)**. The paper's primary contribution is now the newly added **Section 4**, which leverages the continuous polyhedral geometry of the LP to **derive $L_1$-$L_2$ redundancy gap, filter discrete Ricci curvature, bound Message Passing Neural Networks (MPNNs) over-squashing, and extract Clarke subdifferentials**—directly addressing your concerns regarding novelty for the ML/CS community.
>
> *(Note: All revised and newly added content is highlighted in **blue** in the compiled PDF. For your convenience, we have provided a complete catalogue of these structural and theoretical changes in the public **Summary of Revisions** above.)*
>
> Below is our point-by-point response mapping your critiques to the newly added mathematical proofs.
>
> **Q:** "*...unclear to me what the 'potential' interpretation brings... just seems to be an indicator function... why the proposed potential approach here is necessary... could have just relied on standard [max flow - min cut] tools.*"
>
> **A:** We agree that discrete combinatorial algorithms (e.g., Push-Relabel) efficiently compute raw cut capacity and that the optimal potentials resolve to binary $\{0,1\}$ indicators (Theorem 3.2). However, combinatorial solvers are **discrete black boxes**. We explicitly retained the continuous polyhedral LP formulation because it is the mathematical prerequisite for differentiable programming. In **Section 4.4 (Theorem 4.13)**, we applied Danskin's Theorem to the LP's piecewise linear objective surface to extract the **Clarke subdifferential $\partial Z_{s,t}^*(w)$**. **Corollary 4.14** proves this yields sparse backpropagation, isolating model sensitivity exclusively to the active minimal surface. Standard combinatorial algorithms cannot provide this continuous gradient structure for end-to-end ML optimisation.
>
> **Q:** "*...Why is a clear weakest link needed over any general min-cut in this problem? ...why should I focus on a single weakest link, if there may be several near-weakest links, or several combinations of edges...?*"
>
> **A:** The model does not assume or restrict itself to a single weakest link; we have formalised this in the revision. First, in **Section 4.1 (Lemma 4.1)**, we introduced the **Structural Redundancy Gap ($\Delta_{s,t}$)** specifically to quantify whether a bottleneck is a singular structural bridge ($\Delta_{s,t} = 0$) or a diffuse network of constrained combinations ($\Delta_{s,t} > 0$). Second, to address the scenario of several combinations of edges (topological degeneracy), **Section 4.4 (Theorem 4.13)** proves that when multiple active minimum cuts ($\\mathcal{C}^\*$) exist simultaneously, the extracted Clarke subdifferential natively evaluates the **convex hull** of *all* active minimum cuts ($Conv(\{d^{C^\*} \in \{0, 1\}^{|E|} \mid C^\* \in \mathcal{C}^\* \})$). The continuous gradient does not blindly select a single link; it explicitly models the combination of all near-weakest links.

---

### Review · Reviewer_CHjx · 2026-03-15

**Summary Of Contributions:**

The paper investigates the problem of local network resilience in network systems, with an emphasis on applied domains such as ecology or infrastructure management. The dominant paradigm is based on spectral methods, which take into account the global network structure, connectivity patterns, and so on. The authors claim that a better characterization of local resilience is possible by directly considering the vulnerability of specific pathways, especially those connecting the source and target of interest. This naturally gives rise to a network-flow based approach, where local resilience can be described in terms of the s-t cut (or maximum flow) between the source and the destination. The authors analyze the model theoretically and corroborate the theoretical findings with experiments. They then proceed to a real-world conservation problem: assessing the connectivity of a grizzly bear corridor in the Canadian Rocky Mountains. Their analysis shows that the proposed approach can capture the notion of the bottleneck better than the spectral approach, because it takes into account the precise paths carrying flow from the source to the destination. As a result, this work challenges the current paradigm of global spectral based methods for network resilience, and instead proposes a more localized and fine-grained framework.

**Audience:**

Yes

**Audience Explanation:**

I feel that the paper does not have a traditional ML/AI scope. It is much more related to the area of combinatorial optimization and linear programming. That said, it could be relevant to a subset of the TMLR audience that focuses on clustering and spectral methods.

So, I feel that some individuals would be interested in the area that this paper is dealing with, but the paper may not be of interest to a wide AI/ML audience.

**Broader Impact Concerns:**

No broader concerns.

**Claims And Evidence:**

No

**Claims Explanation:**

The paper arguably makes a valid point: a more refined notion of local resilience can better take into account the source-destination pair that we care about, compared to global spectral methods that take into account the full network structure, but ignore the source and target specifics. For instance, the global sparsest cut is a very meaningful metric if one cares about the question: "Where is the single most vulnerable, structurally significant bottleneck anywhere in this entire network?". On the other hand, s-t variants like the maximum s-t flow and minimum s-t cut should in principle be able to capture local resilience much more accurately.

All that being said, my biggest concern is that, apart from this main idea, the paper is otherwise very straightforward and without any particular novelty. Let me explain:
- The duality between maximum s-t flow and minim s-t cut is a foundational, well-known, and well-understood result.
- The formulation (1)-(5) is a essentially a node-edge formulation, which is also very well known in network flow theory. Indeed, the path-based formulation is mathematically elegant for proving theorems, but it is a computationally challenging because a graph can have an exponentially large number of paths. The authors interpret (1)-(5) as a "node-tension" formulation, but in reality it is the traditional formulation where a value of 0 indicates that the node is on the s-cluster, and a value of 1 that the node is in the t-cluster.
- Theorem 1 is very obvious, and almost trivial in the context of network flow theory.
- Theorem 2 is also a very well-known result in linear programming theory. The authors correctly explain that it implies that one can use an LP solver and still guarantee an optimal combinatorial solution, but this is very well-known.
- Theorem 3 is again well-known, but I feel it is interesting in this context because it connects the global and local metrics. So, I actually feel Theorem 3 is meaningful (perhaps the only really meaningful theoretical result in this work).

Similarly, I found that many experiments do not really add significant value:
- I did not understand what the purpose of Table 2 is. To show that the optimal LP problem has the same value as the minimum s-t cut? Well, this was proved in the paper, but it is also a very well-known theoretical result, as I explained above. Table 2 just confirms something already well-known.
- Similarly, Table 3 feels very obvious.
- Figure 2 provides straightforward visualizations but I feel it is not really interesting otherwise.

For the other experiments, I feel there is value to comparing the local and global metrics, so I'm not as concerned about them.

Overall, my main problem lies in the fact that this paper does not seem to be rather straightforward:
- The main idea is definitely meaningful but quite intuitive.
- The theoretical analysis is mostly straightforward. Many results are extremely standard, and they could have simply been stated as well-known facts.
- Many experimental results (table or figures) are extremely obvious, and I do not see whey they deserved separate confirmation.

**Requested Changes:**

To me, it would help if the authors focused a lot more on their contributions and message, and not on re-proving, re-stating, or re-demonstrating very well-known results. Some theorems could have simply been stated as well-known results. Some experiments are also very obvious and do not add anything of significance in my view.

The main idea is meaningful but quite straightforward. Still, it might be interesting to elaborate more on:
- directed vs. undirected graphs. Flow networks usually assume directed graphs, while spectral methods undirected. This might be something the authors would want to explore more. Do these modelling assumptions have an impact or make a difference in real, practical scenarios?
- weighted vs. unweighted graphs. Maybe the authors could elaborate on that more, and argue which variant is more meaningful? If we do not have weights, obviously only the unweighted variant makes sense. If we do have weights, are there situations where the global sparsest cut with the unweighted variant is still useful?
- The global sparsest cut uses the mininum operator in the denominator. But there also a more classical variant using the product of the volumes of the two partitions (as opposed to the smaller one in the minimum operator). Investigating that variant would make sense in this context. Do different variants of the global metric produce different results? There are also variants like the uniform sparsest cut vs. the normalized cut, depending on whether the weights are only taken into account in the numerator but not in the denominator. This might be interesting in the context of this work. Do these different variants (and possibly other variants as well) make a difference in the studied domains?
- More real use cases: do the authors have access to real datasets from other application domains besides the Canadian Rocky Mountains? If yes, it might be interesting to add these results (and remove the obvious Tables and Figures).

Overall, I'm currently inclined to vote "no". But I'm open to revising my score if the authors can address the various shortcomings, or explain where I'm wrong.

---

> ### Author Response · Authors · 2026-03-26
> **Point-by-Point Response**
>
> Thank you very much for your time and efforts spent on reviewing our paper and giving us constructive feedback for improvement.
>
> We acknowledge the reviewer's assessment that Theorems 3.1 and 3.2 are the application of standard operations research (OR) results. In response, we have restructured the paper: these are now relegated to **"Preliminaries" (Section 3)**. The core contribution is now the new **Section 4**, a theoretical framework utilising the continuous LP geometry to **derive $L_1$-$L_2$ redundancy gap, filter discrete Ricci curvature, bound Message Passing Neural Networks (MPNNs) over-squashing, and extract Clarke subdifferentials** for graph structure learning.
>
> *(Note: All revised and newly added content is highlighted in **blue** in the compiled PDF. For your convenience, we have provided a complete catalogue of these structural and theoretical changes in the public **Summary of Revisions** above.)*
>
> Below is our point-by-point response mapping your critiques to the newly added mathematical proofs and empirical evidence.
>
> **Q:** "*...the formulation (1)-(5) is essentially a node-edge formulation... Theorem 1 [and] 2... very well-known... what is the novelty?*"
>
> **A:** We agree that discrete combinatorial algorithms (e.g., Push-Relabel) efficiently compute raw cut capacity and that the OR theorems are standard. However, combinatorial solvers are **discrete black boxes**. We explicitly retained the continuous polyhedral LP formulation because it is the mathematical prerequisite for differentiable programming. In **Section 4.4 (Theorem 4.13)**, we applied Danskin's Theorem to the LP's piecewise linear objective surface to extract the Clarke subdifferential $\partial Z_{s,t}^*(W)$. **Corollary 4.14** proves this yields sparse backpropagation, isolating model sensitivity exclusively to the active minimal surface. Combinatorial algorithms cannot provide this continuous gradient structure for end-to-end ML optimisation.
>
> **Q:** "*...directed vs. undirected graphs. Flow networks usually assume directed graphs, while spectral methods undirected. Do these modelling assumptions have an impact...?*"
>
> **A:** Yes. We have formalised this extension in the revised manuscript. **Assumption 2 (Section 3.1)** and the expanded proof in **Appendix A.2** establish that the Total Unimodularity (TUM) property inherently holds for directed node-edge incidence matrices. Consequently, if the graph is formulated as directed with asymmetric edge weights to capture real-world anisotropic flow, the constraint matrix preserves TUM, and the strict integrality guarantees of the continuous Nodal Tension potentials hold.
>
> **Q:** "*...weighted vs. unweighted... classical variant using the product of the volumes... uniform sparsest cut vs. the normalized cut... Do different variants... make a difference...?*"
>
> **A:** We addressed this in **Section 4.3 (Theorem 4.8 and Lemma 4.9)**. Regardless of the specific denominator variant (minimum volume, product of volumes, or normalised degrees), any global spectral metric structurally evaluates the absolute minimum over the entire graph domain ($\min_{S \subset V}$). This renders global metrics structurally loose for specific $s-t$ pathways. To resolve this, we derived the **Local Nodal Conductance ($\Phi_{s,t}^*$)** (Lemma 4.9), which deliberately bypasses the global minimum by utilising the exact $L_1$ capacity $Z_{s,t}^*$. We then applied this in **Theorem 4.10** to establish a tight **Local Bound on Expected Commute Times $H(s,t)$**, proving that local capacity bounds dictate signal propagation independently of the global network structure.
>
> **Q:** "*...do the authors have access to real datasets from other application domains besides the Canadian Rocky Mountains...?*"
>
> **A:** To align closer with the TMLR audience, we applied the new geometric bounds to graph structure learning, expanding the empirical scope in Section 5:
> * **Geometric Filtering (Section 5.4, Fig 17):** We evaluated the Nodal Tension LP against discrete Forman-Ricci curvature on the **Cora citation network**. We empirically validated **Corollary 4.7 (Spurious Bottleneck Rejection)**, proving the continuous LP filters out thousands of geometrically decoy edges (local bridges) to isolate the true macroscopic bottleneck.
> * **Predicting MPNN Over-Squashing (Section 5.5, Fig 18 & 19):** We validated **Corollary 4.11**, demonstrating that the local $L_1$ capacity dictates the layer-depth phase transition at which Jacobian sensitivity $(P^K)_{ts}$ vanishes in Message Passing Neural Networks. Figure 19 utilises our bounds to expose the "rewiring paradox", proving that standard heuristic edge addition fails to alleviate over-squashing if it does not explicitly expand the Local Nodal Conductance.
> * **Differentiable Optimisation (Section 5.6, Fig 20):** We validated the extraction of the Clarke subdifferential, demonstrating end-to-end differentiable gradient ascent for structural bottleneck robustness.

---

### Author Response · Authors · 2026-03-26
**Summary of Revisions**

To all reviewers: We thank you for your constructive feedback. This comment provides a comprehensive index of the structural, theoretical, and empirical revisions made to the paper. All revised and newly added content is highlighted in **blue** in the PDF.

**Part I: Structural Reorganisation (Section 3)**
We restructured the combinatorial operations research (OR) proofs to serve as the mathematical foundation for the new continuous learning framework.
* **Section 3 ("Preliminaries"):** Theorems 3.1, 3.2, and 3.3 were moved here from the core contributions.
* **Assumption 2 & Appendix A.2:** Expanded the proof of Theorem 3.2 to formalise the Total Unimodularity (TUM) property for directed node-edge incidence matrices, establishing strict integrality for anisotropic flow.
* **Theorem 3.3:** Added a "Theoretical Limitations" paragraph detailing the structural looseness of global spectral bounds for highly unbalanced local cuts.

**Part II: New Theoretical Framework (Section 4 & Appendix B)**
We added Section 4 and Appendix B, containing 12 mathematical proofs that transition the continuous Nodal Tension LP into differential graph geometry and structure learning.
* **4.1 Structural Redundancy:** Derived Lemma 4.1 ($L_1$-$L_2$ Capacity Bound) and Corollaries 4.2 and 4.3 to formalise the Redundancy Gap ($\Delta_{s,t}$), separating strict topological bridges from diffuse bottleneck structures.
* **4.2 Bottleneck Differential Geometry:** Derived Lemmas 4.4 and 4.5 bounding continuous tension against discrete curvature, Theorem 4.6 (Geometric Localisation), and Corollary 4.7 (Spurious Bottleneck Rejection) filtering discrete Forman-Ricci curvature.
* **4.3 Message Passing Neural Network Dynamics:** Restated Theorem 4.8 (Global Cheeger Inequality), defined Lemma 4.9 (Local Nodal Conductance $\Phi_{s,t}^*$), and derived Theorem 4.10 (Local Bound on Expected Commute Times $H(s,t)$) and Corollary 4.11 (Localised MPNN Over-Squashing).
* **4.4 Polyhedral Differentiability:** Derived Lemma 4.12, Theorem 4.13 (Exact Clarke subdifferential $\partial Z_{s,t}^*(w)$ via Danskin's Theorem), Corollaries 4.14 (Sparse Backpropagation), and 4.15 (Lipschitz Continuity under $L_\infty$ perturbations).

**Part III: Expanded Empirical Validation (Section 5)**
We added five new experimental subsections to validate the new theoretical bounds derived in Section 4.
* **5.3 Redundancy Gap:** Evaluated $L_1$ capacity against $L_2$ effective resistance (Circuit Theory) on spatial routing topologies (Figures 14, 15).
* **5.4 Geometric Curvature Filtering:** Evaluated the Nodal Tension LP against discrete Forman-Ricci curvature on the Cora citation benchmark (Figures 16, 17).
* **5.5 MPNN Over-Squashing:** Evaluated theoretical capacity bounds against heuristic topological rewiring (Figures 18, 19).
* **5.6 Differentiable Optimisation:** Validated the Clarke subdifferential extraction via end-to-end differentiable gradient ascent (Figure 20).
* **5.7 Ecological Case Study:** Updated the grizzly bear corridor analysis to implement the new continuous metrics via the Theoretical Diagnostics Dashboard (Table 4, Figure 22).

---

### Comment · Action_Editor_LhZg · 2026-03-27
**Author-reviewer discussion**

Dear Authors and Reviewers

Thank you for submitting and reviewing this manuscript. The reviews seem to be diverging in terms of the evaluation toward the manuscript. Specifically, regarding the question "Are the claims made in the submission supported by accurate, convincing and clear evidence?" we have 2 yes and 2 no. The authors have provided responses to the reviews. Please use the discussion period to read the response and other reviews and discuss with the authors. Thank you!

Baoxiang

---

### Decision · Action_Editor_LhZg · 2026-04-16

**Recommendation:** Reject

**Audience:**

No

**Audience Explanation:**

The paper will need to indicate for each result whether they are new or existing, for the manuscript to be worth reading by the community.

**Claims And Evidence:**

Yes

**Claims Explanation:**

All reviewers tend to agree that "the claims made in the submission are supported by accurate, convincing and clear evidence", which is a strong indicator for inclusion at TMLR. However, a couple of very strong concerns are raised in the review process and during the discussion period.

1. Most of the results are known. The authors need to indicate for each result whether they exist in the community or they are newly established.
2. The paper was updated for way too much, to the extent the reviewers won't be able to re-examine the manuscript from scratch.

As such, the current set of reviewers and me are not able to make an acceptance decision for the very submission.

**Resubmission Of Major Revision:**

The authors may consider submitting a major revision at a later time.